# Developing a machine learning model for accurate nucleoside hydrogels prediction based on descriptors

Weiqi Li [1,2], Yinghui Wen [1,2], Kaichao Wang[1,2], Zihan Ding [1], Lingfeng Wang [1], Qianming Chen [1], Liang Xie [1] ✉, Hao Xu [1] ✉ & Hang Zhao [1] ✉

Supramolecular hydrogels derived from nucleosides have been gaining significant attention in the biomedical field due to their unique properties and excellent biocompatibility. However, a major challenge in this field is that there is no model for predicting whether nucleoside derivative will form a hydrogel. Here, we successfully develop a machine learning model to predict the hydrogel-forming ability of nucleoside derivatives. The optimal model with a 71% (95% Confidence Interval, 0.69–0.73) accuracy is established based on a dataset of 71 reported nucleoside derivatives. 24 molecules are selected via the optimal model external application and the hydrogel-forming ability is experimentally verified. Among these, two rarely reported cation-independent nucleoside hydrogels are found. Based on their self-assemble mechanisms, the cation-independent hydrogel is found to have potential applications in rapid visual detection of $Ag^+$ and cysteine. Here, we show the machine learning model may provide a tool to predict nucleoside derivatives with hydrogel-forming ability.

Recently, supramolecular hydrogels derived from nucleosides have attracted increasing attention in the biomedical field due to their manifold noncovalent interactions, unique properties, and excellent biocompatibility[1–3]. Since Bang et al. reported in 1910 that concentrated solutions of guanylic acid could form a gel[4], significant progress in nucleoside-based hydrogels have been developed and used in various applications, including drug delivery, biosensors, and tissue engineering[5–11]. For example, Lehn et al. showed that a guanosine derivative was a stable supramolecular hydrogel in the presence of metal cations and provided a highly selective and controllable release of bioactive substances, making it an attractive option for drug delivery[9]. Davis et al. designed long-lived guanosine-borate hydrogels, enabling sustained drug release[10]. Based on these findings, our group developed a series of nucleoside-based dual-function hydrogels that show potential applications in the field of biomedicine, including

wound healing and cancer treatment[2,5–7]. A major challenge in this field is the limited comprehension of how to anticipate whether a nucleoside derivative will form a hydrogel. Design suggestions are frequently made, but gelators are usually discovered unintentionally or through the synthetic modification of an existing gelator[12,13]. This fundamental constraint results from the little understanding of the association between the nucleoside structure and hydrogel forming ability.

Machine learning (ML) is a powerful technology that allows model to automatically learn from data and improve their performance over time, helping to automate and optimize processes and improve decision-making[14–16]. The advantage of ML in predicting the hydrogel-forming ability of molecules lies in its ability to learn the structure-property relationships through high-dimensional data, which can be used to excavate new gelators[17–19]. In recent years, substantial advances have been made in the application of ML in hydrogels[17–19]. For example,

[1]State Key Laboratory of Oral Diseases, National Clinical Research Center for Oral Diseases, Research Unit of Oral Carcinogenesis and Management, Chinese Academy of Medical Sciences, West China Hospital of Stomatology, Sichuan University, Chengdu, Sichuan 610041, PR China. [2]These authors contributed equally: Weiqi Li, Yinghui Wen, Kaichao Wang. ✉e-mail: lxie@scu.edu.cn; hao.xu@scu.edu.cn; zhaohangahy@scu.edu.cn

Li et al. developed ML models to learn the correlation between chemical features and the dipeptide hydrogel-forming ability of peptide-like molecules with algorithms including random forest (RF), gradient boosting (GB), and logistic regression (LR)[18]. However, there are no models for accurate nucleoside hydrogels prediction due to the complexity of the self-assemble process of nucleoside derivatives forming hydrogels[13].

Here, we show an optimal ML model which was successfully developed to predict the hydrogel-forming ability of nucleoside derivatives based on feature selection, hyperparameter optimization and algorithm comparison. The simplified illustration of this study is shown in Fig. 1. In this work, the optimal ML model may provide a tool to predict nucleoside derivatives with hydrogel-forming ability in the future.

## Results and discussion

### An optimal ML model was constructed for nucleoside derivatives to predict the hydrogel-forming ability

All the published nucleoside derivatives and the information on whether they have the hydrogel-forming ability were collected by systematic literature review, and 71 molecules were included in the dataset (gelator, $n = 38$, and non-gelator, $n = 33$, Supplementary Data 1, Supplementary Data 2)[5,20–36]. As only a few dozen have clarified hydrogel-forming ability, more nucleoside hydrogels are urgently needed to discover. Molecular descriptors are used as features to construct the models, after removing missing values, 4175 descriptors were obtained for the 71 nucleoside derivatives (Supplementary Data 3, Fig. 2d, Supplementary Fig. 1). Subsequently, a three-step feature selection was utilized based on the 4175 descriptors to avoid overfitting and improve the model accuracy (Fig. 2a). Feature selection including rank-sum test ($n = 144$, Fig. 2b, Supplementary Figs. 2, 3), Spearman correlation ($n = 40$, Fig. 2c, Supplementary Fig. 4) and ML algorithm-based recursive feature elimination (RFE: Extreme gradient boosting, XGBoost, $n = 16$; LR, $n = 24$; Decision tree, DT, $n = 30$; RF, $n = 37$, Fig. 2e, Supplementary Data 4)[37–41]. Taken together, to construct the prediction models comprehensively, different mathematical representations of molecules based on descriptors were used to build prediction models with four ML algorithms, details for the built models were shown in Supplementary Methods 2.1 and Supplementary Table 1. In this study, test accuracy and area under the curve (AUC) were mainly focused on, and the results of precision, recall, and F1 score were used as auxiliary indicators (Supplementary Methods 2.2).

The model performance of LR based on 24 descriptors after RFE, not only provided better results of test accuracy ($0.71 \pm 0.02$) and AUC ($0.84 \pm 0.02$), but also had higher recall ($0.95 \pm 0.01$) and F1 score ($0.78 \pm 0.01$) (Fig. 3a–d, Table 1). So, the model of LR based on 24 descriptors after RFE was finally chosen as the optimal model (Supplementary Data 5). By constructing prediction models of 71 nucleoside derivatives, we demonstrated that the ML model could indeed help in the prediction of the hydrogel-forming ability of nucleoside derivatives. Based on the optimal model, the important features, namely the 24 molecular descriptors, were mainly clustered in 2D matrix-based descriptors, edge adjacency indices, P_VSA-like descriptors, 2D atom pairs, 2D autocorrelations, atom-centered fragments, functional group counts, and pharmacophore descriptors (Fig. 3e, Supplementary Table 2 and Supplementary Data 5). Cross validation without independent test set may overestimate model predictivity. Therefore, we performed a sensitivity analysis on the dataset. Cluster analysis was used to randomly select 15 out of 71 nucleoside derivatives (20%) for the test set (once, not for training, Supplementary Methods 2.3, and Supplementary Figs. 5–7), and the remaining 56 nucleoside derivatives (80%) were used as the training set (with five-fold cross validation). The results were consistent with our previous five-fold cross-validation of 71 nucleoside derivatives, the three-step feature selection based logistic regression (LR-RFE) performed better in both the training set (with five-fold cross-validation: validation accuracy: $0.70 \pm 0.02$, AUC: $0.84 \pm 0.02$, Supplementary Table 3) and test set (accuracy: 0.67, AUC: 0.81, Table 2).

The importance of features was determined by regression coefficients and the permutation feature importance (PFI). We calculated the regression coefficient of LASSO regression for 4175 molecular descriptors and got the feature importance of 70 molecular descriptors after feature selection by LASSO (Supplementary Data 6). The regression coefficients of the 24 molecular descriptors are the feature importance of the optimal model (Fig. 3e, Supplementary Table 2). In addition, the PFI results (mean accuracy decrease) of the 24 molecular descriptors in the optimal model were also provided (Supplementary Table 2, Supplementary Fig. 8). Notably, according to the regression coefficients, there were four molecular descriptors that the feature importance is more than 0.1 in the LR model. These descriptors were mainly related to hydrogen bonding, molecular polarity, and lipid solubility, consisted with previous studies reporting, which we believe they are the key descriptors of the gelator properties[12,13,17,19,42] (Fig. 3e, Supplementary Discussion 1.1).

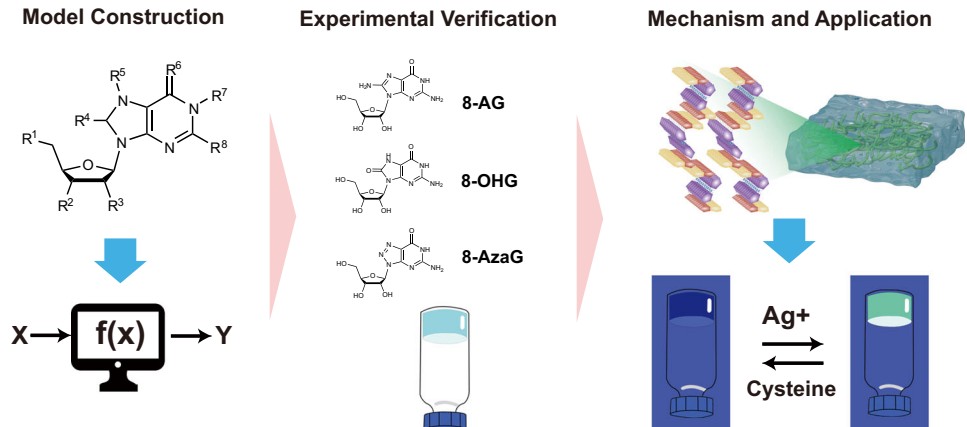

**Fig. 1 | To predict the ability of nucleoside derivatives to form hydrogels based on machine learning.** An optimal model was constructed for nucleoside derivatives hydrogel-forming ability prediction, and potential gelators were selected based on the optimal model external application and the hydrogel-forming ability were experimentally verified. Besides, the self-assembly mechanism of the cation-independent hydrogel was explored, which could be applied in rapid visual detection of Ag⁺ and cysteine.

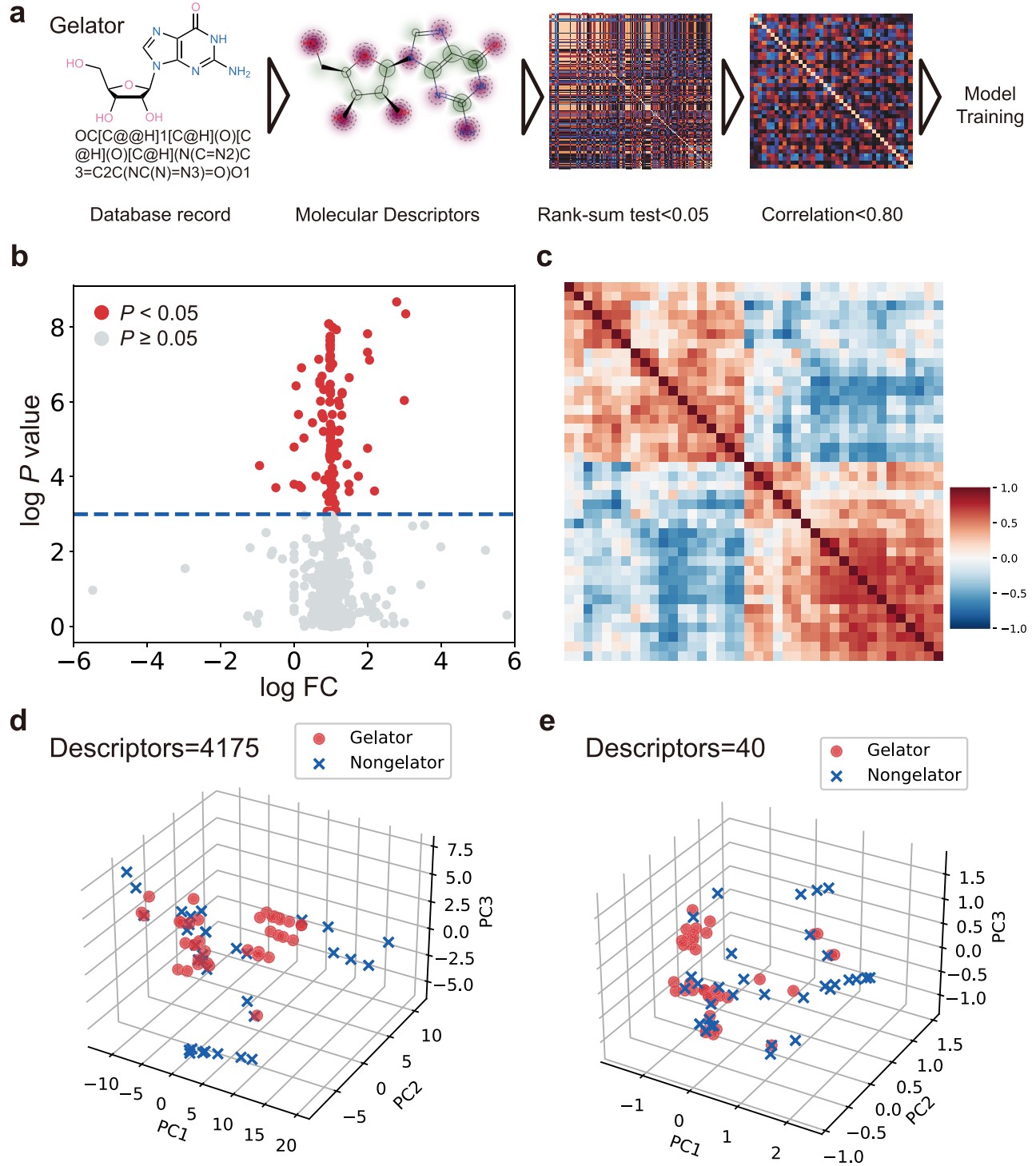

**Fig. 2 | The flowchart of model construction and feature selection of the descriptors. a** Flow chart of model construction. The results of 4175 descriptors were initially obtained, 144 descriptors with significant differences ($P < 0.05$) were selected by the rank-sum test, and 40 descriptors finally remained after excluding one of the pairs of descriptors with a correlation coefficient higher than 0.8 (Rho > 0.8) with Spearman correlation. **b** The results of the rank-sum test. With the logarithm of the $P$-value (log $P$-value) for the vertical coordinate, and the logarithm of the fold change (log FC) between the mean values of the gelator group and non-gelator group for the horizontal coordinate. **c** 40 descriptor correlation heatmaps. All correlations between descriptors were less than 0.80. **d** Three-dimensional (3D) principal component analysis (PCA) of 71 nucleoside derivatives with 4175 descriptors. The results of the PCA visualization with 4175 descriptors displayed of the gelator and non-gelator groups. **e** 3D PCA of 71 nucleoside derivatives with 40 descriptors. The results of the 3D PCA visualization with 40 descriptors displayed of two groups.

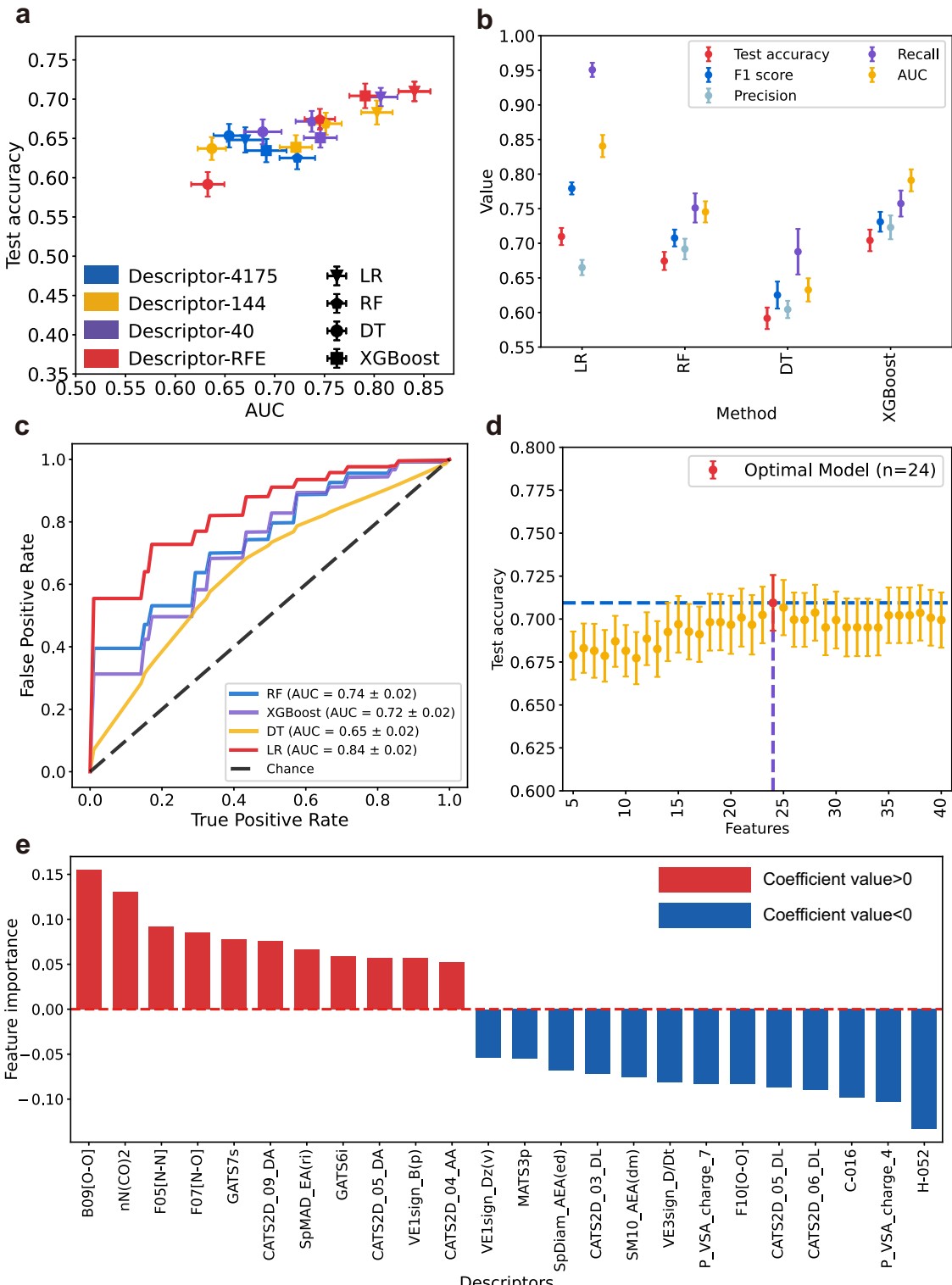

**Fig. 3 | Evaluation indexes of different models and feature importance of optimal models. a** A scatterplot showed the distribution of AUC (area under the curve) and test accuracy for all models. The 4-point shapes represent different ML algorithms: extreme gradient boosting (XGBoost), logistic regression (LR), decision tree (DT), and random forest (RF). Descriptor's part: Initially obtained 4175 descriptors,144 descriptors after rank sum test, 40 descriptors after correlation coefficient selection, and descriptors after recursive feature elimination (RFE). The optimal number of descriptors for RFE of each machine learning (ML) algorithm is different (XGBoost, $n = 16$; LR, $n = 24$; DT, $n = 30$; RF, $n = 37$), data are mean values ± standard error of the mean (SEM). **b** Evaluation indexes of four algorithms using descriptors after RFE. Combining the results of test accuracy, F1 score and AUC, data are mean values ± SEM. **c** Receiver operating characteristic curve for the four algorithms (LR, DT, RF, and XGBoost) using descriptors after RFE. **d** The RFE results of the LR models based on different descriptors within the 40 descriptors, indicated that LR with 24 descriptors had the best performance, data are mean values ± SEM. **e** The results of feature importance of 24 descriptors for the optimal LR model based on the regression coefficients.

Although we found a series of molecular descriptors related to hydrogel-forming ability, the interpretations of some of them are still not easy. According to the previous studies[12,13,17–19,42], four easy-to-understand molecular descriptors related to the gelator properties were selected, such as water solubility and lipophilicity[43–45], that may be relevant to the hydrolgel-forming ability of nucleoside derivatives. The values' distributions of these molecular descriptors had no significant differences ($P > 0.05$) between the gelator and non-gelator groups, which suggested that these molecular descriptors were not related to the hydrogel-forming ability of nucleoside derivatives (Supplementary Fig. 9). Therefore, although the molecular descriptors chosen in the optimal model are not easy to understand, they are valuable and important to predict the hydrogel-forming ability of nucleoside derivatives in this work, and a further exploration of these molecular descriptors is necessary in the future.

For the first part, a dataset of 71 nucleoside derivatives with the information of whether have the hydrogel-forming ability was converted into feature matrices using 4175 molecular descriptors. After feature selection and hyperparameter optimization, four classifier algorithms were included for predicting hydrogel-forming ability and selected an optimal model with 71% accuracy.

**Table 1 | The result of AUC (Area under Curve) and test accuracy for all models based on 71 nucleoside derivatives**

| Models | Features | Test Accuracy | | AUC | |
|---|---|---|---|---|---|
| | | Mean | SEM | Mean | SEM |
| DT* | Descriptor_4175 | 0.65 | 0.01 | 0.65 | 0.02 |
| LR | Descriptor_4175 | 0.65 | 0.02 | 0.67 | 0.02 |
| RF | Descriptor_4175 | 0.63 | 0.01 | 0.72 | 0.02 |
| XGBoost | Descriptor_4175 | 0.63 | 0.01 | 0.69 | 0.02 |
| DT | Descriptor_144 | 0.64 | 0.01 | 0.64 | 0.01 |
| LR | Descriptor_144 | 0.68 | 0.02 | 0.80 | 0.02 |
| RF | Descriptor_144 | 0.67 | 0.01 | 0.75 | 0.02 |
| XGBoost | Descriptor_144 | 0.64 | 0.02 | 0.72 | 0.02 |
| DT | Descriptor_40 | 0.66 | 0.02 | 0.69 | 0.02 |
| LR | Descriptor_40 | 0.70 | 0.01 | 0.81 | 0.02 |
| RF | Descriptor_40 | 0.67 | 0.01 | 0.74 | 0.02 |
| XGBoost | Descriptor_40 | 0.65 | 0.01 | 0.75 | 0.02 |
| DT | Descriptor_ REF # | 0.59 | 0.02 | 0.63 | 0.02 |
| LR | Descriptor_ REF # | 0.71 | 0.01 | 0.84 | 0.02 |
| RF | Descriptor_ REF # | 0.67 | 0.01 | 0.75 | 0.02 |
| XGBoost | Descriptor_ REF # | 0.70 | 0.02 | 0.79 | 0.02 |

*LR Logistic regression, DT Decision tree, RF Random forest, XGBoost Extreme gradient boosting, SEM Standard error of the mean.
#Descriptors-REF: Recursive feature elimination (REF) has different optimal descriptors for different Algorithms: LR, $n = 24$; XGBoost, $n = 16$; DT, $n = 30$; RF, $n = 37$.

**Twenty-four molecules were selected based on the optimal model for external application and the hydrogel-forming ability were experimentally verified**

To test the optimal ML model, external application is crucial. Here, we screened 7257 nucleoside derivatives based on their three-dimensional (3D) similarity from the PubChem database. A grouped box plot (Supplementary Fig. 10) of the 24 descriptors in optimal model (LR) and a 2D-PCA (Supplementary Fig. 11) based on these 24 descriptors showed that their multidimensional features of 7257 nucleoside derivatives are in or near the ranges for 71 nucleoside derivatives used for model construction. The optimal model (LR based on 24 descriptors) was applied to the 7257 nucleoside derivatives and their predictions were ranked based on the prediction probability (Supplementary Data 7).

To validate the model and consider possible subsequent applications, the nucleoside derivatives with the top 10% prediction probability of gelators were selected and 12 nucleoside derivatives (**1**, 1-[3,4-Dihydroxy-5-(hydroxymethyl) oxolan-2-yl]-1,3,5-triazinane-2,4,6-trione, DTT; **2**, xanthosine, XTS; **3**, guanine 5′-monophosphate, GMP; **4**, inosine 5′-monophosphate, IMP; **5**, 5-fluorouridine, 5-FUR; **6**, 8-aminoguanosine, 8-AG; **7**, 2′-deoxyguanosine 5′-monophosphate, dGMP; **8**, 8-hydroxyguanosine, 8-OHG; **9**, 8-azaguanosine, 8-azaG; **10**, inosine-5′-carboxylic acid; I-5′-CA; **11**, 2′-amino-2′-deoxyguanosine, 2′-NH₂-dG, and **12**, 2′-O-Methylguanosine, 2′-OMe-dG) were selected in a relatively homogeneous manner based on our experience and the costs of obtaining and synthesizing nucleoside derivatives (Fig. 4a, Supplementary Fig. 12). To validate the hydrogel-forming ability of the 12 nucleoside derivatives, as well as the strength and stability of the hydrogels, we conducted the tube-inversion tests (Fig. 4a, Supplementary Figs. 13–15, Supplementary Data 8). Furthermore, to be more persuasive, we had additionally chosen 12 nucleoside derivatives predicted to have no hydrogel-forming ability and it was experimentally validated subsequently (Fig. 4a, Supplementary Figs. 16–18, Supplementary Data 8).

To 12 nucleoside derivatives with the top 10% prediction probability, the result shows 10 nucleoside derivatives (**1, 3, 4, 6, 7, 8, 9, 10, 11**, and **12**) formed hydrogels, while the two others (**2** and **5**) did not (Fig. 4b, Supplementary Figs. 13–15, Supplementary Data 8), suggesting the success rate of forming hydrogels is 83.33% (10/12). Specifically, **1, 3, 4, 7, 10**, and **12** formed hydrogels in the presence of AgNO₃. **6** and **8** self-assembled into hydrogels in H₃BO₃ and Tris solution, as well as NaB(OH)₄ and KB(OH)₄ solutions. **9** formed hydrogels in H₃BO₃ and Tris solution, as well as AgNO₃ solution. **11** could self-assemble into hydrogels in KCl, NaCl, NaB(OH)₄ and KB(OH)₄ solutions. Among these nucleoside derivatives, eight nucleoside derivatives (**1,6,7,8, 9, 10,11**, and **12**) have not been reported as gelators. To 12 nucleoside derivatives with low prediction probability, the results show that 10 of the 12 nucleoside derivatives didn't form hydrogels (**14–23**), while the two others formed (**13** and **24** formed hydrogels in AgNO₃ solution, accuracy rate is 83.33%, Supplementary Data 8 and Supplementary Figs. 16–19). Thus, for 24 nucleoside derivatives from external dataset,

**Table 2 | The result of AUC (Area under Curve) and accuracy for models based on test set of 14 nucleoside derivatives**

| Models | Features | Test set performance | | | | |
|---|---|---|---|---|---|---|
| | | Accuracy | F1 Score | Precision | Recall | AUC |
| DT | Descriptor_ REF # | 0.60 | 0.67 | 0.75 | 0.60 | 0.60 |
| LR | Descriptor_ REF # | 0.67 | 0.76 | 1.00 | 0.61 | 0.81 |
| RF | Descriptor_ REF # | 0.53 | 0.59 | 0.63 | 0.56 | 0.53 |
| XGBoost | Descriptor_ REF # | 0.60 | 0.57 | 0.50 | 0.67 | 0.61 |

*LR Logistic regression, DT Decision tree, RF Random forest, XGBoost Extreme gradient boosting.
#Descriptors-REF: Recursive feature elimination (REF) has different optimal descriptors for different Algorithms: LR, $n = 34$; XGBoost, $n = 33$; DT, $n = 23$; RF, $n = 26$.

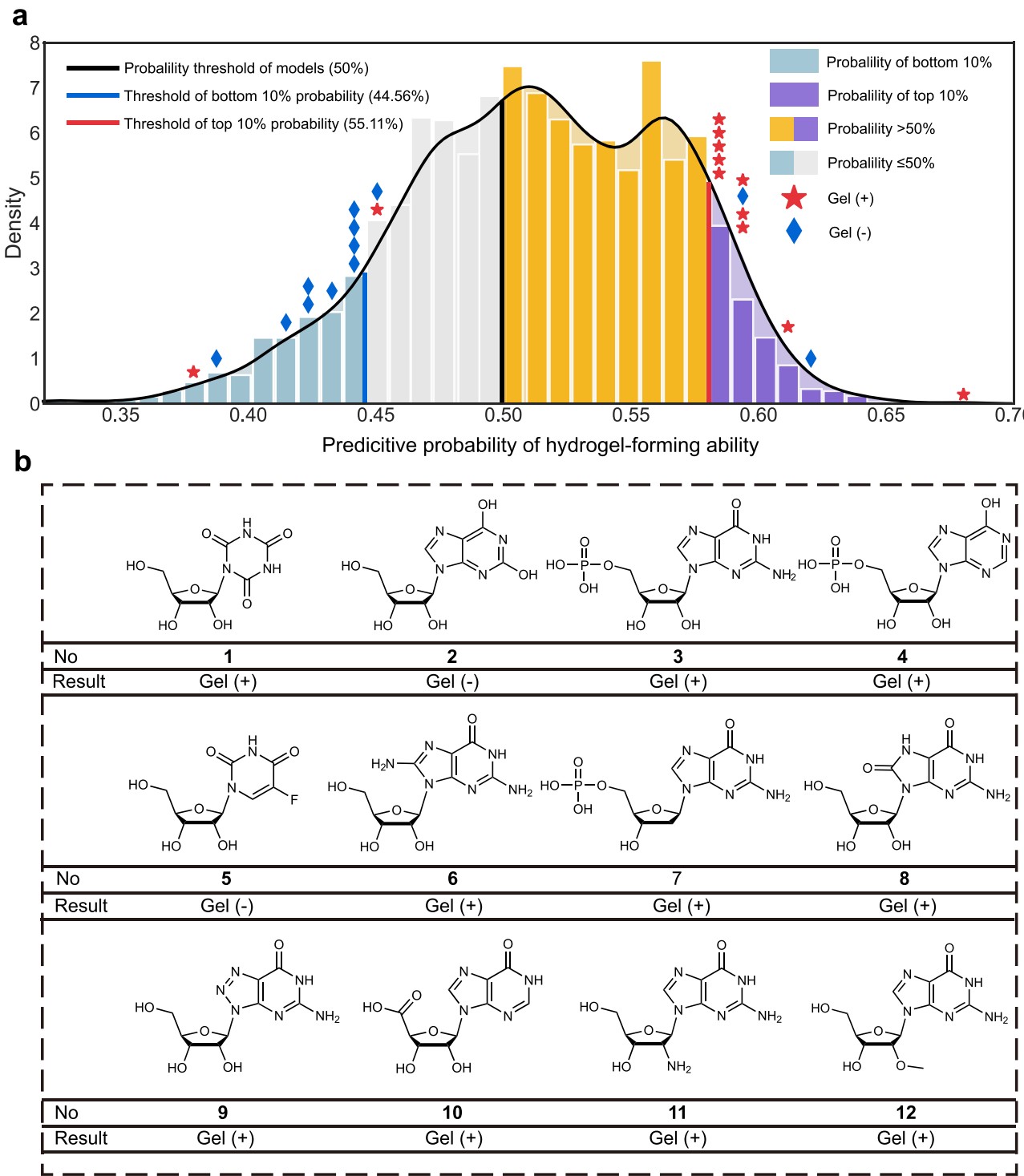

**Fig. 4 | Prediction and verification of untested nucleoside derivatives. a** 24 nucleoside derivatives were selected (12 high probability and 12 low probability) in a relatively homogeneous manner based on our experience and the costs of obtaining and synthesizing nucleoside derivatives. **b** 12 nucleoside derivatives with high probability of hydrogel-forming ability were selected. The result shows 10 nucleoside derivatives (**1, 3, 4, 6, 7, 8, 9, 10, 11**, and **12**) formed hydrogels, while the two others (**2** and **5**) did not. **1**, 1-[3,4-Dihydroxy-5- (hydroxymethyl) oxolan-2-yl] −1,3,5-triazinane-2,4,6-trione, DTT; **2**, xanthosine, XTS; **3**, guanine 5′-monophosphate, GMP; **4**, inosine 5′-monophosphate, IMP; **5**, 5-fluorouridine, 5-FUR; **6**, 8-aminoguanosine, 8-AG; **7**, 2′-deoxyguanosine 5′-monophosphate, dGMP; **8**, 8-hydroxyguanosine, 8-OHG; **9**, 8-azaguanosine, 8-azaG; **10**, inosine-5′-carboxylic acid; I-5′-CA; **11**, 2′-amino-2′-deoxyguanosine, 2′-NH$_2$-dG, and **12**, 2′-O-methylguanosine, 2′-OMe-dG.

the accuracy of the optimal model for predicting hydrogel-forming ability was 83.33% (20/24).

In the process of validating the synthesis of nucleoside hydrogels and exploring the synthesis conditions, it was surprising that **6** and **8**

exhibited the same hydrogel-forming abilities and can self-assemble into hydrogels (8AG-T and 8OHG-T hydrogels) through a simple one-pot method in the absence of cations, and the hydrogels display long lifetime stability of 6 months (Fig. 5a). Although guanosine-derived

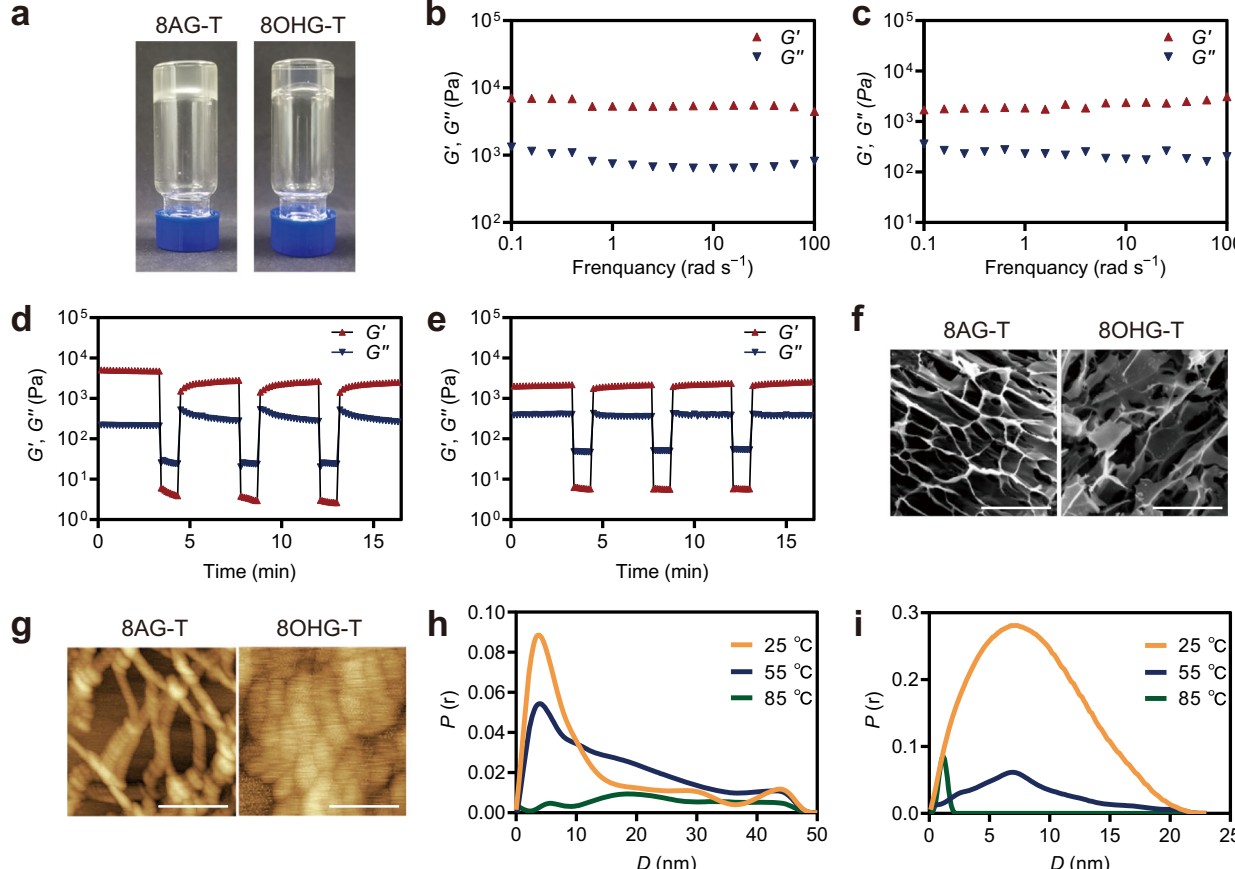

**Fig. 5 | The characterizations of hydrogels. a** Photographs of 8AG-T and 8OHG-T hydrogels were prepared for 6 months. **b, c** Evolution of G' and G" as a function of frequency for of 8AG-T (**b**) and 8OHG-T (**c**) hydrogels. **d, e** The self-healing of 8AG-T (**d**) and 8OHG-T (**e**) hydrogels by rheological measurements. **f** Scanning electron microscopy (SEM, scale bar: 50 μm) images of 8AG-T and 8OHG-T hydrogels. **g** Atomic force microscopy (AFM, scale bar: 200 nm) images of 8AG-T and 8OHG-T hydrogels. **h, i** The pair distances distribution functions (PDDF) profiles from variable-temperature small-angle X-ray scattering (VT-SAXS) experiments of 8AG-T (**h**) 8OHG-T (**i**) hydrogels.

supramolecular hydrogels have potential applications in the fields of drug delivery, targeted release, and tissue engineering, the need for excess cations hinders their widespread applications. To date, the reports on developing cation-independent guanosine-derived supramolecular hydrogels are rare. This has piqued our great interest to further explore these two unknown nucleoside hydrogels (8AG-T and 8OHG-T hydrogels).

To better understand two cation-independent hydrogels, G-T, 8AG/8OHG/G-Na$^+$, and 8AG/8OHG/G-K$^+$ hydrogels were prepared as controls. Different from the long-term stability of 8AG-T and 8OHG-T hydrogels, the 8AG-K$^+$ and 8OHG-K$^+$ hydrogels are weak and collapse within hours, and a precipitate generates in the G-T hydrogel within hours. These findings suggest that the 8AG-T and 8OHG-T hydrogels are more stable than the 8AG-K$^+$ and 8OHG-K$^+$ hydrogels, respectively, but the opposite is true for the G-T/Na$^+$/K$^+$ hydrogels (Supplementary Fig. 14 and Supplementary Fig. 20). To further explore the solid-like state of the hydrogels, rheological measurements were performed. The 8AG-T and 8OHG-T hydrogels possessed higher storage modulus (G') values than loss modulus (G") values, suggesting that they were solid-like hydrogels over the entire applied frequency range (Fig. 5b, c). The G' values of the 8AG-T and 8OHG-T hydrogels were higher than those of the 8AG-Na$^+$/K$^+$ and 8OHG-Na$^+$/K$^+$ hydrogels, respectively (Supplementary Fig. 21). Supplementary Figs. 22–23 illuminate the shear-thinning capabilities and gel–sol transitions of these hydrogels, respectively. The self-healing ability of hydrogels is an important property for biomedical applications. The circle strain time sweep

rheology assessments show that the 8AG-T and 8OHG-T hydrogels exhibit excellent self-healing properties, but the 8AG-K$^+$ and 8OHG-K$^+$ hydrogels do not (Fig. 5d, e, Supplementary Fig. 24). Therefore, the cation-independent 8AG-T and 8OHG-T hydrogels are stable and self-healing, which suggests extensive application prospects.

To characterize the microstructures of the hydrogels, scanning electron microscopy (SEM), atomic force microscopy (AFM) and variable-temperature small-angle X-ray scattering (VT-SAXS) measurements were performed. SEM images illustrate that the 8AG-T and 8OHG-T hydrogels are porous structures (Fig. 5f), and the porous structures of the 8AG-T hydrogel are finer ordered than 8OHG-K$^+$ hydrogel (Supplementary Fig. 25). Intertwined nanofibers and aggregated rod-like structures of 8AG-T and 8OHG-T hydrogels are observed under AFM, respectively (Fig. 5g). As shown in Supplementary Fig. 26, the 8AG-T hydrogel shows more robust fibers than 8AG-K$^+$ hydrogel, and the size of the rod-like structure in the 8OHG-T hydrogel is larger than that in the 8OHG-Na$^+$ and 8OHG-K$^+$ hydrogels. Conversely, the G-T hydrogel exhibits weaker fibers than the G-K$^+$ hydrogels. The VT-SAXS results reveal that at 25 °C, the 8AG-T hydrogel fibers were nanowires with diameters of approximately 12 nm, and the fibers disappeared as the temperature rose to 85 °C (Fig. 5h). The diameters of the rods in the 8OHG-T hydrogel were approximately 23 nm at 25 °C, and the rods turned into slender nanowires with diameters of approximately 2 nm at 85 °C (Fig. 5i). The fibers disappeared in the G-T hydrogel at 55 °C, while fibers even existed in the G-K$^+$ hydrogel at 95 °C (Supplementary Fig. 27). Therefore, these results

demonstrated that the 3D porous networks of the 8AG-T and 8OHG-T hydrogels were constructed by intertwined nanofibers and aggregated rod-like structures, respectively. According to the results of the tube-inversion test, rheological measurements and microstructures, the cation-independent hydrogels constructed by **6** and **8** exhibited more robust and self-healing properties than their cation-dependent hydrogels.

For the second part, the optimal ML model was applied to predict the hydrogel-forming ability of 7257 publicly available nucleoside derivatives. Twelve potential gelators were selected based on the optimal model for external application and the hydrogel-forming ability was experimentally verified. Among these, two rarely reported cation-independent nucleoside hydrogels were found.

## Self-assembly mechanism of the cation-independent hydrogel and its application in rapid visual detection of Ag$^+$ and cysteine

Davis et al. found the vicinal diol group in G could form monoesters and diesters with $H_3BO_3$, thus introducing borate diesters into the hydrogel network to improve the gel-forming[11]. Therefore, we speculated that the formation of dynamic borate diester bonds contributed to form stable and self-healing hydrogels in the absence of cations. To further confirm our hypothesis, firstly, we explored the hydrogelation of **6** and **8** in different conditions. The tube-inversion test results show that **6** and **8** fail to form hydrogels in NaCl, KCl, and $H_3BO_3$ solutions, but self-assemble into stable hydrogels in $H_3BO_3$ solution after adding Tris to avoid the hydrolysis of borate diesters in acidic solution (Supplementary Fig. 14, Supplementary Discussion 1.2). The presence of borate diester bonds was further confirmed by $^{11}B$ nuclear magnetic resonance (NMR) and Alizarin Red S (ARS) experiments (Fig. 6a, b, Supplementary Figs. 28, 29). Thus, these results indicated that the dynamic borate diester bonds helped form stable and self-healing hydrogels in the absence of cations.

Considering G-quartets generally require stabilization by templating alkali metal cations, such as K$^+$ [12], we further investigated whether **6** and **8** didn't self-assemble into G-quartets in the absence of cations. First, the thioflavin T (ThT) assay demonstrated that G-quartets exist in G-K$^+$ hydrogels but not in 8AG-T and 8OHG-T hydrogels (Fig. 6c, Supplementary Fig. 30, Supplementary Discussion 1.3). Second, Fig. 6d illustrates that there might be no G-quartets' stacking in the 8AG-T and 8OHG-T hydrogels based on the difference of positive peaks and troughs between 8AG/8OHG-T hydrogels and G-K$^+$ hydrogels. Then, the g-factor (ratio between circular dichroism and absorption intensities) was used to distinguish G-ribbon from G-quartet because the g-factor of the former is smaller than that of the latter. The ultraviolet (UV) spectra show that the g-factor values of the 8AG-T and 8OHG-T hydrogels were smaller than that of the G-K$^+$ hydrogel (Supplementary Figs. 31, 32, Supplementary Table 4), indicating that **6** and **8** might not form G-quartets in the 8AG-T and 8OHG-T hydrogels.

To investigate the self-assembly pattern of **6** and **8** at a molecular level, a single crystal of **6** was successfully obtained from a 70% dimethyl sulfoxide (DMSO) solution. Given the single molecule of **6** in its single-crystal structure, only one conformer of *anti* was observed (Fig. 6e). According to its hydrogen-bonding geometry (Supplementary Table 5), it is confined by six intramolecular hydrogen bonds (HBs), which is rare for nucleoside crystals. Notably, 8-NH$_2$ groups contribute to the construction of half of the six intramolecular HBs (N8-H8B···O4', N8-H8B···O5' and C2'-H2'···N8), which strengthen the *anti*-conformation of **6** (Supplementary Table 6). Theoretical calculations were performed to calculate the free energy differences of the *anti*/*syn*-conformations of **6** and **8**. The relative energies of *anti*-8AG relative to *syn*-8AG and *anti*-8OHG relative to *syn*-8OHG are −1.8 and −2.7 kcal mol$^{-1}$, respectively (Supplementary Figs. 33, 34), suggesting that both **6** and **8** favor the *anti*-sugar base conformation. $^1H$–$^1H$ nuclear overhauser effect (NOE) experiments also provided

unambiguous evidence concerning the *anti*-glycosidic bond preference of **6** as strong signal was observed between 2-NH$_2$ and H1' in 8AG-T hydrogels (Fig. 6f).

The conformations of *anti* and *syn* influence the self-assembling of guanosine derivatives. In the *anti*-conformation, the N2 and N3 positions allow for the formation of intermolecular H-bonds to form G-ribbons[12], while these positions in the *syn*-conformation are blocked by the sugar, which accelerates the formation of G-quartets and blocks the formation of ribbons[46]. Therefore, we hypothesized that **6** and **8** might form G-ribbons but not G-quartets to self-assemble into cation-independent hydrogels. The base-pair pattern of **6** in its single-crystal structure was used to test this hypothesis. The base-pair pattern is achieved by infinite chains of three intermolecular HBs (N8-H8A···O6, N2-H2B···O6 and N1-H1···N7) with a bending angle of 5.25° (Fig. 6g, Supplementary Fig. 35). Instead of forming a G-quartet in the base part, the involvement of 8-NH$_2$ in **6** enables the formation of multiple ribbon-like base-pair layers. These layers are connected by solvent molecules to form the final crystal structure of **6**. The holistic single crystal structures were shown in Fig. 6h, Supplementary Figs. 36–41, Supplementary Table 7, and Supplementary Discussion 1.4. Theoretical calculations results show that the relative energy of the G-quartet relative to the G-ribbon of **6** is 6.2 kcal mol$^{-1}$ (Supplementary Fig. 42), indicating that **6** favors forming a G-ribbon but not a G-quartet. Powder X-ray diffractometry (PXRD) analysis was performed to determine the stacking pattern of the hydrogels. The 8AG-T and 8OHG-T hydrogels displayed significant peaks at $2\theta \approx 28°$ (d = 3.3 Å; Fig. 6i), which was in line with the $\pi - \pi$ stacking of the benzene rings. In summary, we speculate that the self-assembly processes of the cation-independent 8AG-T and 8OHG-T hydrogels are as follows (Fig. 6j): dynamic borate ester bonds were initially formed. Later, the *anti*-glycosidic bond preference of **6** and **8** helped them form G-ribbon not G-quartet bonds due to intermolecular hydrogen bonds. The G-ribbon assembled via $\pi$–$\pi$ stacking and formed nanowires or rod-like structures. Finally, nanofibers intertwined, and rod-like structures aggregated by holding large amounts of water to eventually form the 8AG-T and 8OHG-T hydrogels, respectively.

Inspired by the molecular chaperones of G-quartet hydrogels, which could dock in a planar conformation to G-quartets and thus exhibit strong fluorescence[47–50], we explored whether the dyes could bind to the G-ribbon of 8AG-T and 8OHG-T hydrogels specifically. Interestingly, the 8AG-T and 8OHG-T hydrogels could quench the fluorescence of rhodamine 123 (Rho123), while G-K$^+$ hydrogels hardly quenched the fluorescence of Rho123 (Fig. 7a, b, Supplementary Figs. 43–45), indicating that Rho123 might specifically bind to the G-ribbons of 8AG-T and 8OHG-T hydrogels. Based on the cation-independent feature of 8AG-T and 8OHG-T hydrogels, we investigated whether metal cations could bind to the cation-independent hydrogels specifically to explore potential applications. The responses of 8AG-T and 8OHG-T hydrogels to ions including Li$^+$, Na$^+$, K$^+$, Cs$^+$, Rb$^+$, Ag$^+$, Ca$^{2+}$, Mg$^{2+}$, Ba$^{2+}$, Zn$^{2+}$, Cu$^{2+}$, Cr$^{3+}$, and Al$^{3+}$ were observed. The results show that these ions triggered no change in the 8AG-T hydrogel (Supplementary Fig. 46). The 8OHG-T hydrogel collapsed after 10 min for Ag$^+$, 24 h for Cr$^{3+}$, and 48 h for Al$^{3+}$, while there was no response to other ions (Fig. 7c). The stoichiometric titration experiments were performed to preliminarily investigate whether Ag$^+$ binds with 8OHG-T hydrogel. Supplementary Fig. 47 demonstrate there might be three binding sites of Ag$^+$ and 8OHG-T hydrogel through the weak interaction, and $^1H$ NMR results suggest the Ag$^+$ binding sites of the 8OHG-T hydrogel might be N1H, N7H, and 2-NH$_2$ groups (Fig. 7d, e and Supplementary Fig. 48). The specific mechanisms need further in-depth study.

Ag$^+$ is a wide range of contaminants with serious toxicity and easily accumulated in the human body through the food chain, causing the increases in the risk of neurodegenerative, oncological, and cardiovascular diseases[51]. Cysteine plays an important role in life

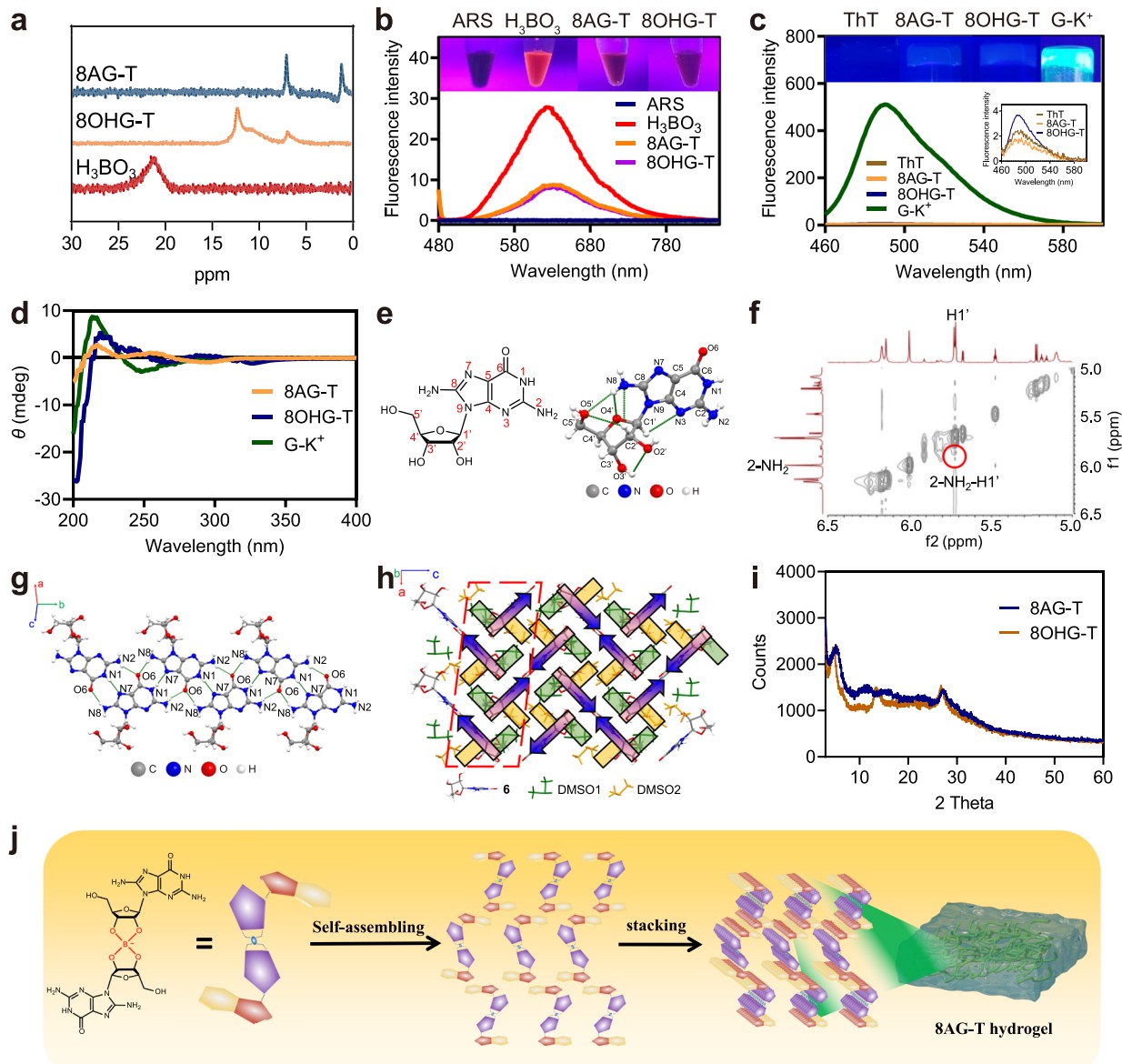

**Fig. 6 | Self-assembly mechanism of the cation-independent hydrogels. a** [11]B nuclear magnetic resonance (NMR) spectra of 8AG-T and 8OHG-T hydrogels. **b** Fluorescence intensity of Alizarin Red S (ARS) in 8AG-T and 8OHG-T hydrogels. **c** Thioflavin T (ThT) assay of 8AG-T and 8OHG-T hydrogels. **d** Circular dichroism spectra of 8AG-T and 8OHG-T hydrogels. **e** The chemical structure and single crystal structure of **6**. **f** [1]H–[1]H nuclear overhauser effect (NOE) of 8AG-T hydrogels. **g** The single crystal structure of the base-pair pattern. **h** The schematic diagram of the single crystal of **6**. The red dashed box includes the interactions between dimethyl sulfoxide (DMSO) and 8AG. **i** The Powder X-ray diffractometry (PXRD) spectrum of 8AG-T and 8OHG-T hydrogels. **j** Schematic illustration of the formation of an 8AG-T hydrogel.

activities, and its concentration fluctuation is closely related to many diseases, such as neurodegenerative diseases[52]. Their traditional detection methods rely on expensive and large-scale instrument as well as specially trained personnel[53,54]. It is important to develop simple and rapid detection methods. As Ag[+] specifically triggered the collapse of the 8OHG-T hydrogel quickly, we attempted to apply it to rapid detection. Figure 7f illustrates the process of the detection of Ag[+] and cysteine based on the 8OHG-T hydrogel. 8OHG-T turned off the fluorescence of Rho123, and the fluorescence was restored by the addition of Ag[+] at a concentration of 10 μM in 10 min. Then, the fluorescence could be quenched again with the addition of 10 μM cysteine in 10 min because cysteine is a strong Ag[+]-binder. These results indicate that the 8OHG-T hydrogel can be used for rapid visual detection of Ag[+] and cysteine. The visual detection method is simple to operate with rapid measurement and eliminates the reliance on

professional personnel and large equipment, thus having potential application in portable detection equipment for Ag[+] or cysteine detection in the future.

For the final part, the self-assembly mechanism of the hydrogels was investigated systematically and show that dynamic borate ester bonds and G-ribbon structure play an important role in the process of hydrogel self-assembly. Furthermore, the cation-independent hydrogel shows potential application in rapid visual detection for Ag[+] or cysteine.

In conclusion, an optimal ML model with 71% accuracy is the most effective prediction model of nucleoside hydrogel-forming ability developed to date. The optimal model was tested with external database, twenty-four nucleoside derivatives were selected for experimental validation, and the accuracy is 83.33% (20/24). In addition, two rarely reported cation-independent hydrogels, 8AG-T and 8OHG-T

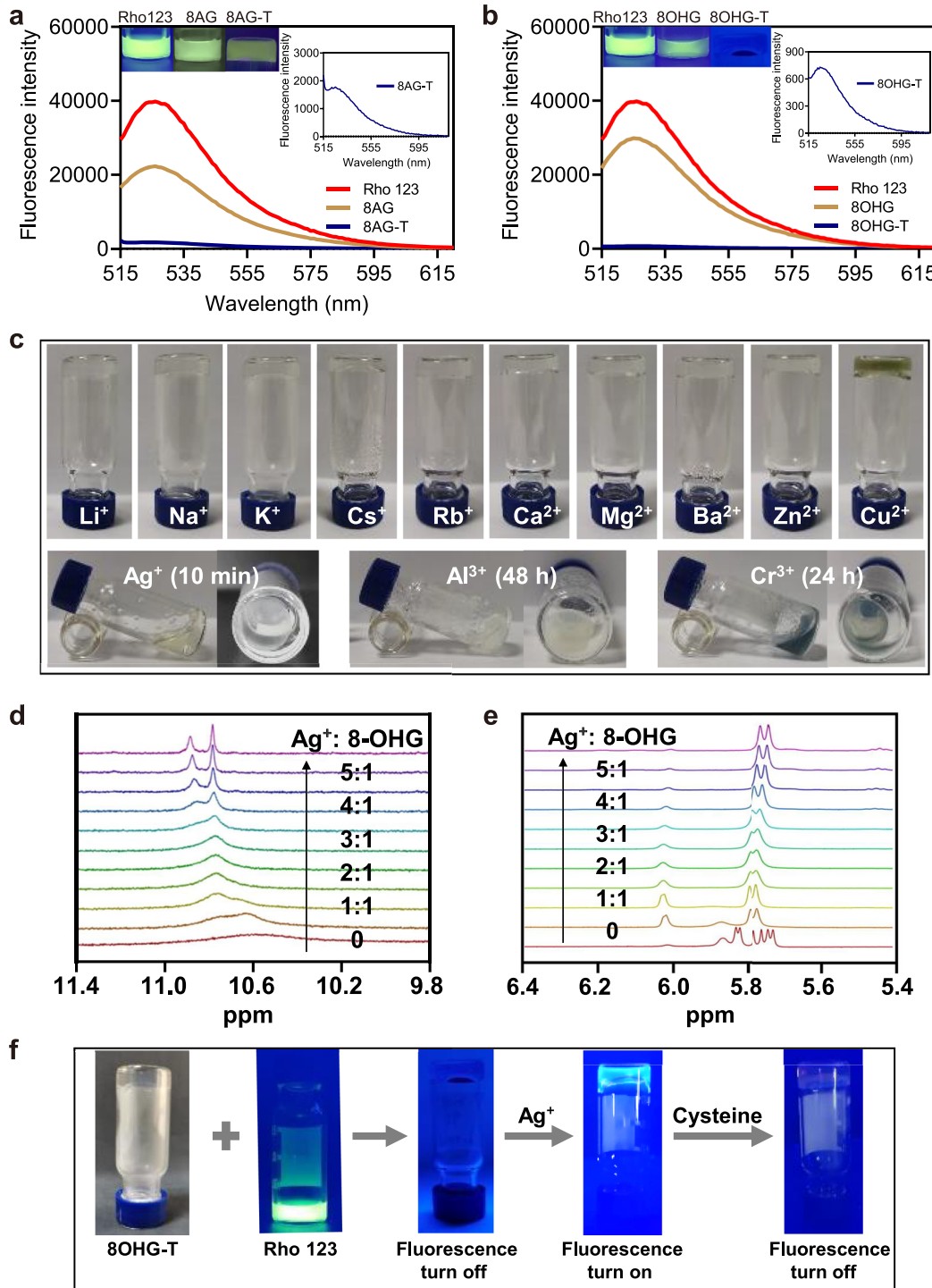

**Fig. 7 | Detection of Ag⁺ and cysteine based on the 8OHG-T hydrogel. a, b** The fluorescence of the 8AG-T (**a**) and 8OHG-T (**b**) hydrogels after adding Rho123. **c** Photographs of the 8OHG-T hydrogels after adding ionic solutions. **d** ¹H nuclear magnetic resonance (NMR) spectrophotometric titration of the 8OHG-T hydrogel with increasing Ag⁺. The peaks represent N1H of **8**. **e** ¹H NMR spectrophotometric titration of the 8OHG-T hydrogel with increasing Ag⁺. The peaks represent N7H of **8**. **f** The process of the detection of Ag⁺ and cysteine based on the 8OHG-T hydrogel.

were found. The investigation of their self-assembly mechanisms shows the dynamic borate ester bonds and G-ribbon may contribute to form hydrogels. Furthermore, the 8OHG-T hydrogel was found to have potential applications in rapid visual detection of Ag⁺ and cysteine. Although the limited number of known nucleoside hydrogels

constrains the accuracy of the prediction model, we believe that increasing the diversity of nucleoside hydrogel information will improve the predictive accuracy of the model. This study represents a step towards using ML as a tool to predict and excavate nucleoside-based hydrogels.

## Methods

### Model construction

**Literature review.** Medical Subject Headings (MeSH) were used to collect all subjects and free terms for nucleosides (uridine, thymidine, adenosine, guanosine, and cytidine) and their derivatives, and a system-wide search was conducted on Medline, Web of Science, and SciFinder for all nucleoside-related studies. The specific information on the search strategy is shown in Supplementary Data 9.

The initial search identified 882 articles, and after excluding duplicate studies and searching titles and abstracts, according to the inclusion criteria, 79 articles remained. After further reading of the full text and excluding the duplicate structures, 18 articles with 71 molecules were included.

The inclusion criteria for the publications were as follows:
(i) A clear definition of gelator/non-gelator and the analysis results to determine gelation, such as the detailed results of tube-inversion tests or rheological tests were provided.
(ii) The exact chemical structure was provided in the study.
(iii) The solvent was pure water or an aqueous solution.
(iv) Nucleosides and their derivatives were included; however, base derivatives or nucleotide derivatives were excluded.

Finally, a dataset was constructed based on the 71 nucleoside derivatives collected. This dataset including the structures of nucleoside derivatives and whether they have the hydrogel-forming ability was used to build the prediction models in the subsequent work.

**Definition of gelators.** Groupings were definition based on gelator/non-gelator, as clearly defined in the study. In the study, some substances used as a reference and a control are often not explicitly specified as to whether they are a gelator or not. If the analytical results such as tube-inversion tests or rheological measurements were provided, then the groupings were determined based on the analytical results. We considered nucleoside derivatives to be hydrogel-forming ability if they formed gels under arbitrary conditions in pure water or aqueous solutions; in addition, results containing organic solvents were excluded. Since gelator molecules tend to be the focus of studies, with few cases of non-gelators reported, to facilitate training accuracy, we combined structures in non-gelator cases such as solution and precipitation into the non-gelator group. Then the grouping was determined according to the test results.

**Forming the dataset.** The molecular structures of the 71 nucleoside derivatives were redrawn by ChemDraw (version 20.0) to let them present and characterized uniformly, and the nucleoside derivative structures were converted to SMILES (Simplified Molecular Input Line Entry System) strings using ChemDraw[33,55]. Then, molecular descriptors of the nucleoside derivatives were calculated.

A total of 5666 molecular descriptors were calculated for 71 nucleoside derivatives using the Python (Version 3.9.12) package alvaDescCLIWrapper (Version 1.1.1) in this study, the alvaDescCLIWrapper package was used to access the functionalities of alvaDesc software[56] (Version 2.0.12). And 4175 molecular descriptors were kept for subsequent analysis by removing 1491 descriptors with missing values.

Taken together, the dataset included the molecular structures, SMILES, information about whether form hydrogel, and molecular descriptors of 71 nucleoside derivatives.

**Machine learning models.** LR[37], RF[38], XGBoost[39], and DT[40] were implemented using Scikit-learn[57] (Version 1.1.1) by Python. Hyperparametric optimization was performed using the training dataset on a high-performance computing server. Due to the limited size of the training dataset ($n = 71$), we used a fivefold stratified cross-validation to estimate the 95% confidence intervals of the five evaluation indexes.

Divided the data into five equal parts by maintaining the ratio of gelators to non-gelators and took four of them as the training set to train the model and the other one as the test set, training the model 5 times without repetition[58]. On top of this, 10 times of random fivefold stratified cross-validation was performed to estimate parameters[59,60]. The model was first trained using the training set and then evaluated using the test set to determine its ability to predict unseen data. Thus, the training and test scores for each model are the average of 10 crossovers. The probability threshold was 50% was used for the training and testing of the ML models. For supervised ML, the training process aims to fit and modify the models by parameter tuning to minimize the cost function (the difference between the predicted and actual values).

**Sensitivity analysis.** A sensitivity analysis was performed by dividing the data set into training set and test set[61]. We use 5-fold cross validation to train and hyperparameter the model on the training set and evaluate the model's generalization ability on an additional test set. To ensure that the test set covers the same area as the training set, we stratified sampling based on the results of clustering and hydrogel-forming ability, dividing nucleoside derivatives into 80% as training set ($n = 56$) and 20% as test set ($n = 15$).

**Feature selection of molecular descriptors.** We performed a three-step feature selection for molecular descriptors[59,60].

Firstly, the rank-sum test was used as univariate difference analysis. The results demonstrated that there are significant differences ($P < 0.05$) between the gelator ($n = 38$) and non-gelator ($n = 33$) group, which means it may have potential association with the hydrogel-forming ability. In this study, 144 descriptors were obtained by removing the 4031 of 4175 descriptors which have no significant association with hydrogel-forming ability. In this study, $P$-value $< 0.05$ was considered significant.

Secondly, the Spearman correlation coefficient was calculated between the 144 descriptors in pairs. Then, the pair of descriptors whose correlation coefficient is higher than 0.8 (Rho > 0.8) was selected, and one of the pair was excluded to avoid collinearity. After this step, 40 descriptors were kept for subsequent model training.

Finally, we used ML algorithm based RFE to obtain the optimal combination of descriptors and maximize model performance. Four ML algorithms mentioned above were utilized in this step. For the four ML algorithms, the training process aims to fit and modify the models by parameter tuning to minimize the cost function. For the model with molecular descriptors, we used 10 times fivefold stratified cross-validation to perform the RFE of the model.

In addition, we also used Least Absolute Shrinkage and Selection Operator (LASSO)[62] and Multiple linear regression with expectation maximization (MLREM)[61,63] for feature selection as the comparison to test the effectiveness of the feature selection methods. LASSO and MLREM as the feature selection methods did not yield better results than above three-step feature selection (Supplementary Table 8).

**Feature importance of molecular descriptors.** The regression coefficients of 4175 molecular descriptors in the model were calculated using LASSO regression[62], and the features whose regression coefficients were greater than the Scikit-learn recommended threshold 1e-5 were screened out. Finally, the feature importance of 70 molecular descriptors was obtained. The regression coefficients of the 24 molecular descriptors are the feature importance of the optimal model. And the PFI method was used to train the model for 1000 times, and the Mean and SEM for mean accuracy decrease was obtained when a single feature value was randomly excluded.

**Hyperparameter optimization.** Bayesian optimization works by constructing a posterior distribution of functions (Gaussian process) that

best describes the function to be optimized[64]. The hyperparameters of all models were optimized using Bayesian optimization by Python package Optuna[64] (Version 3.1.0), and each step was conducted 10 times with fivefold stratified cross-validation. Using bayesian optimization, we can explore the parameter space more smartly and thus reduce the time required to perform this process. All data and code are included in the GitHub repository linked to the work.

**The specific selection of nucleoside derivative structures from PubChem.** The nucleoside derivative structures were selected from PubChem[65] (https://pubchem.ncbi.nlm.nih.gov). Firstly, we searched five basic nucleosides (uridine, thymidine, adenosine, guanosine, and cytidine) from PubChem. Then obtained derivatives for the five nucleosides based on 3D similarity by using PubChem3D and chose chemical vendors available[66]. We used shape-Tanimoto (ST) and color-Tanimoto (CT) as indexes for 3D similarity. Only nucleoside derivatives with 3D similarity for the five basic nucleosides of ST $\geq$ 0.80 and CT $\geq$ 0.50 were included. A total of 11,406 structures were initially incorporated (uridine, $n$ = 1023; thymidine, $n$ = 3269; adenosine, $n$ = 2760; guanosine, $n$ = 1840; cytidine, $n$ = 2514). Finally, 7257 structures remained after excluding duplicate structures.

## Experimental verification

**General experimental.** All the chemicals were commercially available and of analytical grade. 1-[3,4-dihydroxy-5- (hydroxymethyl) oxolan-2-yl]-1,3,5-triazinane-2,4,6-trione (**1**; DTT; 95%) was purchased from Hitgen (Chengdu, China). Xanthosine (**2**; XTS; 95%) was obtained from Alfa Aesar (Shanghai, China). Guanine 5′-monophosphate (**3**; GMP; 98%), inosine 5′-monophosphate (**4**; IMP; 98%), 5-fluorouridine (**5**; 5-FUR; 98%), and gemcitabine (**20**; GCTB; 98%) were obtained from TCI Shanghai (Shanghai, China). 8-aminoguanosine (**6**; 8-AG; 98%) and 2′-deoxyguanosine 5′-monophosphate (**7**; dGMP; 98%) were obtained from Yuanye (Shanghai, China). 8-hydroxyguanosine (**8**; 8-OHG; 97%), 2′-O-methylguanosine (**12**; 2′-OMe-dG; 98%), 5,6-dichlorobenzimidazole riboside (**13**; DRB; 99.73%), 9-(2-tetrahydropyranyl) adenine (**14**; 9-THPA; 97%), 9-(2-tetrahydrofuryl)adenine (**15**; 9-THFA; 99.74%), 2′,5′-dideoxyadenosine (**18**; 2′,5′-DA; 99.68%), 2-chloro-9-(2-tetrahydropyranyl)adenine (**22**; 2-Cl-9-THPA; 95%), 7-deaza-2′-C-methyladenosine (**23**; 7-D-2′-MeA; 95%), 2′-C-methylcytidine (**24**; 2′-MeC; 99.69%), and inosine (**10a**; 98%) were purchased from Bidepharm (Shanghai, China). 8-azaguanosine (**9**; 8-azaG; 95%) was purchased from Chempartner (Shanghai, China). Inosine-5′-carboxylic acid (**10**; I-5′-CA; 95%) was synthesized out according to the reference[67], and the synthesis methods in detail were shown later. 2′-amino-2′-deoxyguanosine (**11**; 2′-NH₂-dG; 98%) was obtained from Fluorochem (Derbyshire, England). 2-thiocytidine (**16**; 2-TC; 95%) was purchased from Target Molecule (Boston, USA). 2′,3′-dideoxy-2′,3′-didehydroadenosine (**17**; 2′,3′-DA; 97%) and 2-chloro-2′,3′-O-isopropylideneadenosine-5′-N-ethylcarboxamide (**21**; 2-ClA; 95%) were obtained from Jiangsu Aikon (Nanjing, China). 2′-C-methyladenosine (**19**; 2′-MeA; 99.63%) was obtained from Haohong (Shanghai, China). Deionized water was purified using the Milli-Q Plus System. Guanosine (G) was purchased from Sigma–Aldrich (St Louis, USA). Deionized water was purified using the Milli-Q Plus System. All reagents and materials were used as received unless otherwise noted.

**Synthesis of Inosine 5′- carboxylic acid.** The synthesis route refers to Middleton RJ, et al. (Supplementary Fig. 49)[67].

**Synthesis of 2′,3′-O-Isopropylideneinosine (10b).** To a suspension of inosine (**10a**; 5 g, 18.64 mmol, 1 eq.) in 100 mL of acetone was added perchloric acid (2.24 mL, 37.28 mmol, 2 eq.), and the reaction mixture was stirred at room temperature for 24 h (TLC monitoring). Add saturated NaHCO₃ (100 mL) and stir the solution for another 2 h. Filter the suspension and wash the product with cold water (3 ×150 mL). The

reaction mixture was filtered, and the white solid was dried over vacuum. Yield: 4.77 g (83.0%).¹H NMR (400 MHz, DMSO-$d_6$) $\delta$ 12.45 (s, 1H, NH), 8.31 (s, 1H, C2-H), 8.10 (s, 1H, C8-H), 6.10 (d, $J$ = 2.9 Hz, 1H, C1′-H), 5.26 (dd, $J$ = 6.2, 2.9 Hz, 1H, C2′-H), 5.14 (t, $J$ = 5.3 Hz, 1H, C5′-OH), 4.93 (dd, $J$ = 6.2, 2.5 Hz, 1H, C3′-H), 4.22 (td, $J$ = 4.8, 2.6 Hz, 1H, C4′-H), 3.54 (t, $J$ = 5.0 Hz, 2H, C5′-H), 1.52 (s, 3H, CH₃), 1.31 (s, 3H, CH₃). ¹³C NMR (101 MHz, DMSO-$d_6$) $\delta$ 156.59, 147.83, 146.11, 138.81, 124.49, 113.14, 89.65, 86.69, 83.86, 81.30, 61.49, 27.05, 25.19. HRMS (ESITOF) m/z: calcd for $C_{13}H_{16}N_4O_5$ [M + H⁺] 309.1199, found 309.1193 (Supplementary Figs. 50–52).

**Synthesis of 2′,3′-O-Isopropylideneinosine 5′-carboxylic acid (10c).** To a suspension of 2′,3′-O-Isopropylideneinosine (2 g, 6.49 mmol, 1 eq.), 2,2,6,6-tetramethyl-1-piperidinyloxy free radical (TEMPO) (508 mg, 3.25 mmol, 0.5 eq.), iodobenzene diacetate (BAIB) (4.8 g, 14.93 mmol, 2.3 eq.) in acetonitrile/water (1:1, v/v, 30 mL), The reaction mixture was stirred at room temperature for 6 h. Filter and wash the residue with ether and acetone. Dry under vacuum to obtain white solid product. Yield: 1.50 g (72.0%). ¹H NMR (400 MHz, DMSO-$d_6$) $\delta$ 12.39 (s, 1H, NH), 8.21 (s, 1H, C2-H), 8.01 (s, 1H, C8-H), 6.32 (s, 1H, C1′-H), 5.50 – 5.44 (m, 1H, C2′-H), 5.42 (d, $J$ = 6.0 Hz, 1H, C3′-H), 4.71 (d, $J$ = 1.7 Hz, 1H, C4′-H), 3.54 (d, $J$ = 4.6 Hz, 1H, C5′-H), 1.50 (s, 3H, CH₃), 1.34 (s, 3H, CH₃).¹³C NMR (101 MHz, DMSO-$d_6$) $\delta$ 170.80, 156.60, 148.04, 145.62, 139.85, 124.32, 112.75, 89.73, 85.46, 83.76, 83.59, 26.44, 24.88. HRMS(ESITOF) m/z: calcd for $C_{13}H_{14}N_4O_6$ [M + H⁺] 323.0992, found 323.0987 (Supplementary Figs. 53–55).

**Synthesis of inosine-5′- carboxylic acid (10).** To a suspension of 2′,3′-O-Isopropylideneinosine 5′-carboxylic acid (1 g, 3.10 mmol, 1 eq.) in 1 N HCl (20 mL) under argon atmosphere, the reaction mixture was stirred at room temperature for 2 h. The solution was cooled and then dripped into acetone to precipitate. The solid was filtered and dried by freeze-drying machine to yield a white powder. Yield: 746 mg (85.2 %). ¹H NMR (400 MHz, DMSO-$d_6$) $\delta$ 12.46 (s, 1H, NH), 8.44 (s, 1H, C8-H), 8.10 (s, 1H, C2-H), 6.02 (d, $J$ = 6.4 Hz, 1H, C1′-H), 4.48 (dd, $J$ = 6.4, 4.5 Hz, C2′-H), 4.42 (d, $J$ = 2.5 Hz, 1H, C4′-H), 4.30 (dd, $J$ = 4.4, 2.6 Hz, 1H, C3′-H). ¹³C NMR (101 MHz, DMSO-$d_6$) $\delta$ 172.06, 156.56, 148.70, 146.28, 138.39, 123.96, 87.09, 82.70, 74.41, 73.37. HRMS (ESITOF) m/z: calcd for $C_{10}H_{10}N_4O_6$ [M + H⁺] 283.0679, found 283.0674 (Supplementary Figs. 56–58).

**Procedure for hydrogel preparation.** The nucleoside derivative was precisely weighed and dissolved in 0.4 mL solvent and heated to a transparent liquid. The concentrations of the nucleoside derivatives and solvents are as follows: The concentrations of G, **6**, and **8** were 100 mmol. The concentrations of **1 - 5**, **7**, **9 - 24** were 50 mmol. The concentrations of KCl, NaCl, and AgNO₃ were 0.2 M. The quantity addition of H₃BO₃, Tris, KB(OH)₄, and NaB(OH)₄ were the half of the equimolar quantities of nucleoside derivatives. Then, the sample was cooled to room temperature to form the hydrogels and subjected to a tube-inversion test. Hydrogel formation was confirmed if no sample flow was observed upon inversion of the tube. 8AG-T, 8OHG-T, and G-T hydrogels were respectively prepared by **6**, **8** and G in Tris and H₃BO₃ solutions. 8AG/8OHG/G-Na⁺ hydrogels and 8AG/8OHG/G-K⁺ hydrogels were respectively prepared by **6, 8**, and G in NaB(OH)₄ or KB(OH)₄ solutions, respectively.

**Rheology measurements.** Rheology measurements were performed using an MCR302 rheometer (Anton Paar, Graz, Austria) equipped with plate-plate geometry (25 mm in diameter, 1 mm gap). The hydrogel was heated until it became liquid and was immediately transferred to a plate to be measured. Silicone oil was added along the plate to reduce the evaporation of water during testing. The plate was preheated to 75 °C in case the samples gelled before rheology measurements. A time sweep test was performed at a temperature from 75 °C to 25 °C, an

angular frequency ($\omega$) of 10 rad s$^{-1}$, and a strain ($\gamma$) of 0.1%. A frequency sweep test was performed with $\omega$ from 100 to 0.1 rad s$^{-1}$ at $\gamma = 0.1\%$. A strain-dependent test was performed under $\gamma$ from 0.1% to 150% at $\omega = 10$ rad s$^{-1}$. A shear strain test was conducted to record gel viscosity at $\gamma$ from 0.1% to 100%. Tests with $\gamma$ values of 0.1% and 100% were conducted to determine the self-healing ability of the samples.

**Scanning electron microscopy (SEM).** The hydrogel samples were prepared and then freeze-dried to form xerogels. The xerogel was attached to the silica wafer and coated with gold. SEM measurements were performed using an Inspect F50 scanning electron microscope (FEI, USA).

**Atomic force microscopy (AFM).** The hydrogel samples were prepared and then diluted with water. 5 μL of diluted samples were dropped onto mica plates and dried in air at room temperature. AFM images were obtained using an SPM-9700 scanning probe microscope (Shimadzu, Japan).

**Variable-temperature small-angle X-ray scattering (VT-SAXS).** The hydrogel samples were placed in a Hilgenberg quartz capillary with an outside diameter of 2 mm and a wall thickness of 0.01 mm. The exposure time of samples was 30 min. VT-SAXS measurement was performed by a NanoStar instrument (Bruker, Germany) with Cu target (50 Kv/50 mA). The detector used was VÅNTEC-2000 2D. The systems used were Montel-P multilayer optics with a pinhole collimation system and low background collimation system with SCATEX two-pinhole setup.

## Mechanism and application

**Nuclear magnetic resonance (NMR).** The hydrogel samples were prepared and then freeze-dried to xerogel. 8OHG-T/Na$^+$/K$^+$ and G-T/Na$^+$/K$^+$ xerogels was dissolved in D$_2$O. 8AG-T/Na$^+$/K$^+$ and G-T/Na$^+$/K$^+$ xerogels were dissolved in DMSO-$d_6$. The concentrations of samples for $^{11}$B NMR and $^1$H–$^1$H Nuclear Overhauser Effect (NOE) experiments were 40 mg mL$^{-1}$. 0.5 mL samples were transferred into NMR tubes. The $^{11}$B NMR spectra were obtained using an AV II spectrometer (Bruker, Germany) at 600 MHz. NOE experiments were performed on an Avance III-800 MHz spectrometer with a Quadruple Cryo Inverse probe at 25 °C (Bruker company).

**Alizarin Red S experiment.** The hydrogel samples were prepared and mixed with Alizarin Red S (ARS) solution. Samples were detected by a Cary Eclipse fluorescence spectrophotometer (Agilent, USA) with excitation/emission wavelengths at 500/530 nm.

**Thioflavin T experiment.** 200 μL hydrogel was heated to a transparent liquid and mixed with 2 μL Thioflavin T (ThT) solution (1 mmol). Then, the sample was cooled to room temperature to form hydrogels. The fluorescence of the sample was observed at 365 nm by a dark-box ultraviolet analyzer (JY02S, Beijing Junyi, China), and detected by a Cary Eclipse fluorescence spectrophotometer (Agilent, USA) with excitation/emission wavelengths at 450/485 nm.

**Circular dichroism (CD) Spectra.** The hydrogel samples were prepared and diluted to a certain ratio according to the limitation of the detection range of the instrument. The CD spectra between 200 and 400 nm were recorded by a CD spectrometer (J-810, JASCO, Japan) at 25 °C. The CD spectra were obtained by the average of six consecutive scans. The scan rate was 100 nm min$^{-1}$, and the scan bandwidth was 2.0 nm.

**Ultraviolet (UV) Spectra.** The hydrogel samples were prepared and diluted to a certain ratio according to the limitation of the detection range of the instrument. The UV spectra between 200 and 400 nm were recorded by a Cary Series UV-Vis spectrophotometer (Agilent, USA).

**Single Crystal X-ray Diffraction.** The results of X-ray diffractions for single crystal of **6** were obtained at 100(2) K on a D8 VENTURE diffractometer (Bruker, Germany) using Mo-Kα radiation ($\lambda = 0.71073$ Å). SAINT program was used to obtain the integration and scaling of intensity data. SADABS was used to correct the data for the effects of absorption. The structures of single crystal were solved by direct method, and they were carried out by a full-matrix least-squares method using SHELX-2014 software. All the non-hydrogen atoms in single crystal were refined with anisotropic displacement parameters. All the hydrogen atoms in single crystal were placed in calculated positions and refined with a standard riding model. The results of the Hirshfeld surface analysis and intermolecular interaction energy calculations for the single crystal were performed by CrystalExplorer software[68–70].

**Theoretical calculation.** Density-functional theory (DFT) calculations were conducted using the Gaussian 09 program[71]. The B3LYP[72,73]/6-31 G(d)[73] computational models were used to optimize structures. Then, single point energy calculations were carried out at the B3LYP-D3 /6-311++G(d, p) theoretical level for optimized structures[74]. In addition, the Solvation Model Based on Density (SMD)[74] was used to reduce the influence of solvent on the energy when calculating the single point energy. The atomic coordinates of the optimized computational models were provided in Supplementary Data 10.

**Powder X-ray diffractometry (PXRD).** The hydrogel samples were prepared and then freeze-dried to xerogel. Xerogel powder was analyzed by a PXRD diffractometer (X'Pert Pro MPD, Netherlands). The voltage was 40 kV. The current was 40 mA. The PXRD pattern of samples were recorded from 3° to 60° (2$\theta$).

**The fluorescence detection of hydrogels with dyes.** 200 μL hydrogel was heated to a transparent liquid and mixed with 2 μL dye solution (1 mmol). The dyes were rhodamine 123 (Rho123), rhodamine B, rhodamine 6 G, fluorescein, safranin O, fluorescein, thiazole orange, crystal violet, rosolic acid, basic fuchsin, thioflavin T, methylene blue, and rose bengal. Then, the sample was cooled to room temperature to form hydrogels. The fluorescence of the sample was observed at 365 nm by a dark-box ultraviolet analyzer (JY02S, Beijing Junyi, China). Hydrogels with Rho123 were detected by a Cary Eclipse fluorescence spectrophotometer (Agilent, USA) with excitation/emission wavelengths at 507/529 nm.

**The responses of hydrogels to ions.** 200 μL hydrogel was prepared in a vial. 50 μL solutions of metal ion (50 mmol) were added to the hydrogel. The ion solutions were LiCl, NaCl, KCl, CsCl, RbCl, AgNO$_3$, CaCl$_2$, MgSO$_4$, BaCl$_2$, ZnSO$_4$, CuSO$_4$, Cr(NO$_3$)$_3$, and AlCl$_3$. The responses of hydrogels to metal ions were observed for one week.

## Data availability

All relevant data supporting the key findings of this study are available. The existing datasets analyzed as well as datasets generated during the study also have been made available in GitHub (https://github.com/leescu/NHGPM). The chemical information of 71 nucleoside derivatives is available at https://www.nhgpm.com. The X-ray crystallographic coordinates for structures reported in this study have been deposited at the Cambridge Crystallographic Data Centre (CCDC), under deposition numbers 2253566. These data can be obtained free of charge from The Cambridge Crystallographic Data Centre via www.ccdc.cam.ac.uk/data_request/cif. Source data to re-create figures has been deposited on zenodo: https://doi.org/10.5281/zenodo.10723552.

## Code availability

All codes used in this study are available git repositories https://github.com/leescu/NHGPM (https://doi.org/10.5281/zenodo.10723747)[75].

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

## Acknowledgements

This work was supported by the National Natural Science Foundations of China (No. 82271035 to H.Z.), the National Key R&D Program of China (2022YFC2402900 to H.Z.), the funding from the Sichuan Provincial Department of Science and Technology (2021-YJ-0021 to H. Z.), the CAMS Innovation Fund for Medical Sciences (CIFMS,2019-I2M-5-004 to H.Z.). The authors thank Zheng Wang, Xin Xia, Liying Hao, Ting Li, Bomiao Cui and Qiang Wei (State Key Laboratory of Oral Diseases, West China Hospital of Stomatology, Sichuan University) for helping us perform AFM, NMR, SEM, and single crystal studies. The authors also thank Yani Xie and Xiaoyan Wang (the Analysis and Testing Center of Sichuan University) for helping us perform CD and NMR studies. The authors also thank Siyuan Hao for helping data collection.

## Author contributions

Q.C., H.X. and H.Z. conceived and directed the project; W.L. and Y.W. wrote the paper; Y. W., K.W. and Z.D. performed most experiments under the supervision of H.X., H.Z. and L.X.; H.X., W.L. and L. W. constructed the models; W.L. and Y.W. performed external validation of the model; L.X. and Y.W. explored the mechanism and application of the cation-independent hydrogel. All authors discussed the results and commented on the manuscript. W.L., and Y.W., and K.W. contributed equally to this work.

## Competing interests

The authors declare no competing interests.
