## [Peer Review File · Nature Communications]

Developing a machine learning model for accurate nucleoside hydrogels prediction based on descriptorsEditorial Note: Parts of this Peer Review File have been redacted as indicated to remove third-party material where no permission to publish could be obtained.

REVIEWER COMMENTS

Reviewer #1 (Remarks to the Author):

The contributions summarize the creation of gelation machine learning (ML) models for a 71 nucleoside gelators from the literature. The aim was to predict gel/no gel behaviours of the materials. The distribution of gel-forming versus non gel-forming was good (38 vs 33)

The topic is of considerable interest as relatively few models claim to predict the gelation behaviour of molecules.

However, there are considerable methodological issues with the models, their interpretation, and their use to find new gelators. My review is largely concerned with the ML models as I have limited expertise in the physical analysis of gels.

The manuscript is not written particularly clearly, and grammar is an issue. It is difficult to understand the sequence of events, going from thousands of descriptors down to a much smaller number. It is not clear from the text and figures whether the 4175 descriptors were calculated by a package, then the 4 types of fingerprint descriptors added, or whether the fingerprints were part of the 4175.

Additionally, the authors state that because the number of data points available for the models was small (71) they 'copied' the data set to generate 5 copies, 4 of which were used to train the model and one was the 'test set'. If this is true then such an approach is invalid, there is no independent test set to assess model predictivity for new data. The model is only predicting its training data. If overfitting occurs, then the model could look very good but be incapable of generalizing to new data. Perhaps the authors meant that the 71 compounds were split into 80% (57) for the training set and 20% (14) for the test set. This must be clarified.

Inspection of Table S1 suggests that almost all models have similar predictivity when the uncertainties in their accuracies, expressed by multiple metrics) are compared. There is no obvious reason why LR should be 'much' better than the other nonlinear ML methods. XGBoost routinely provides a superior model prediction (often by a small margin)

The descriptors chosen are quite arcane, making interpretation difficult. It is not clear why modern sparse feature selection method like LASSO or MLREM was not used to reduce the dimensionality of the descriptor space. It is not clear how the ranked sum test was used to reduce the number of descriptors. In Figure S5 it is not clear why multiple versions of logP and PSA were used, surely these would have been highly intercorrelated and removed by the correlation filter?

Why does Table S1 have two entries for each of the 4 ML methods with 144 descriptors ('descriptors' is misspelt)

Even assuming the ML model was valid, using it to screen 7257 new nucleoside analogues is very unwise unless these new materials fall within or near the domain of applicability of the model. There is no discussion of what the domain is and whether the new nucleoside analogues are within it. It seems very unlikely that a model with limited accuracy could reliably predict the properties of >7000 new materials when trained on only 71 (or 57) nucleosides.

It was not explained why visual detection of Ag⁺ or cysteine was important

Care should be taken with copy editing e.g., careless use of RFE vs REF, glue-forming vs gel-forming etc.

Reviewer #2 (Remarks to the Author):

COMMENTS

Weiqi Li et al. present a sort of machine learning model for hydrogels prediction. This an interesting MS with important concepts in the field of hydrogels.

However, despite interesting elements this MS fails the technical screening so this version should be rejected.

Comments are the following:

-Figures quality is insufficient

- Page 6: "The 3D PCA with 40 descriptors can better distinguish the two groups (Fig. 1e)." and legend to Figure 1e: "PCA results for 40 descriptors, and the two groups tend to separate."

By looking at Figure 1, it is not obvious that the gelator/non-gelator groups are better separated in plot 1e than in plot 1b.

- Page 8: "LR based on 24 descriptors after RFE (red color) considered to be the optimal model (test accuracy, 0.71)"

What probability threshold was used for the training and testing of the ML models?

As discussed later on page 10 and shown in Figure 3b, the ML model predicts a compound as gelator or non-gelator with a given probability. When the models make good or bad predictions, it would be interesting to know the associated probabilities.

- Page 8: "Beside the 5 descriptors mentioned above, some other descriptors are likely to be related to hydrogel-forming ability. Including hydrophilic factor (Hy) 63 , topological polar surface area (TPSA), octanol-water partition coefficient (LogP), and solubility (ESOL), while no significant difference was found (Fig. 3a and S5)." and legend to Figure 3a: "grouped box plot of 4 descriptors with potential relevance to glue-forming ability for 71 nucleoside derivatives"

Why would these 4 compounds be relevant since they cannot distinguish gelator/non-gelator groups? I don't see the pertinence of this.

- Page 10: "Based on previous experience and knowledge of the hydrogel-forming ability of nucleoside derivatives, we selected three guanosine derivatives for validation among the top 10% ranked structures." and "The specific structures and gelator probabilities of the three nucleoside derivatives are in the forefront (Fig. 3c-e, probabilities of gelator: 1, 0.590; 2, 0.590; 3, 0.587)."

In Figure 3b the vast majority of the compounds have a gelator probability between 0.4 and 0.6. That does not seem to be very strong.

Also, the authors selected 3 compounds out of the top 10%, however, 10% of 7257 compounds tested is 726, which is quite a large subset. What about the compounds that are at the very top of the distribution? Selection was made based on "previous experience and knowledge": how were these 3 compounds precisely selected?

Also in Figure 3b, I assume that the blue dots (labelled as "alternative") correspond to the 3 selected molecules? Note also that they are just above the 10% cut-off and not at the very top of distribution.

-p10 and Fig 3. Why lifetime hydrogels were tested in H₃BO₃ with Tris solution for 8-AG and 8-OHG or AgNO₃ solution for 8-AzaG

-p10 and Fig 3. Why H₃BO₃ with Tris solution and AgNO₃ solution was used as the solvent to form hydrogel?

-p11 "This is the first-time cation-independent guanosine-derived hydrogels have been discovered." Not correct see for example: i) ACS Omega 2018, 3, 2, 2230–2241,

<https://doi.org/10.1021/acsomega.7b02039>, ii) soft mater DOI <https://doi.org/10.1039/C8SM00299A>, iii) Organic & Biomolecular Chemistry

2018, DOI: 10.1039/C8OB01023D, iv) Coordination Chemistry Reviews

Volume 488, 1 August 2023, 215170 <https://doi.org/10.1016/j.ccr.2023.215170>, etc

-p11 Results very similar to Org. Biomol. Chem (DOI: 10.1039/C8OB01023D)

-Fig. 4c. G' is not \gg G''

-Fig. 4a. caption no mention of terms "Pre" or "Sol" in the Fig. 4a.

-Fig. 4d, e caption seems do not match the figure ?

-Fig. 4j. Figure is missing

-p15 "borate diester" was not introduced before. Give context.

-Fig 5c inset illegible

-Fig 6d. Completely wrong. Abs at 3 = no photon. Inset fit is inappropriately drawn and (voluntary) misleading

- Pages 23 and 24 (Mat & Meth): "Due to the limited size of the training dataset (n=71), we divided all the data into five copies. One copy of the data was selected as the test set, and the remaining four copies were selected as the training set. A 5-fold cross-validation was randomly performed 20 times to improve the accuracy of the fitting results"

I guess the authors mean "subsets" instead of "copies"? For cross-validation the training and test sets must be independent. Using multiple copies of the same data and using some of them as training and the others as tests would be an invalid procedure because the same data would be used in both sets. Also, cross-validation is not done to improve accuracy, but to calculate or estimate it.

Note that these sentences (and the next 3, until "... (the difference between the predicted and actual values)" are exactly repeated on pages 23 and 24.

Other minor details:

_Fix the language. There are sentences with unclear grammar at several places in the manuscript. For ex.:

Page 5: ... and four kinds of molecular fingerprints were used to present above nucleoside derivatives derived by SMILES

Page 8: _Compared to the new important features concluded by ML model, the descriptors summarized

by existing experience are not better predictors for the hydrogel-forming ability of nucleoside derivatives.

Page 4: "Firstly, a dataset of 71 nucleoside derivatives with hydrogel-forming ability": rephrase because some are gelators and some are not (page 5: "These nucleoside derivatives were then divided into two groups, namely, gelator (n=38) and nongelator (n=33)").

Page 5: "Pearson correlation coefficients ($r < 0.8$)", whereas on page 24 (Mat & Meth) it says "We performed a Spearman's rank correlation test on the remaining descriptors to exclude one of the pairs of descriptors with correlations higher than 0.8"

"Descriptors" instead of "Descriptores" on Figures 1 and 2

"descriptor-last" on Figure 2a: I guess it refers to the final descriptor with 24 parameters. Maybe write "descriptor-24"? Also, "descriptor-40" not shown.

Figure 2b: the "Recall" and "Precision" metrics are not mentioned anywhere else in the manuscript. Maybe useful to remind what Recall/Precision/F1 score/AUC represent for non-ML expert readers?

Figure 4i: what are PDDF profiles?

Page 23: the reference to Scikit-learn is missing:

Scikit-learn: Machine Learning in Python, Pedregosa et al., JMLR 12, pp. 2825-2830, 2011.

Page 24: "We first filtered descriptors with a Wilcoxon rank-sum test less than 0.05": rank-sum test p-value less than 0.05

Page 25: "Specific selection of nucleoside derivative structures" should be a title in bold. "The nucleoside derivative structures were selected from PubChem ()": missing reference.

"Data availability" section: "Source data are provided with this paper and

<https://github.com/leescu/NHGPM>": remove as it is redundant with the previous sentence.

Page 31, Ref 30: Huang, J. et al. Identification of potent antimicrobial peptides via a machine-learning pipeline that mines the entire space of peptide sequences. Nature Biomedical Engineering, n/a, NA, (2023). Maybe include the doi: 10.1038/s41551-022-00991-2

Reviewer #3 (Remarks to the Author):

This manuscript may be of interest to Nature Comm. Potentially the results are quite good but in the current incarnation this manuscript does not meet the standards required for publication.

After the introduction, which is quite well-written, the results and discussion is littered with abbreviations and acronyms that do not make much sense to the reader... For example, the opening sentences of this section read: "To construct the prediction model, all the published nucleoside derivatives and their hydrogel-forming ability were collected by literature review, and 71 molecules were included 5,7,40-55. To unify the molecular structure, the Chemdraw software (Version 20.0) was utilized to reproduce the 71 molecular structures."

What does this mean? How many is "all published"? What does it mean to collect the hydrogel forming ability by literature review? Does this mean that every published nucleoside derivative has a documented ability to form, or not, a hydrogel? What does it mean that 71 molecules were included? How does one unify the molecular structure? And what does ChemDraw have to do with reproducing the molecular structures?

I could go on, but these two sentences are sufficient to illustrate the point that the results, methods and discussion of this work is not sufficiently clearly articulated to allow a realistic judgement on the quality of the work.

There are other significant issues with the manuscript that raises some real concerns for this reviewer:

1. There are a lot of references but a number of these are not really related to the thing being referenced. For example, reference 57 is a Nature paper on the development of an ML model for planning chemical syntheses. But the authors use it to reference a molecular fingerprint method (ECFP4) which is indeed used in the article but is not defined there. Equally, the literature on the prediction and discovery of hydrogels is not representative of the major steps in this field.
2. The authors make some claims that are not evidenced in their results - e.g., "Under the appropriate conditions, three hydrogels display a long lifetime stability of 6 months (1 and 2 in H₃BO₃ and Tris solutions, and 3 in AgNO₃ solution, Fig. 3c-e)". But Figure 3c-e only show photos of the hydrogels at a single time point? How is this 6 months timepoint to be evidenced?
3. The results shown in Figure 4d-e, are identical to the results in Figure 5a-b. This is probably a simple mistake as the figure captions indicate that the Figure 4 panels should show something else. But this carelessness is concerning and makes it very difficult to judge the veracity of the results.

In short, I like the idea, the work could be publishable but this manuscript needs a lot of work.

Response letter

Reviewer #1 (Remarks to the Author):

Comment #1: The manuscript is not written particularly clearly, and grammar is an issue. It is difficult to understand the sequence of events, going from thousands of descriptors down to a much smaller number. It is not clear from the text and figures whether the 4175 descriptors were calculated by a package, then the 4 types of fingerprint descriptors added, or whether the fingerprints were part of the 4175.

Response: We thank the referee for this good comment. We are sorry for the confusing description in our manuscript, and we have modified these parts in the revised manuscript. We would like to reply to this comment from the following two aspects.

1). It is not clear from the text and figures whether the 4175 descriptors were calculated by a package, then the 4 types of fingerprint descriptors added, or whether the fingerprints were part of the 4175.

The 4175 molecular descriptors and four kinds of molecular fingerprints are five different mathematical representations for describing molecules, the five mathematical representations were used separately to build prediction models for the hydrogel-forming ability of nucleoside derivatives in this work. Herein, we provide the detailed calculation processes of molecular descriptors and fingerprints as follows:

Firstly, molecular descriptors are defined as mathematical representations of molecules' properties that are generated by algorithms and can be calculated by the alvaDesc software (Version 2.0.12). The alvaDescCLIWrapper package (Version 1.1.1) was used to access alvaDesc functionalities from Python (Version 3.9.12) in this study. This package can compute up to 5666 descriptors for 33 domains, such as molecular properties, topological indices, and pharmacophore (Mauri, A., Ecotoxicological QSARs, Humana Press, 2020)¹. A

total of 5666 molecular descriptors were calculated for 71 nucleoside derivatives using alvaDesc software, and 4175 molecular descriptors were left for subsequent analysis by removing 1491 descriptors with missing values.

Secondly, Molecular fingerprints can convert the molecules into a series of binary representations, which are also called bit strings. The commonly used bit strings include four kinds of fingerprints (2048-bit ECFP4², ECFP6², Atom Pair³ and Topological Torsion⁴), which were all involved in this study. The four kinds of molecular fingerprints all include 2048-bits calculated by the RDKit package (Version 2022.3.4), which is an open-source toolkit for cheminformatics and is often used to perform molecular fingerprint generation⁵. Unlike the molecular descriptors, the individual bit string of fingerprints doesn't have specific chemical meanings, so we didn't perform the feature selection for the fingerprints.

Finally, the selected descriptors from each step of the feature selection and four kinds of molecular fingerprints were used to build the prediction models to predict the hydrogel-forming ability respectively. The molecular fingerprints are not involved in the construction of descriptors prediction models, and they constructed fingerprint prediction models, separately.

2). It is difficult to understand the sequence of events, going from thousands of descriptors down to a much smaller number.

The molecular descriptors which include 5666 descriptors for 71 nucleoside derivatives were obtained by using alvaDesc, and after removing 1491 descriptors with missing values, we obtained a preliminary set of 4175 descriptors. To improve the prediction model performance, we conducted feature selection to reduce the 4175 descriptors down to a much smaller number. The feature selection consisted of the following three steps.

Firstly, the rank-sum test was used as a univariate difference analysis. The results demonstrate that there are significant differences ($P < 0.05$) between the gelator ($n=38$) and non-gelator ($n=33$) groups (defined based on their hydrogel-

forming ability), which means it may have a potential association with the hydrogel-forming ability. In this study, 144 descriptors were obtained by removing the 4031 of 4175 descriptors that have no significant association with the hydrogel-forming ability (rank-sum test, $P > 0.05$).

Secondly, the Spearman correlation coefficient was calculated between the 144 descriptors in pairs. Then, the pair of descriptors whose correlation coefficient is higher than 0.8 ($Rho > 0.8$) was selected, and one of the pair is excluded to avoid collinearity. After this step, 40 descriptors were kept for subsequent model training.

Finally, we used machine learning (ML) algorithm-based recursive feature elimination (RFE) to obtain the optimal combination of descriptors and maximize model performance. Four ML algorithms including logistic regression (LR)⁶, decision tree (DT)⁷, random forest (RF)⁸ and extreme gradient boosting (XGBoost)⁹ were utilized in this work. The results indicated that the optimal number of descriptors for each ML algorithm is different (XGBoost, $n=16$; LR, $n=24$; DT, $n=30$; RF, $n=37$). After RFE, we obtained an LR model constructed with 24 descriptors with the best performance as the final prediction model (**Fig.R1**):

Fig.R1 Flow Chart of feature selection

Taken together, the molecular descriptors and four kinds of molecular fingerprints were used separately to build prediction models for the hydrogel-forming ability. The descriptors calculated by alvaDesc were further selected from each step of feature selection, and then used to build the descriptors prediction models. The four kinds of molecular fingerprints calculated by RDKit package were used to build the fingerprint prediction models, which were used

to predict the hydrogel-forming ability respectively. The prediction models built were shown in **Table R1** (The table is assigned to be **Supplementary Table 1** in revised Supporting Information).

Table R1. Characteristics of the prediction models

Features	Algorithms	Rows	Columns
Molecular Descriptors			
Descriptors-4175	LR, DT, RF, XGBoost [#]	71 [*]	4175
Descriptors-144	LR, DT, RF, XGBoost	71	144
Descriptors-40	LR, DT, RF, XGBoost	71	40
Descriptors-REF ^{&}	LR, DT, RF, XGBoost	71	16-37 ^{&}
Molecular Fingerprints			
ECFP4	LR, DT, RF, XGBoost	71	2048
ECFP6	LR, DT, RF, XGBoost	71	2048
AtomPair	LR, DT, RF, XGBoost	71	2048
Topological Torsion	LR, DT, RF, XGBoost	71	2048

Notes: ^{*} Rows: Whether the nucleoside derivatives have the hydrogel-forming ability; [#]Algorithms: Logistic regression (LR), decision tree (DT), random forest (RF), and extreme gradient boosting (XGBoost); [&]Descriptors-REF: Recursive feature elimination (REF) has different optimal descriptors for different Algorithms: LR, n=24; XGBoost, n=16; DT, n= 30; RF, n=37.

Reference:

1. Mauri, A. AlvaDesc: A Tool to Calculate and Analyze Molecular Descriptors and Fingerprints. Ecotoxicological QSARs. (Humana, New York, 2020).
2. Rogers, D. & Hahn, M. Extended-Connectivity Fingerprints. *J. Chem. Inf. Model.* **50**, 742-754 (2010).
3. Carhart, R. E., Smith, D. H. & Venkataraghavan, R. Atom pairs as molecular features in structure-activity studies: definition and applications. *J. Chem. Inf. Comput. Sci.* **25**, 64-73 (1985).
4. Nilakantan, R., Bauman, N., Dixon, J. S. & Venkataraghavan, R. Topological torsion: a new molecular descriptor for SAR applications. Comparison with other descriptors. *J. Chem. Inf. Comput. Sci.* **27**, 82-85 (1987).
5. RDKit: Open-source cheminformatics. <https://www.rdkit.org>. (2016).
6. Yu, H.-F., Huang, F.-L. & Lin, C.-J. Dual coordinate descent methods for logistic regression and maximum entropy models. *Mach. Learn.* **85**, 41-75 (2011).
7. Quinlan JR. Induction of decision trees. *Mach. Learn.* **1**, 81-106 (1986).
8. Breiman, L. Random Forests. *Mach. Learn.* **45**, 5-32 (2001).
9. Chen, T. & Guestrin, C. Proceedings of the 22nd ACM SIGKDD International Conference on Knowledge Discovery and Data Mining (Association for Computing Machinery, San Francisco, 2016).

Comment #2: Additionally, the authors state that because the number of data points available for the models was small (71) they ‘copied’ the data set to generate 5 copies, 4 of which were used to train the model and one was the ‘test set’. If this is true then such an approach is invalid, there is no independent test set to assess model predictivity for new data. The model is only predicting its training data. If overfitting occurs, then the model could look very good but be incapable of generalizing to new data. Perhaps the authors meant that the 71 compounds were split into 80% (57) for the training set and 20% (14) for the test set. This must be clarified.

Response: Thank you very much for the professional comments and we fully agree with this matter. What we want to express in the manuscript is that the 71 compounds were split into 80% (57) for the training set and 20% (14) for the test set. Fivefold stratified cross-validation was used to estimate the intervals of the parameters¹, which was widely used in the prediction model building (Enot, D. P. et al. *Nat. Protoc.*, 2008; Kohoutová, L. et al., *Nat. Protoc.*, 2020.)²⁻³. The 71 nucleoside derivatives were divided into five equal parts by maintaining the ratio of gelators to non-gelators and four of them were used as the training set to train the model and the other one as the test set. The model was trained five times without repetition. On top of this, 10 times random fivefold stratified cross-validations were performed to obtain more reliable information. In other words, 50 times of model training were performed.

In the revised manuscript we have corrected this misleading. "copied" was not supposed to be used here and we have corrected it as "subsets".

Reference:

1. Kohavi, R. Proceedings of the 14th international joint conference on Artificial intelligence (Morgan Kaufmann Publishers Inc., Montreal, 1995).
2. Enot, D. P. et al. Preprocessing, classification modeling and feature selection using flow injection electrospray mass spectrometry metabolite fingerprint data. *Nat. Protoc.* **3**, 446-470 (2008).
3. Kohoutová, L. et al. Toward a unified framework for interpreting machine-learning models in neuroimaging. *Nat. Protoc.* **15**, 1399-1435 (2020).

Comment #3: Inspection of Table S1 suggests that almost all models have similar predictivity when the uncertainties in their accuracies, expressed by multiple metrics) are compared. There is no obvious reason why LR should be ‘much’ better than the other nonlinear ML methods. XGBoost routinely provides a superior model prediction (often by a small margin)

Response: Thank you very much for this comment. We chose LR as the optimal model after comprehensively considering various parameters. The reason why we chose LR but not XGBoost is as follows:

i). The reason why LR was chosen as the optimal model:

Firstly, to test accuracy, AUC, precision, recall and F1 score are commonly used to determine the optimal model in related research (Theodoris, C. V. et al., *Nature*, 2023, AUC and F1 score; Jablonka, K. M. et al., *Nat. Chem.*, 2021, accuracy, precision, and recall; Han, T. et al., *Nat. Mach. Intell.*, 2022, AUC, precision and recall)²⁻⁴, the five parameters were used to evaluate the performance of models comprehensively here. To select the optimal model, we mainly focused on test accuracy and AUC, and the results of precision, recall and F1 score were also used as auxiliary references. In this study, LR not only provided better results of test accuracy (0.71 ± 0.01) and AUC (0.84 ± 0.02), but also had higher recall (0.95 ± 0.01) and F1 score (0.78 ± 0.01) (**Table R2; Fig. R2**, the figure is assigned to be **Fig. 2b** in revised manuscript). So, we finally chose LR as the optimal model.

ii). The reason why XGBoost was not the optimal model in our study:

Generally, LR is a classification model and is often used for binary classification. It is popular in the industry for its simplicity, parallelizability, and interpretability. The essence of LR is that the data is assumed to obey this distribution, and then the parameters are estimated using the maximum likelihood estimation. XGBoost is the abbreviation of "Extreme Gradient Boosting", which is a high-performance ML algorithm that benefits from great interpretability potential. It is fast and effective in processing large-scale data sets and does not require high

hardware resources such as memory.

As you said, XGBoost routinely provides a superior model prediction. However, as an ensemble learning algorithm based on tree boosting, XGBoost may not have a good performance in small dataset. In this study, there are only 71 nucleoside derivatives included, and just 38 of them own the hydrogel-forming ability, which is also one of the biggest bottlenecks affecting the development of the nucleoside hydrogel field. We suspect this may be one reason why XGBoost did not stand out in our results. In addition, LR was also found to be the optimal model in previous studies of prediction models building (Pavlović, M. et al., *Nat. Mach. Intell.*, 2021)⁵, and XGBoost was not always the optimal choice of a predictive model algorithm in some circumstances (Theodoris, C. V. et al., *Nature*, 2023)⁶.

Taken together, LR was chosen as the optimal prediction model in this study and this point was also discussed in the revised manuscript.

Fig. R2. Parameters of 4 models using descriptors after recursive feature elimination.

Table R2. Parameters of 4 models using descriptors after recursive feature elimination.

	Accuracy		AUC [#]		F1 Score		Precision		Recall	
	Mean	Se	Mean	Se	Mean	Se	Mean	Se	Mean	Se
LR*	0.71	0.01	0.78	0.01	0.66	0.01	0.95	0.01	0.84	0.02
RF	0.67	0.01	0.70	0.01	0.68	0.02	0.75	0.02	0.74	0.02
DT	0.59	0.02	0.63	0.02	0.60	0.01	0.69	0.03	0.63	0.02
XGBoost	0.70	0.02	0.73	0.01	0.72	0.02	0.76	0.02	0.79	0.02

Notes: *: Logistic regression (LR), decision tree (DT), random forest (RF), and extreme gradient boosting (XGBoost)

#: AUC: Area Under Curve

Reference:

1. Lipton, Z. C., Elkan, C. & Naryanaswamy, B. Optimal thresholding of classifiers to maximize F1 measure. *Mach. Learn. Knowl. Discov. Databases*. **8725**, 225–239 (2014).
2. Theodoris, C. V. et al. Transfer learning enables predictions in network biology. *Nature* **618**, 616–624 (2023).
3. Jablonka, K. M., Ongari, D., Moosavi, S. M. & Smit, B. Using collective knowledge to assign oxidation states of metal cations in metal–organic frameworks. *Nat. Chem.* **13**, 771–777 (2021).
4. Han, T. et al. Image prediction of disease progression for osteoarthritis by style-based manifold extrapolation. *Nat. Mach. Intell.* **4**, 1029–1039 (2022).
5. Pavlović, M. et al. The immuneML ecosystem for machine learning analysis of adaptive immune receptor repertoires. *Nat. Mach. Intell.* **3**, 936–944 (2021).
6. Theodoris, C. V. et al. Transfer learning enables predictions in network biology. *Nature* **618**, 616–624 (2023).

Comment #4: The descriptors chosen are quite arcane, making interpretation difficult. It is not clear why modern sparse feature selection method like LASSO or MLREM was not used to reduce the dimensionality of the descriptor space. It is not clear how the ranked sum test was used to reduce the number of descriptors.

Response: Thank you very much for the professional comments and suggestions. Feature selection is an important part of this study, we would like to response to these comments from the following aspects.

1). The descriptors chosen are quite arcane, making interpretation difficult.

Firstly, molecular descriptors are the mathematical representation of chemicals, and they serve as the input for the data analysis methods to build quantitative structure-activity relationships (QSAR) models. It is now widely used to construct prediction models of the physical, chemical, and biological properties of molecules. For instance, Tiihonen, A. et al. developed an input model that predicted antimicrobial activity of conjugated oligoelectrolyte molecules on the assumption of 21 descriptors (*J. Am. Chem. Soc.*, 2021)¹; Lyu, R. et al. reported a supervised ML workflow for the dimensionality prediction of low dimensional Pb-I perovskites on the basis of 21 descriptors (*J. Am. Chem. Soc.*, 2021)²; Ye, S. et al. developed a full-color-tunable polymer platform guided by ML algorithms based on 9 descriptors for Multiple linear regression with expectation maximization (MLREM) and 19 descriptors for Bayesian regularized artificial neural network with a Laplacian prior (BRANNLP) (*Chem*, 2022)³.

Secondly, we performed a comprehensive calculation of molecular descriptors to discover potential information that may be relevant to the formation of nucleoside hydrogel. And the descriptors in the final prediction model were chosen by the three-step feature selection mentioned above (**Comment #1**). Important features for the optimal model were mainly clustered in 2D matrix-based descriptors, edge adjacency indices, P_VSA-like descriptors, 2D atom pairs, 2D autocorrelations, atom-centered fragments, functional group counts, and pharmacophore descriptors. As these molecular descriptors express the underlying properties of molecules, there is no easy-to-understand explanation for the chosen descriptors. However, they have specific meanings, which are given in the **Supplementary Data S5**.

Thirdly, the conversion of molecular structures to molecular descriptors is currently a classical approach to building ML models in chemistry. Although the collective meaning of these molecular descriptors is difficult to understand, this does not affect the predictive performance of our optimal model based on 24

molecular descriptors (test accuracy, 0.71 ± 0.01 ; AUC, 0.84 ± 0.02 ; recall, 0.95 ± 0.01 ; F1 score, 0.78 ± 0.01). The tube-inversion tests were utilized for the selected nucleoside derivatives with high gelling probability, and the success rate of form hydrogel is 83.33% (10/12). Given the lack of prediction models for the hydrogel-forming ability of nucleoside derivatives, this study is an initial attempt and the molecular descriptors obtained are not mentioned in previous studies. However, further study of exploring understandable and effective descriptors to express nucleoside derivatives and improving the interpretability of these chosen descriptors, is an interesting topic and is what we are doing. We hope to provide a better interpretation of these molecular descriptors in the future.

According to your suggestion, we added the discussion about this issue in the revised manuscript.

2). It is not clear why modern sparse feature selection method like LASSO or MLREM was not used to reduce the dimensionality of the descriptor space.

Firstly, we agree with the matter that LASSO (Least Absolute Shrinkage and Selection Operator) and MLREM (Multiple Linear Regression with Expectation Maximization) are indeed commonly used methods for the selection of features using molecular descriptors. Secondly, we previously tried LASSO regression as a method of feature selection, and we implemented the feature selection method of LASSO through the scikit-learn package (Version 1.1.1) in Python (Version 3.9.12). After optimizing the hyperparameters of LASSO via cross-validation, feature selection was performed using LASSO and 70 molecular descriptors were obtained. Considering the parameters of the models based on 70 molecular descriptors were not as good as the results of the optimal model based on the three-step, we did not put these results in the manuscript. Thirdly, this is our neglect, and we are deeply sorry about that we did not pay attention to MLREM before. Thank you very much for your suggestion. According to your suggestion, we also used the MLREM for feature selection. We used the R

software (Version 4.2.2) to code the feature selection method of MLREM, and finally obtained 28 descriptors. The parameters of the model constructed on this basis were also not as good as the results of the optimal model based on the three-step. Detailed results of prediction models built based on LASSO and MLREM were presented in **Table R3** (The table is assigned to be **Supplementary Table 8** in revised Supporting Information). Now we have added them as a supplement in our revised manuscript. The codes were also added in the github (<https://github.com/leescu/NHGPM>).

Table R3. Characteristics of the models constructed by LASSO and MLREM for feature selection.

	LR*		RF		DT		XGBoost	
	Mean	SE	Mean	SE	Mean	SE	Mean	SE
LASSO#								
Accuracy	0.70	0.01	0.64	0.01	0.68	0.01	0.67	0.02
F1 score	0.73	0.02	0.68	0.02	0.72	0.01	0.71	0.01
Precision	0.72	0.02	0.66	0.01	0.69	0.02	0.68	0.02
Recall	0.77	0.02	0.72	0.02	0.78	0.02	0.75	0.02
AUC	0.74	0.02	0.73	0.02	0.69	0.02	0.74	0.02
MLREM								
Accuracy	0.63	0.01	0.67	0.01	0.68	0.01	0.68	0.02
F1 score	0.69	0.01	0.70	0.00	0.71	0.01	0.71	0.02
Precision	0.63	0.01	0.68	0.01	0.70	0.01	0.69	0.01
Recall	0.79	0.02	0.75	0.02	0.74	0.02	0.75	0.02
AUC	0.67	0.02	0.75	0.02	0.69	0.02	0.75	0.02

Notes: *: Logistic regression (LR), decision tree (DT), random forest (RF), and extreme gradient boosting (XGBoost)

#: LASSO: Least Absolute Shrinkage and Selection Operator, AUC: Area Under Curve, MLREM: Multiple linear regression with expectation maximization.

3). It is not clear how the ranked sum test was used to reduce the number of descriptors.

The rank sum test was used to feature selection and the specific steps are as follows: Firstly, the Wilcoxon rank test was used to test whether there is a significant difference of each molecular descriptor between gelator and non-gelator groups. Secondly, the descriptors without significant difference were excluded. Thirdly, the descriptors with significant difference ($P < 0.05$) were

retained and subjected to the next step of feature selection.

The reason why we used the rank sum test is that the univariate difference test including the rank sum test is a classical feature selection method and is one of the most used feature screening methods for ML (Liang, W. et al., *Clin. Cancer. Res.*, 2019; Errington, N. et al., *eBioMedicine*, 2021)^{4,5}. Considering the characteristics of the molecular descriptors involved in this study, we chose the rank sum test as a method of univariate feature selection to select the descriptors. We used the Wilcoxon test to independently test whether there is a significant difference between gelator and non-gelator groups and screened out features without significant relation with the hydrogel-forming ability ($P>0.05$). After selection, we obtained 144 molecular descriptors that were significantly related to the hydrogel-forming ability. We have added the corresponding description in the revised manuscript.

Reference:

1. Tiihonen, A. et al. Predicting Antimicrobial Activity of Conjugated Oligoelectrolyte Molecules via Machine Learning. *J. Am. Chem. Soc.* **143**, 18917-18931 (2021).
2. Lyu, R., Moore, C. E., Liu, T., Yu, Y. & Wu, Y. Predictive Design Model for Low-Dimensional Organic–Inorganic Halide Perovskites Assisted by Machine Learning. *J. Am. Chem. Soc.* **143**, 12766-12776 (2021).
3. Ye, S. et al. Machine learning-assisted exploration of a versatile polymer platform with charge transfer-dependent full-color emission. *Chem* **9**, 924-947 (2023).
4. Liang, W. et al. A Combined Nomogram Model to Preoperatively Predict Histologic Grade in Pancreatic Neuroendocrine Tumors. *Clin. Cancer. Res.* **25**, 584-5945 (2019)
5. Errington, N. et al. A diagnostic miRNA signature for pulmonary arterial hypertension using a consensus machine learning approach. *eBioMedicine* **69**, 103444 (2021).

Comment #5: In Figure S5 it is not clear why multiple versions of logP and PSA were used, surely these would have been highly intercorrelated and removed by the correlation filter?

Response: Many thanks for this comment. We apologize for the confusing description and now we have modified this part in the revised manuscript. These descriptors (logP and PSA) in original **Figure S5** were not among the 40 molecular descriptors obtained by the feature selection and did not participate in the construction of our prediction models. We just tried to explore the association between these molecular descriptors and the hydrogel-forming ability. The reason why multiple versions of logP and PSA is to explore their association from multiple perspectives. Detail descriptions are as follows:

Firstly, as the descriptors chosen in the final model are not easily to understand, we also try to initially explore the other descriptors which express chemical properties that may be relevant to the hydrogel-forming ability of nucleoside derivatives. According to previous studies (Van Lommel, R et al., *Chem. Sci.*, 2020; Li, F. et al., *Proc. Natl Acad. Sci. USA.*, 2019; Gupta, J. K. et al., *Chem. Sci.*, 2016)¹⁻³, related descriptors to water solubility and lipophilicity (hydrophilicity, Hy⁴; topological polar surface area, TPSA⁵; octanol-water partition coefficient, LogP ⁶; and solubility, ESOL⁷) were selected. We used RDKit calculated all kinds of Hy, TPSA, LogP and ESOL to explore the association between these molecular descriptors and the hydrogel-forming ability from multiple perspectives.

Then we tried to show the values' distributions of the four descriptors between the gelator and non-gelator groups, and between the gelator group and the top 10% of nucleoside derivatives with high probability of hydrogel-forming ability. The results demonstrated that there was no significant difference between groups ($P > 0.05$), suggesting that these molecular descriptors were not related to the hydrogel-forming ability of nucleoside derivatives.

Therefore, although the molecular descriptors we chose based on the above three-step feature selection are not easy to understand, it is more effective to

be used to predict the hydrogel-forming ability of nucleoside derivatives in this work. Since we did not find a link between these molecular descriptors and the hydrogel-forming ability of the nucleoside derivatives. To avoid confusing the reader, we replaced the original **Fig.3a** to **Supplementary Fig. 5** and removed the original **Figure S5**.

References:

1. Van Lommel, R., Zhao, J., De Borggraeve, W. M., De Proft, F. & Alonso, M. Molecular dynamics-based descriptors for predicting supramolecular gelation. *Chem. Sci.* **11**, 4226-4238 (2020).
2. Li, F. et al. Design of self-assembly dipeptide hydrogels and machine learning via their chemical features. *Proc. Natl Acad. Sci. USA.* **116**, 11259-11264 (2019).
3. Gupta, J. K., Adams, D. J. & Berry, N. G. Will it gel? Successful computational prediction of peptide gelators using physicochemical properties and molecular fingerprints. *Chem. Sci.* **7**, 4713-4719 (2016).
4. Todeschini, R., Vighi, M., Finizio, A. & Gramatica, P. 3D-Modelling and Prediction by WHIM Descriptors. Part 8. Toxicity and Physico-Chemical Properties of Environmental Priority Chemicals by 2D-TI and 3D-WHIM Descriptors. *SAR. QSAR. Environ. Res.* **7**, 173–193 (1997).
5. Ertl, P., Rohde, B. & Selzer, P. Fast calculation of molecular polar surface area as a sum of fragment-based contributions and its application to the prediction of drug transport properties. *J. Med. Chem.* **43**, 3714–3717 (2000).
6. Ghose, A. K., Viswanadhan, V. N. & Wendoloski, J. J. Prediction of Hydrophobic (Lipophilic) Properties of Small Organic Molecules Using Fragmental Methods: An Analysis of ALOGP and CLOGP Methods. *J. Phys. Chem. A* **102**, 3762–3772 (1998).
7. Delaney, J. S. ESOL: estimating aqueous solubility directly from molecular structure. *J. Chem. Inf. computer Sci.* **44**, 1000–1005 (2004).

Comment #6: Why does Table S1 have two entries for each of the 4 ML methods with 144 descriptors (“descriptors’ is misspelt)

Response: We feel deeply sorry about our mistake, and we have corrected it in our revised manuscript (**Table R4**).

Table R4. The corresponding updated part of **Supplementary Table 2**.

Models	Features	Test Accuracy		AUC	
		Mean	Se	Mean	Se
DT*	Descriptor_4175	0.65	0.01	0.65	0.02
LR	Descriptor_4175	0.65	0.02	0.67	0.02
RF	Descriptor_4175	0.63	0.01	0.72	0.02
XGBoost	Descriptor_4175	0.63	0.01	0.69	0.02
DT	Descriptor_144	0.64	0.01	0.64	0.01
LR	Descriptor_144	0.68	0.02	0.80	0.02
RF	Descriptor_144	0.67	0.01	0.75	0.02
XGBoost	Descriptor_144	0.64	0.02	0.72	0.02
DT	Descriptor_40	0.66	0.02	0.69	0.02
LR	Descriptor_40	0.70	0.01	0.81	0.02
RF	Descriptor_40	0.67	0.01	0.74	0.02
XGBoost	Descriptor_40	0.65	0.01	0.75	0.02
DT	Descriptor_REF#	0.59	0.02	0.63	0.02
LR	Descriptor_REF#	0.71	0.01	0.84	0.02
RF	Descriptor_REF#	0.67	0.01	0.75	0.02
XGBoost	Descriptor_REF#	0.70	0.02	0.79	0.02

Notes: *: Logistic regression (LR), decision tree (DT), random forest (RF), and extreme gradient boosting (XGBoost);

#: Descriptors-REF: Recursive feature elimination (REF) has different optimal descriptors for different Algorithms: LR, n=24; XGBoost, n=16; DT, n= 30; RF, n=37.

Comment #7: Even assuming the ML model was valid, using it to screen 7257 new nucleoside analogues is very unwise unless these new materials fall within or near the domain of applicability of the model. There is no discussion of what the domain is and whether the new nucleoside analogues are within it. It seems very unlikely that a model with limited accuracy could reliably predict the properties of >7000 new materials when trained on only 71 (or 57) nucleosides.

Response: Thank you very much for your comment. In fact, the 7257 nucleoside derivatives were screened based on their three-dimensional (3D) similarity from PubChem (<https://pubchem.ncbi.nlm.nih.gov>), and we tried to get their structures as close as possible to the trained 71 nucleoside derivatives' structures.

PubChem is an open repository of small molecules and their experimental biological activities; it contains over 115 million compounds and hundreds of thousands of which are nucleoside derivatives¹. As you know, the nucleosides are composed of nucleobases and pentose. To ensure the structural similarity, we obtained 7257 structures from PubChem based on 3D similarity by using PubChem3D project.

In our study, shape-Tanimoto (ST) and color-Tanimoto (CT) were used to estimate the 3D similarity. Only nucleoside derivatives owning the high 3D similarity with the five basic nucleosides (uridine, thymidine, adenosine, guanosine, and cytidine) were included, and the criteria are the $ST \geq 0.80$ and $CT \geq 0.50^2$. Therefore, we think the 7257 nucleoside derivatives, screened from PubChem based on the 3D similarity, may fall within or be close to the present model's applicability domain.

We applied our prediction model to the 7257 nucleoside derivatives which own the high 3D similarity with the nucleosides. And 12 nucleoside derivatives were selected with a high probability of hydrogel-forming ability for experimental validation (top 10% probability of gelator). The tube-inversion tests were utilized for these nucleoside derivatives, and the success rate of form hydrogel is 83.33%

(10/12, **Table R5**, this table is assigned to be **Supplementary Data S7**). This indicates that the prediction model is useable for these 7257 nucleoside derivatives based on 3D similarity. Therefore, we believe that the ML model was valid at this stage, and we have added related discussion in the revised manuscript to make it clearer according to your suggestions. Although this is just a starting point, better and more accurate models will be built as more nucleoside hydrogels are discovered in the future.

Table R5. The validation for the hydrogel-forming ability of the 12 nucleoside derivatives

No.	Nucleoside derivatives	PMID	P for Gelability	Rank for Gelability	Test Result
1	DTT 	21826754	0.680	2 (0.1%)	Gel (+)
2	XTS 	77518952	0.621	57 (0.7%)	Gel (-)
3	GMP 	135398631	0.610	118 (1.6%)	Gel (+)
4	IMP 	135398640	0.593	365 (5.0%)	Gel (+)
5	5-FUR 	9427	0.591	409 (5.6%)	Gel (-)

6	8-AG		135518164	0.590	454 (6.3%)	Gel (+)
7	dGMP		135596592	0.590	454 (6.3%)	Gel (+)
8	8-OHG		135407175	0.590	454 (6.3%)	Gel (+)
9	8-AzaG		135763231	0.587	553 (7.6%)	Gel (+)
10	I-5'-CA		13542524	0.585	599 (8.3%)	Gel (+)
11	2'-NH ₂ -dG		135491415	0.585	599 (8.3%)	Gel (+)
12	2'-OMe-dG		136441961	0.582	680 (9.4%)	Gel (+)

References:

1. Wang, Y. L. et al. Pubchem's bioassay database. *Nucleic Acids Res.* **40**, D400–D412 (2012).
2. Bolton, E. E., Kim, S. & Bryant, S. H. PubChem3D: Similar conformers. *J. Cheminform.* **3**, 13, (2011).

Comment #8: It was not explained why visual detection of Ag⁺ or cysteine was important.

Response: Thank you for your constructive comments. According to your valuable suggestion, the importance of visual detection of Ag⁺ or cysteine has been added into the revised manuscript. Herein, we would like to explain why visual detection of Ag⁺ or cysteine was important as follows.

Ag⁺ is a wide range of contaminants with serious toxicity, which is hardly degraded naturally in the environment and is easily accumulated in the human body through the food chain, causing increases of the risk of neurodegenerative, oncological, and cardiovascular diseases (Skvortsov, A. et al., *J. Hazard. Mater.*, 2023)¹. Cysteine plays an important role in life activities, and its concentration fluctuation is closely related to many diseases, such as neurodegenerative diseases (Paul B.D. et al, *Trends Pharmacol. Sci.*, 2018)². The traditional detection methods of Ag⁺ are atomic absorption spectrometry, inductively coupled plasma mass spectrometry and electrochemical analysis (Rievaj, M. et al., *Nanomaterials*, 2023)³. The traditional detection methods of cysteine are high-performance liquid chromatography, electrochemical analysis and flow injection analysis (Zhang R. et al, *Coordin. Chem. Rev.*, 2020)⁴. Although these methods have good selectivity and accuracy, they rely on expensive and large-scale instruments as well as specially trained personnel. The visual detection method is simple to operate with rapid measurement and eliminates the reliance on professional personnel and large equipment, so it has potential application in portable detection equipment for better practical application scenarios.

References

1. Skvortsov, A. N., Ilyechova, E. Y. & Puchkova, L. V. Chemical background of silver nanoparticles interfering with mammalian copper metabolism. *J. Hazard. Mater.* **451**, 131093 (2023).
2. Paul, B.D., Sbodio, J.I. & Snyder, S.H. Cysteine Metabolism in Neuronal Redox Homeostasis, *Trends Pharmacol. Sci.* **39**, 513-524 (2018).
3. Rievaj, M., Culková, E., Šandorová, D., Durdiak, J., Bellová, R. & Tomčík, P. A Review of Analytical Techniques for the Determination and

Separation of Silver Ions and Its Nanoparticles. *Nanomaterials* **13**, 1262 (2023).

4. Zhang, R., Yong, J., Yuan, J. & Xu, Z. P. Recent advances in the development of responsive probes for selective detection of cysteine. *Coordin. Chem. Rev.* **408**, 213182 (2020).

Comment #9: Care should be taken with copy editing e.g., careless use of RFE vs REF, glue-forming vs gel-forming etc.

Response: Thank you for your valuable comments. We apologize for such mistakes, and we have carefully checked and corrected the relevant problems.

Reviewer #2 (Remarks to the Author):

Comment #1: -Figures quality is insufficient.

Response: Thank you very much for your reminder. We have revised the insufficient figures to make them clearer.

Comment #2: Page 6: "The 3D PCA with 40 descriptors can better distinguish the two groups (Fig. 1e)." and legend to Figure 1e: "PCA results for 40 descriptors, and the two groups tend to separate." By looking at Figure 1, it is not obvious that the gelator/non-gelator groups are better separated in plot 1e than in plot 1b.

Response: Thanks for your comment and we fully agree with this matter. Initially, we used 3D PCA here to visually demonstrate whether the descriptors after feature selection could more representatively distinguish the gelator and non-gelator groups. Based on the results of the analysis of the subsequent models, the performance of the models constructed with 40 descriptors generally tended to outperform the models constructed with 4175 descriptors, and we believe that it played a positive role in the process of feature selection. However, considering that although there may be a potential trend in the 3D PCA (**Fig.1e**), it really does not distinguish gelator and non-gelator groups significantly, our terminology is not very appropriate. Therefore, we have changed the description of the 3D PCA in the revised manuscript according to your suggestion.

Comment #3: What probability threshold was used for the training and testing of the ML models? As discussed later on page 10 and shown in Figure 3b, the ML model predicts a compound as gelator or non-gelator with a given probability. When the models make good or bad predictions, it would be interesting to know the associated probabilities.

Response: Thanks for your professional comments. A probability threshold of 50% was preset for the training and testing of the ML models. When the prediction probability of gel formation is greater than 50%, the nucleoside derivative is considered a possible gelator, and when it is less than 50%, the nucleoside derivative is considered a possible non-gelator. Given the small sample size of this work this ML model is only a preliminary exploration of the nucleoside hydrogel prediction. Therefore, we primarily set the 50% as the probability threshold in this study. In **Fig. R3** (The **Fig. R3** is original **Fig. 3b**, and it is now shown in **Supplementary Fig. 6** in revised Supporting Information), the "Top 10%" is the range we chose to experimentally verify the hydrogel-forming ability of nucleoside derivatives, not the probability threshold of training and testing of the ML models, and the probability threshold of "Top 10%" is 58.11%.

According to your suggestion, the "probability threshold of ML models was 50%" and the "probability threshold of Top 10% was 58.11%" were also added in the revised manuscript and **Fig. R4** (also shown in **Fig. 3a** in revised manuscript).

Fig. R3. The 12 chosen alternative nucleoside derivatives based on gelator probabilities of top 10%. The x-axis representing the prediction hydrogel-forming probability and the y-axis representing the ranking of the prediction probabilities from smallest to largest.

Fig. R4. Prediction and verification of untested nucleoside derivatives. The nucleoside derivatives with the top 10% prediction probability of hydrogel-forming ability were selected and 12 nucleoside derivatives were selected in a relatively homogeneous manner based on our previous experience and the costs of obtaining and synthesizing nucleoside derivatives.

Comment #4: Page 8: "Beside the 5 descriptors mentioned above, some other descriptors are likely to be related to hydrogel-forming ability. Including hydrophilic factor (Hy) 63, topological polar surface area (TPSA), octanol-water partition coefficient (LogP), and solubility (ESOL), while no significant difference was found (Fig. 3a and S5)." and legend to Figure 3a: "grouped box plot of 4 descriptors with potential relevance to glue-forming ability for 71 nucleoside derivatives". Why would these 4 compounds be relevant since they cannot distinguish gelator/non-gelator groups? I don't see the pertinence of this.

Response: Many thanks for this comment. We apologize for the wrong description and now we have modified this part in the revised manuscript. These 4 descriptors in original **Fig. 3a** were not directly relevant to the hydrogel-forming ability for 71 nucleoside derivatives. These descriptors expressed some other chemical properties that may be relevant to hydrogel-forming ability of nucleoside derivatives according to the previous studies. We just tried to explore the relationship between these molecular descriptors with the hydrogel-forming ability, which is described in detail as follows:

Firstly, as the descriptors chosen in the final model are not easy to understand, we also try to initially explore the other descriptors which express chemical properties that may be relevant to the hydrogel-forming ability of nucleoside derivatives. According to the previous studies (Van Lommel, R et al., *Chem. Sci.*, 2020; Li, F. et al., *Proc. Natl Acad. Sci. USA.*, 2019; Gupta, J. K. et al., *Chem. Sci.*, 2016)¹⁻³, related descriptors to water solubility and lipophilicity (hydrophilicity, Hy⁴; topological polar surface area, TPSA⁵; octanol-water partition coefficient, LogP⁶; and solubility, ESOL⁷) were selected.

Then we tried to show the values' distributions of the four descriptors between the gelator and non-gelator groups, and between the gelator group and the top 10% of nucleoside derivatives with high probability of hydrogel-forming ability. The results demonstrated that there was no significant difference between groups ($P > 0.05$), suggesting that these molecular descriptors were not related

to the hydrogel-forming ability of nucleoside derivatives.

Therefore, although the molecular descriptors we chose based on the above three-step feature selection are not easy to understand, it is more effective to be used to predict the hydrogel-forming ability of nucleoside derivatives in this work. Since we did not find a link between these molecular descriptors and the hydrogel-forming ability of the nucleoside derivatives. To avoid confusing the reader, we replaced the original **Fig.3a** with **Supplementary Fig. 5** and removed the original **Fig. S5**.

References:

1. Van Lommel, R., Zhao, J., De Borggraeve, W. M., De Proft, F. & Alonso, M. Molecular dynamics-based descriptors for predicting supramolecular gelation. *Chem. Sci.* **11**, 4226-4238 (2020).
2. Li, F. et al. Design of self-assembly dipeptide hydrogels and machine learning via their chemical features. *Proc. Natl Acad. Sci. USA.* **116**, 11259-11264 (2019).
3. Gupta, J. K., Adams, D. J. & Berry, N. G. Will it gel? Successful computational prediction of peptide gelators using physicochemical properties and molecular fingerprints. *Chem. Sci.* **7**, 4713-4719 (2016).
4. Todeschini, R., Vighi, M., Finizio, A. & Gramatica, P. 3D-Modelling and Prediction by WHIM Descriptors. Part 8. Toxicity and Physico-Chemical Properties of Environmental Priority Chemicals by 2D-TI and 3D-WHIM Descriptors. *SAR. QSAR. Environ. Res.* **7**, 173-193 (1997).
5. Ertl, P., Rohde, B. & Selzer, P. Fast calculation of molecular polar surface area as a sum of fragment-based contributions and its application to the prediction of drug transport properties. *J. Med. Chem.* **43**, 3714-3717 (2000).
6. Ghose, A. K., Viswanadhan, V. N. & Wendoloski, J. J. Prediction of Hydrophobic (Lipophilic) Properties of Small Organic Molecules Using Fragmental Methods: An Analysis of ALOGP and CLOGP Methods. *J. Phys. Chem. A* **102**, 3762-3772 (1998).
7. Delaney, J. S. ESOL: estimating aqueous solubility directly from molecular structure. *J. Chem. Inf. computer Sci.* **44**, 1000-1005 (2004).

Comment #5: -Page 10: "Based on previous experience and knowledge of the hydrogel-forming ability of nucleoside derivatives, we selected three guanosine derivatives for validation among the top 10% ranked structures." and "The specific structures and gelator probabilities of the three nucleoside derivatives are in the forefront (Fig. 3c-e, probabilities of gelator: 1, 0.590; 2, 0.590; 3, 0.587)."

In Figure 3b the vast majority of the compounds have a gelator probability between 0.4 and 0.6. That does not seem to be very strong. Also, the authors selected 3 compounds out of the top 10%, however, 10% of 7257 compounds tested is 726, which is quite a large subset. What about the compounds that are at the very top of the distribution? Selection was made based on "previous experience and knowledge": how were these 3 compounds precisely selected?

Also in Figure 3b, I assume that the blue dots (labelled as "alternative") correspond to the 3 selected molecules? Note also that they are just above the 10% cut-off and not at the very top of distribution.

Response: Thank you very much for the professional comments and suggestions. We would like to explain this comment in three parts.

Firstly, as the formation of nucleoside hydrogels is a complex process, there were no prediction models available to predict the hydrogel-forming ability of nucleoside derivatives in the past. Inspired by these, we collated the structures of all published nucleoside hydrogels as a dataset through a systematic literature review. Then the dataset of 71 nucleoside derivatives with the information of whether they have the hydrogel-forming ability was converted into feature matrices using 4175 molecular descriptors and four kinds of fingerprints (2048-bit ECFP4, ECFP, AtomPair, and Topological Torsion). After three-step feature selection and hyperparameter optimization, four classifier algorithms (RF, DT, LR, XGBoost) were used for predicting hydrogel-forming ability. Then an optimal model with 71% accuracy was selected, and most distributions of gelling probabilities are in the range of 0.4-0.6. The present prediction model is our initial attempt, but we still hope that the present model

can be applied to the prediction of nucleoside hydrogels.

Secondly, although the predicted gelling probabilities were not strong, the tube-inversion tests were utilized for the selected nucleoside derivatives with high gelling prediction probability, and the success rate of form hydrogel is 83.33%. Initially, to validate the model and consider possible subsequent applications, the nucleoside derivatives with the top 10% probability of gelators were screened and 12 nucleoside derivatives (**1**, 1-[3,4-Dihydroxy-5-(hydroxymethyl)oxolan-2-yl]-1,3,5-triazinane-2,4,6-trione, DTT; **2**, xanthosine, XTS; **3**, guanine 5'-monophosphate, GMP; **4**, inosine 5'-monophosphate, IMP; **5**, 5-fluorouridine, 5-FUR; **6**, 8-aminoguanosine, 8-AG; **7**, 2'-deoxyguanosine 5'-monophosphate, dGMP; **8**, 8-hydroxyguanosine, 8-OHG; **9**, 8-azaguanosine, 8-azaG, **10**, inosine-5'-carboxylic acid; I-5'-CA; **11**, 2'-amino-2'-deoxyguanosine, 2'-NH₂-dG, and **12**, 2'-O-Methylguanosine, 2'-OMe-dG) were selected in a relatively homogeneous manner based on our previous experience and the costs of obtaining and synthesizing nucleoside derivatives. The experiment results indicated that 10 of the 12 nucleoside derivatives can form hydrogels (**1**, **3**, **4**, **6**, **7**, **8**, **9**, **10**, **11**, and **12**, **Table R5**, this table is assigned to be **Supplementary Data S7, Fig. R5-R7**, these figures are assigned to be **Supplementary Figure 7-9** in revised Supporting Information), which further demonstrates the value of our predictive model. Therefore, this study greatly accelerates the discovery of new nucleoside hydrogels compared to previous inadvertent discoveries or modifications of existing gelators.

Thirdly, as our group mainly focused on developing guanosine-based hydrogels and studying their application in stomatology¹⁻⁴, we paid more attention to guanosine-derived hydrogels during the process of verifying whether nucleoside derivatives can form hydrogels. Among the 12 nucleoside derivatives we selected to validate the model, five guanosine derivatives (**6**, **8**, **9**, **11**, and **12**) could form hydrogels. Interestingly, **6**, **8**, and **9** formed rarely reported cation-independent guanosine hydrogels. Therefore, we chose the three compounds for further study in the manuscript before. Thanks for your

valuable suggestion, we've added the detailed screening process of hydrogels in the revised manuscript.

Table R5. The validation for the hydrogel-forming ability of the 12 nucleoside derivatives

No.	Nucleoside derivatives	PMID	P for Gelability	Rank for Gelability	Test Result
1	DTT 	21826754	0.680	2 (0.1%)	Gel (+)
2	XTS 	77518952	0.621	57 (0.7%)	Gel (-)
3	GMP 	135398631	0.610	118 (1.6%)	Gel (+)
4	IMP 	135398640	0.593	365 (5.0%)	Gel (+)
5	5-FUR 	9427	0.591	409 (5.6%)	Gel (-)
6	8-AG 	135518164	0.590	454 (6.3%)	Gel (+)

7	dGMP		135596592	0.590	454 (6.3%)	Gel (+)
8	8-OHG		135407175	0.590	454 (6.3%)	Gel (+)
9	8-AzaG		135763231	0.587	553 (7.6%)	Gel (+)
10	I-5'-CA		13542524	0.585	599 (8.3%)	Gel (+)
11	2'-NH ₂ -dG		135491415	0.585	599 (8.3%)	Gel (+)
12	2'-OMe-dG		136441961	0.582	680 (9.4%)	Gel (+)

Notes: *: **1**, 1-[3,4-Dihydroxy-5-(hydroxymethyl)oxolan-2-yl]-1,3,5-triazinane-2,4,6-trione, DTT; **2**, xanthosine, XTS; **3**, guanine 5'-monophosphate, GMP; **4**, inosine 5'-monophosphate, IMP; **5**, 5-fluorouridine, 5-FUR; **6**, 8-aminoguanosine, 8-AG; **7**, 2'-deoxyguanosine 5'-monophosphate, dGMP; **8**, 8-hydroxyguanosine, 8-OHG; **9**, 8-azaguanosine, 8-azaG, **10**, inosine-5'-carboxylic acid; I-5'-CA; **11**, 2'-amino-2'-deoxyguanosine, 2'-NH₂-dG, and **12**, 2'-O-Methylguanosine, 2'-OMe-dG.

Fig. R5. Photographs of hydrogels or samples assembled from nucleoside derivatives in different solutions. Sol: solution. Pre: precipitate.

Fig. R6. Photographs of hydrogels or samples assembled from nucleoside derivatives in different solutions. Sol: solution. Pre: precipitate.

Fig. R7. Photographs of hydrogels or samples assembled from nucleoside derivatives in different solutions. Sol: solution. Pre: precipitate.

Reference:

1. Zhao, H., Guo, X., He, S. et al. Complex self-assembly of pyrimido[4,5-d] pyrimidine nucleoside supramolecular structures. *Nat. Commun.* **5**, 3108 (2014).
2. Zhao, H., Feng, H., Liu, J., Tang, F., Du, Y., Ji, N., Xie, L., Zhao, X., Wang, Z. & Chen, Q. Dual-functional guanosine-based hydrogel integrating localized delivery and anticancer activities for cancer therapy. *Biomaterials* **230**, 119598 (2020).
3. Wang, Z. et al. High-Strength and Injectable Supramolecular Hydrogel Self-Assembled by Monomeric Nucleoside for Tooth-Extraction Wound Healing. *Adv. Mater.* **34**, 2108300, (2022).
4. Liu, T. et al. pH-responsive dual-functional hydrogel integrating localized delivery and anti-cancer activities for highly effective therapy in PDX of OSCC. *Mater. Today* **62**, 71-97, (2023).

Comment #6: p10 and Fig 3. Why lifetime hydrogels were tested in H₃BO₃ with Tris solution for 8-AG and 8-OHG or AgNO₃ solution for 8-AzaG. p10 and Fig 3. Why H₃BO₃ with Tris solution and AgNO₃ solution was used as the solvent to form hydrogel?

Response: We sincerely appreciate your valuable comments. To better explain the former comment, we try to respond to the latter comment first.

1). p10 and Fig 3. Why H₃BO₃ with Tris solution and AgNO₃ solution was used as the solvent to form hydrogel?

A major challenge in the field of nucleoside hydrogels is how to anticipate whether a nucleoside derivative will form a hydrogel. Design suggestions are frequently made, but gelators are usually discovered unintentionally or through the synthetic modification of an existing gelator (Adams, D. J., *J. Am. Chem. Soc.*, 2022) ¹. Inspired by these, we developed a ML model to predict the hydrogel-forming ability of nucleoside derivatives. However, the ML model couldn't predict the appropriate conditions for the formation of nucleoside hydrogels because the self-assembly process of nucleoside hydrogels is complex, and always affected by many factors. Based on our previous experience and related literature, the hydrogel-forming conditions of guanosine derivatives are summarized as the following four aspects:

i). Alkali metal cations mostly Na⁺ and K⁺ in stabilizing the G-quartet were used to help form guanosine-derived hydrogels (Sreenivasachary N. et al. *Proc. Natl. Acad. Sci. USA*, 2005; Davis J.T., *Angew. Chem. Int. Ed.*, 2004)²⁻³.

ii). The borate anions in the alkaline solution connect with two ribose sugar units of guanosines via the formation of diesters bonds, thus increasing the stability of guanosine-derived hydrogels (Zhao, H. et al., *Biomaterials*, 2020; Peters, G. M. et al., *J. Am. Chem. Soc.*, 2014)⁴⁻⁵.

iii). Some metal ions such as Ag⁺ serve as bridges linking two base pair motifs of guanosines, leading to the formation of hydrogels (Li, T. et al., *ACS Appl. Mater. Inter.*, 2023; Adhikari B. et al., *J. Mater. Chem.B.*, 2014)⁶⁻⁷.

iv). Introducing mixture gelators favors the formation of binary gels (Plank, T. N. et al., *Chem. Commun.*, 2016; Das, R. N. et al., *Chem. - A Eur. J.*, 2012)⁸⁻⁹.

Therefore, we made attempts to verify the hydrogel-forming abilities of 8-AG, 8-OHG, and 8-AzaG according to the above methods. The results of the tube-inversion test show that 8-AG and 8-OHG couldn't form hydrogels in water, NaCl, KCl, H₃BO₃, and AgNO₃ solutions (**Fig. R6**), and self-assemble into hydrogels (8AG-T and 8OHG-T hydrogels) in the presence of H₃BO₃ and Tris. 8-AzaG failed to form hydrogels in water, NaCl and KCl solutions, formed viscous solution in H₃BO₃ solution, and successfully self-assembled into 8Aza-T and 8AzaG-Ag⁺ hydrogels in H₃BO₃ with Tris solution and AgNO₃ solutions, respectively (**Fig. R7**). Therefore, H₃BO₃ with Tris solution and AgNO₃ solution were used as the solvents to form hydrogels.

2). p10 and Fig 3. Why lifetime hydrogels where tested in H3BO3 with Tris solution for 8-AG and 8-OHG or AgNO3 solution for 8-AzaG

The reasons why we tested the lifetime hydrogels in H₃BO₃ with Tris solution for 8-AG and 8-OHG or AgNO₃ solution for 8-AzaG are as follows:

i). Interestingly, the 8AG-T, 8OHG-T, and 8AzaG-T hydrogels we developed are rarely reported cation-independent guanosine-derived supramolecular hydrogels. As 8-AG and 8-OHG exhibited the same gel-forming abilities, we made an in-depth study of 8AG-T together with 8OHG-T hydrogels. Therefore, we tested the lifetime hydrogels in H₃BO₃ with Tris solution for 8-AG and 8-OHG.

ii). For the 8AzaG-Ag⁺ hydrogel, the introduction of Ag⁺ might make it exhibit bactericidal properties and have potential biomedical applications, so we tested the lifetime hydrogels in AgNO₃ for 8-AzaG.

For the readers to better understand the screening processes of the solvents of hydrogels, we have added the results that show the processes of validating the hydrogel-forming abilities of 8-AG, 8-OHG, and 8-AzaG in H₂O, KCl, NaCl, H₃BO₃, H₃BO₃ and Tris, NaB(OH)₄, KB(OH)₄, and AgNO₃ solutions in revised manuscript.

References:

1. Adams, D. J. Personal Perspective on Understanding Low Molecular Weight Gels. *J. Am. Chem. Soc.* **144**, 11047-11053 (2022).
2. Sreenivasachary, N. & Lehn, J.M. Gelation-driven component selection in

- the generation of constitutional dynamic hydrogels based on guanine-quartet formation. *Proc. Natl. Acad. Sci. USA* **102**, 5938-5943 (2005).
3. Davis, J.T. G-quartets 40 years later: from 5'-GMP to molecular biology and supramolecular chemistry. *Angew. Chem. Int. Ed.* **43**, 668-698 (2004).
 4. Zhao, H., Feng, H., Liu, J., Tang, F., Du, Y., Ji, N., Xie, L., Zhao, X., Wang, Z. & Chen, Q. Dual-functional guanosine-based hydrogel integrating localized delivery and anticancer activities for cancer therapy. *Biomaterials* **230**, 119598 (2020).
 5. Peters, G. M., Skala, L. P., Plank, T. N., Hyman, B. J., Manjunatha Reddy, G. N., Marsh, A., Brown, S. P. & Davis, J. T. A G4·K⁺ Hydrogel Stabilized by an Anion. *J. Am. Chem. Soc.* **136**, 12596-12599 (2014).
 6. Li, T., Luo, Y., Wu, S., Xia, X., Zhao, H., Xu, X. & Luo, X. Super-Rapid In Situ Formation of a Silver Ion-Induced Supramolecular Hydrogel with Efficient Antibacterial Activity for Root Canal Disinfection. *ACS Appl. Mater. Inter.* **15**, 29854-29865 (2023).
 7. Adhikari, B., Shah, A. & Kraatz, H.B. Self-assembly of guanosine and deoxy-guanosine into hydrogels: monovalent cation guided modulation of gelation, morphology and self-healing properties, *J. Mater. Chem. B.* **2**, 4802-4810 (2014).
 8. Plank, T.N., Skala, L.P. & Davis, J.T. Supramolecular Hydrogels for Environmental Remediation: G4-quartet Gels that Selectively Absorb Anionic Dyes from Water. *Chem. Commun.* **53**, 6235-6238 (2016).
 9. Das, R. N., Kumar, Y. P., Pagoti, S., Patil, A. J. & Dash, J. Diffusion and Birefringence of Bioactive Dyes in a Supramolecular Guanosine Hydrogel. *Chem. - A Eur. J.*, **18**, 6008–6014 (2012).

Comment #7: p11 "This is the first-time cation-independent guanosine-derived hydrogels have been discovered." Not correct see for example: i) ACS Omega 2018, 3, 2, 2230–2241, <https://doi.org/10.1021/acsomega.7b02039>, ii) soft mater DOI <https://doi.org/10.1039/C8SM00299A>, iii) Organic & Biomolecular Chemistry 2018, DOI: 10.1039/C8OB01023D, iv) Coordination Chemistry Reviews Volume 488, 1 August 2023, 215170 <https://doi.org/10.1016/j.ccr.2023.215170>, etc

Response: Thank you for your careful review. The responses to the four references are as follows:

i). The review of "ACS Omega 2018, 3, 2, 2230–2241, <https://doi.org/10.1021/acsomega.7b02039>" mentioned that "Interestingly, the G-dots could utilize the self-assembly property of 5'-GMP to form fluorescent hydrogels without any externally added monovalent cations". In the original reference (Ghosh, A. et al., *Chem. Commun.*, 2016) ¹, 5'-GMP is guanosine 5'-monophosphate, sodium salt, which itself contains sodium. Furthermore, the article mentioned "the presence of a small number of guanine motifs of 5'-GMP would facilitate the self-assembly of the G-dots with the help of sodium to form extended structures", indicating the hydrogel formed with the help of sodium. The scheme (**Fig. R8**) in the article also illustrates the hydrogel forms by templating alkali metal cations of sodium. Therefore, the hydrogels in this reference are not cation-independent hydrogels.

[REDACTED]

Fig. R8. Formation of 5'-GMP carbon dots (G-dots).

ii). In the article “soft mater DOI <https://doi.org/10.1039/C8SM00299A>”, the gelator is guanosine 5'-monophosphate, potassium salt. The presence of potassium suggests the formed hydrogel is not cation independent.0

iii). We would like to explain the differences between the article of “Organic & Biomolecular Chemistry 2018, DOI: 10.1039/C8OB01023D” and the description of “the first-time cation-independent guanosine-derived hydrogels” in the following two aspects. Firstly, guanosine-3'- (1,2-dipalmitoyl-sn-glycero-3-phosphate) (diC16-3'-dG) formed hydrogel in waters, without any externally added cations. However, diC16-3'-dG itself contains the cation of Et_3NH^+ , which may help form gel, because G4-quartets may readily form in the presence of ammonium ions or quaternary ammonium ions (Peters, G.M. et al., *Chem. Soc. Rev.*, 2016)². Secondly, diC16-3'-dG is a derivative of guanylic acid, not guanosine. However, in our description is guanosine derivative. Therefore, the description of “This is the first-time cation-independent guanosine-derived hydrogels have been discovered” is not in conflict with the findings in the article of “Organic & Biomolecular Chemistry 2018, DOI: 10.1039/C8OB01023D”.

iv). We have read the review of “Coordination Chemistry Reviews Volume 488, 1 August 2023, 215170 <https://doi.org/10.1016/j.ccr.2023.215170>” carefully. **Table R6** is the summary of different guanosine-boronate esters-based self-assembled hydrogels and their biomedical applications, and it shows that metal cations are used to form hydrogels except for the reference of 55 (Plank T. N. at al., *Chem. Commun.*, 2016)³. However, we found that in the reference of 55, the hydrogel was made by 5'-deoxy-5'-iodoguanosine and KB(OH)_4 , and K^+ coordinated G-quartet assembly (**Fig. R9**). Therefore, the hydrogels in this reference are not cation-independent hydrogels.

Based on these, the results in the above references are not in conflict with the description of "This is the first-time cation-independent guanosine-derived hydrogels have been discovered". To avoid the misunderstanding of readers, we would like to revise it to be more appropriate and rigorous based on your professional comments. “the first-time cation-independent guanosine-derived hydrogels” has been changed to “the rarely reported cation-independent guanosine-derived hydrogels” in the revised manuscript:

Table R6. Summary of different guanosine-boronate esters based on self-assembled hydrogels and their biomedical applications.

Sl No	Boric/Boronic Acid derivative	Guanosine/metal cations	Incorporation of Other Components	Investigation of hydrogels in a purpose/application	Reference
1		G/K ⁺	–	Study of self-assembly of G4K ⁺ hydrogel in presence of boric acid	52
2		G/Li ⁺ , G/Na ⁺ , G/K ⁺ , G/Rb ⁺ and G/Cs ⁺	–	Investigation of mechanism of G-quartet formation and effect of different cations on the stability of GB hydrogel	53
3		G/Li ⁺	–	Study of Thioflavin T as a molecular chaperone for G4 hydrogel	54
4		5'-deoxy-5'-iodoguanosine	–	Demonstration of self-destruction of Guanosine borate hydrogel	55
5		G/Na ⁺	Isoguanosine	Dual role as a drug carrier and antitumor agent	72
6		G/Na ⁺	–	On demand release of Acyclovir drug	73
7	Boric Acid	G/Na ⁺	α,β,γ,δ-Tetrakis(1-methylpyridinium-4-yl) porphyrin p-Toluenesulfonate (TMPyP)	hemoperfusion device (blood lead elimination)	102
8		G/K ⁺	Dopamine-conjugated Pt (IV) complex	Photoactivatable Anticancer hydrogels	82
9		G/K ⁺	Hem, Dox	Release of doxorubicin and hemin for cascade chemo dynamic therapy (CDT)	88
10		G/Na ⁺	Dopamine and Au nanoparticles	Photothermal therapy antitumor activity	89
11		G/K ⁺	Recombinant human collagen	Wound healing application	70
12		L-G/K ⁺ and D-G/K ⁺	–	3D cell culture	75
13		G/K ⁺	Hem, Au Nanoparticles	Electrochemical detection for extracellular H ₂ O ₂	103
14		G/K ⁺	Hem, hyaluronic acid, polyaniline	Rapid wound care application	76
15		G/K ⁺	Polydopamine	3D bioprinting and tissue engineering	99
16	Phenyl boronic acid, 4-nitrophenyl boronic acid and 4-methoxyphenyl boronic acid	G/K ⁺	–	Bio inks for 3D bioprinting	57
17		G/K ⁺	tris(2-aminoethyl) amine	Zero order drug release	64
18		G/K ⁺	4-arm-NH ₂ -polyethylene glycol (PEG)	Delivery of doxorubicin drug	65
19	2-Formylphenylboronic acid	G/K ⁺	aminoglycoside	Stimuli-responsive antibacterial hydrogel	66
20		G/K ⁺	Putrescine, Hem, GOx	A cascade reactor for antibacterial wound dressing	94
21		G/K ⁺	–	A highly K ⁺ selective hydrogel used to determine K ⁺ level in human blood serum	100
22	4-Formylphenylboronic acid	G/K ⁺	cytosine-functionalized nucleopeptide	Inherent antibacterial hydrogel	68
23	Pyridine-4-boronic acid	G/K ⁺	–	Electrochemical biosensing applications	59
24	1-Naphthaleneboronic acid	G/K ⁺	–	pH responsive release of vitamins and doxorubicin	58
25		G/K ⁺ , G/Ba ²⁺	Mg ²⁺	Cell growths application	60
26	Benzene-1,4-diboronic acid or 1,4-Phenylenediboronic acid	G/K ⁺	Tannic acid	Antibacterial hydrogel for wound dressing	61
27		G/K ⁺	Rutin	For treatment of inflammatory bowel disease	77
28		isoG/K ⁺	Catechin	Antitumor activity (HPV-associated OSCC)	84
29		G/K ⁺	–	Antitumor activity on PDX of OSCC	85

[REDACTED]

Fig. R9. A hydrogel is made when guanosine or 5'-deoxy-5'-iodoguanosine reacts with KB(OH)_4 to form GB esters that self-assemble into G_4 -wires stabilized by K^+ .

References

1. Ghosh, A., Parasar, B., Bhattacharyya, T. & Dash, J. Chiral carbon dots derived from guanosine 5'-monophosphate form supramolecular hydrogels. *Chem. Commun.*, **52**, 11159-11162 (2016).
2. Peters, G.M. Davis & J.T. Supramolecular gels made from nucleobase, nucleoside and nucleotide analogs, *Chem. Soc. Rev.* **45**, 3188-3206 (2016).
3. Plank, T. N. & Davis, J. T. A $\text{G}_4\text{-K}^+$ hydrogel that self-destructs. *Chem. Commun.*, **52**, 5037-5040 (2016).

Comment #8: p11 Results very similar to Org. Biomol. Chem (DOI: 10.1039/C8OB01023D)

Response: Thank you for your insightful comments. Although there are some similarities between the hydrogels in the article of Org. Biomol. Chem (DOI: 10.1039/C8OB01023D) and that in our manuscript, there are many differences in their substance. We would like to expound on their differences in the following four aspects:

i). Chemical structures. The gelator of guanosine-3'- (1,2-dipalmitoyl-sn-glycero-3-phosphate) (diC16-3'-dG) is a derivative of guanylic acid, not guanosine. The gelators of cation-independent hydrogels are guanosine derivatives.

ii). Ammonium salts. G-quartets may readily form in the presence of ammonium ions or quaternary ammonium ions (Peters, G.M. et al., *Chem. Soc. Rev.*, 2016)¹, thus self-assemble into gels. Actually, the diC16-3'-dG itself contains Et₃NH⁺, which may contribute to forming hydrogels. In our study, no ammonium ions exist.

iii). Self-assembly mechanisms. There are two main differences in self-assembly mechanisms between the former article and our study. Firstly, the dynamic borate diester bonds helped form stable and self-healing hydrogels in the absence of cations. Secondly, the stacked G-quartets displays the bands of opposite sign at 240 and 260 nm in circular dichroism spectra (Peters, G. M. et al., *J. Am. Chem. Soc.*, 2014)². diC16-3'-dG' a positive band at 240 nm and a negative band at 260 nm, suggesting diC16-3'-dG might form stacked G-quartets. However, 8-AG and 8-OHG formed G-ribbons but not G-quartets in 8AG-T and 8OHG-T hydrogels.

iv). Application prospects. The diC16-3'-dG hydrogel exhibited drug-controlled release properties, so it has application prospects in drug delivery. The 8OHG-T hydrogel in this work can be used for rapid visual detection of Ag⁺ and cysteine, and thus it has potential application in portable detection equipment for Ag⁺ or cysteine detection in the future.

Taken together, the hydrogels in the article of Org. Biomol. Chem (DOI:

10.1039/C8OB01023D) and our manuscript are differences in the chemical structures, ammonium salts, self-assembly mechanisms, and application prospects.

References

1. Peters, G.M. Davis & J.T. Supramolecular gels made from nucleobase, nucleoside and nucleotide analogs, *Chem. Soc. Rev.* **45**, 3188-3206 (2016).
2. Peters, G. M., Skala, L. P., Plank, T. N., Hyman, B. J., Manjunatha Reddy, G. N., Marsh, A., Brown, S. P. & Davis, J. T. A G4·K⁺ Hydrogel Stabilized by an Anion. *J. Am. Chem. Soc.* **136**, 12596-12599 (2014).

Comment #9: Fig. 4c. G' is not >> G''

Response: Thank you for your valuable comments. The commonly used methods of confirming hydrogel formation are tube-inversion test and rheological measurements (Zhou, X. et al., *Adv. Sci.*, 2020; Zhong, R. et al., *Adv. Mater.*, 2018)¹⁻². To verify the formation of hydrogels, firstly, we performed the tube-inversion tests, and the results demonstrated 8AG-T and 8OHG-T hydrogels were successfully formed. Then, rheological measurement was used to assess the solid-like characteristics of hydrogels. If the storage modulus (G') is higher than the loss modulus (G''), the sample displays a solid-like characteristic (Zhong, R. et al., *Adv. Mater.*, 2018)². As shown in **Fig. 4c**, the hydrogel possessed a higher G' compared to G'' over the entire applied frequency range, suggesting the formation of a solid-like hydrogel. The purpose of **Fig. 4c** is to assess whether 8OHG-T hydrogel forms, so G' is not need to be much greater than G''.

References

1. Zhou, X., He, X., Shi, K., Yuan, L., Yang, Y., Liu, Q., Ming, Y., Yi, C. & Qian, Z. Injectable thermosensitive hydrogel containing Erlotinib - Loaded hollow mesoporous silica nanoparticles as a localized drug delivery system for NSCLC therapy. *Adv. Sci.*, **7**, 202001442 (2020).
2. Zhong, R.; Tang, Q.; Wang, S.; Zhang, H.; Zhang, F.; Xiao, M.; Man, T.; Qu, X.; Li, L.; Zhang, W. & Pei, H. Self-Assembly of Enzyme-Like Nanofibrous G-Molecular Hydrogel for Printed Flexible Electrochemical Sensors. *Adv. Mater.* **30**, e1706887 (2018).

Comment #10: Fig. 4a. caption no mention of terms "Pre" or "Sol" in the Fig. 4a.

Response: Thank you for your comments and we are very sorry for this error. We have deleted the terms "Pre" or "Sol" in the caption of original **Fig. 4a** (It is now shown in **Supplementary Fig. 8** in revised Supporting Information).

Comment #11: Fig. 4d, e caption seems do not match the figure?

Response: We apologize for the error that the same figures are given in **Fig. 4d, e** and **Fig. 5a, b**. We have corrected **Fig. 4d, e** (**Fig. R10** in the response letter) in the revised manuscript.

Fig. R10. The characterizations of hydrogels.

a Photographs of 8AG-T and 8OHG-T hydrogels prepared for 6 months. **b, c** Evolution of G' and G'' as a function of frequency for of 8AG-T (**b**) and 8OHG-T(**c**) hydrogels. **d, e** The self-healing of 8AG-T (**d**) and 8OHG-T(**e**) hydrogels by rheological measurements. **f** SEM (scale bar: 50 μm) images of 8AG-T and 8OHG-T hydrogels. **g** AFM (scale bar: 200 nm) images of 8AG-T and 8OHG-T hydrogels. **h, i** The pair distances distribution functions (PDDF) profiles from VT-SAXS experiments of 8AG-T (**h**) 8OHG-T (**i**) hydrogels.

Comment #12: Fig. 4j. Figure is missing

Response: We are very sorry for this error. The caption of **Fig. 4j** is redundant and has been deleted in the revised manuscript.

Comment #13: p15 "borate diester" was not introduced before. Give context.

Response: We deeply appreciate your helpful and very constructive comments. We have added the content of introducing "borate diester" in the revised manuscript.

Comment #14: Fig 5c inset illegible

Response: Thank you very much for your suggestion. We have revised **Fig. 5c** (The figure is assigned to be **Fig. R11** in the response letter) to make it clearer.

Fig. R11. Self-assembly mechanism of the cation-independent hydrogels.

a ^{11}B NMR spectra of 8AG-T and 8OHG-T hydrogels. **b** Fluorescence intensity of ARS in 8AG-T and 8OHG-T hydrogels. **c** ThT assay of 8AG-T and 8OHG-T hydrogels. **d** CD spectra of 8AG-T and 8OHG-T hydrogels. **e** The chemical structure and single crystal structure of **1**. **f** ^1H - ^1H NOE of 8AG-T hydrogels. **g** The single crystal structure of the base-pair pattern; **h** The schematic diagram of the single crystal of **6**. The red dashed box includes the interactions between DMSO and 8AG. **i** The PXRD spectrum of 8AG-T and 8OHG-T hydrogels. **j** Schematic illustration of the formation of an 8AG-T hydrogel. Atoms are coded as follows: red, oxygen; blue, nitrogen; gray, carbon; white, hydrogen.

Comment #15: Fig 6d. Completely wrong. Abs at 3 = no photon. Inset fit is inappropriately drawn and (voluntary) misleading

Response: Thank you for your professional comments. Initially, we considered the upper limit of absorbance in quantitative analysis of ultraviolet spectrophotometry during the experiments. As you know, the most basic and fundamental basis of quantitative analysis in ultraviolet spectrophotometry is the Lambert–Beer law, which is as follows:

$$A = \log (I_0/I_t) = \log (1/T) = k l C \quad (1)$$

where A is the absorbance, I_0 and I_t are the intensities of incident light and transmission light, T is the light transmittance, l and C are the thickness (1.0 cm) and the concentration (g/L) of the solution, and k is the absorbance coefficient.

As your professional comment suggests, T is only 0.1% as $A=3$, indicating transmission light is only 0.1% of incident light, and photons detected by instrument is 0.1% of photons of incident light. In this case, the ultraviolet spectrophotometer requires high sensitivity to detect photons accurately. With the development of ultraviolet spectrophotometer, its sensitivity to photon detection has improved significantly. Besides, the factors of non-monochromatic light, stray light, noise, and spectral bandwidth may make work curve deviate from the law of Lambert-Beer in the spectrum analysis. With the progress of technology, these factors have been minimized to improve the instrumental errors significantly. At present many new-generation ultraviolet spectrophotometers exhibit good photometric linearity with the absorbances up to 3 or even higher. For example, the specification sheet (**Fig. R12**) of the Cary 100 UV-Vis spectrophotometer (Agilent, USA) we used in our study shows that the photometric range of absorbance is up to 3.7. The specification sheet (**Fig. R13**) of the Cary 6000i UV-Vis spectrophotometer (Agilent, USA) even shows the photometric linearity of absorbance up to 9.

In addition, to verify the photometric linearity of the Cary 100 UV-Vis spectrophotometer we used with the absorbance up to 3.0 at 204 nm, we detected a series of the acidic potassium dichromate solutions, which are the

reference materials used for controlling absorbance of ultraviolet-visible spectroscopy in the United States Pharmacopoeia Convention (2023) ¹. The results demonstrate it has good linearity with the absorbance range of 0.39-3.03 at 204 nm (**Fig. R14**).

Taken together, although the absorbance in **Fig. 6d** is up to 3, the accuracy of the stoichiometric titration result is reliable. Thanks for your valuable comment. For avoiding misleading readers, we have transferred **Fig. 6d** from manuscript to **Supplementary Fig. 37**.

[REDACTED]

Fig. R12. The specification sheet of the Cary 100 UV-Vis spectrophotometer (Agilent, USA)

[REDACTED]

Fig. R13. The specification sheet of the Cary 6000i UV-Vis spectrophotometer (Agilent, USA)

Fig. R14. The ultraviolet spectrum. **a** The ultraviolet spectrum of the acidic potassium dichromate solutions with the concentrations of 10 – 90 mg L⁻¹. **b** The calibration curve of the acidic potassium dichromate solutions in ultraviolet spectrophotometry at 204 nm.

References:

1. United States Pharmacopeial Convention. Ultraviolet-Visible Spectroscopy. doi.org/10.31003/USPNF_M3209_04_01 (2023).

Comment #16: - Pages 23 and 24 (Mat & Meth): "Due to the limited size of the training dataset (n=71), we divided all the data into five copies. One copy of the data was selected as the test set, and the remaining four copies were selected as the training set. A 5-fold cross-validation was randomly performed 20 times to improve the accuracy of the fitting results."

I guess the authors mean "subsets" instead of "copies"? For cross-validation the training and test sets must be independent. Using multiple copies of the same data and using some of them as training and the others as tests would be an invalid procedure because the same data would be used in both sets.

Response: Thank you very much for the professional comments and we fully agree with this matter. What we really want to convey is that the 71 compounds were split into 80% (57) for the training set and 20% (14) for the test set. Fivefold stratified cross-validation was used to estimate the intervals of the parameters¹, which was widely used in the prediction model building (Enot, D. P. et al. *Nat. Protoc.*, 2008; Kohoutová, L. et al., *Nat. Protoc.*, 2020)²⁻³. The 71 nucleoside derivatives were divided into five equal parts by maintaining the ratio of gelators

to non-gelators and four of them were used as the training set to train the model and the other one as the test set. The model was trained five times without repetition. On top of this, 10 times random fivefold stratified cross-validations were performed to obtain more reliable information. In another word, 50 times of model training were performed.

According to your professional suggestions, we have corrected the related parts about this comment, and corrected "copied" to "subsets" in the revised manuscript.

Reference:

1. Kohavi, R. Proceedings of the 14th international joint conference on Artificial intelligence (Morgan Kaufmann Publishers Inc., Montreal, 1995).
2. Enot, D. P. *et al.* Preprocessing, classification modeling and feature selection using flow injection electrospray mass spectrometry metabolite fingerprint data. *Nat. Protoc.* **3**, 446-470 (2008).
3. Kohoutová, L. *et al.* Toward a unified framework for interpreting machine-learning models in neuroimaging. *Nat. Protoc.* **15**, 1399-1435 (2020).

Comment #17: Also, cross-validation is not done to improve accuracy, but to calculate or estimate it. Note that these sentences (and the next 3, until "...(the difference between the predicted and actual values)" are exactly repeated on pages 23 and 24.

Response: Thank you very much for your valuable comments, here is the description we used incorrectly. As you reminded us, we corrected to "fivefold stratified cross-validation was performed to estimate parameters".

Other minor details:

Comment #18: _Fix the language. There are sentences with unclear grammar at several places in the manuscript. For ex.:

Page 5: ... and four kinds of molecular fingerprints were used to present above nucleoside derivatives derived by SMILES.

Response: Thank you very much for your comments, and we've corrected the description in the revised manuscript.

Comment #19: Page 8: _Compared to the new important features concluded by ML model, the descriptors summarized by existing experience are not better predictors for the hydrogel-forming ability of nucleoside derivatives.

Response: Thank your detailed comments, we apologize for the confusing description in our manuscript, and we have changed it in the revised manuscript.

Comment #20: Page 4: "Firstly, a dataset of 71 nucleoside derivatives with hydrogel-forming ability": rephrase because some are gelators and some are not (page 5: "These nucleoside derivatives were then divided into two groups, namely, gelator (n=38) and nongelator (n=33)").

Response: Thank you for your comments, and we are very sorry for this error. We have corrected it in the revised manuscript.

Comment #21: Page 5: "Pearson correlation coefficients ($r < 0.8$)", whereas on page 24 (Mat & Meth) it says "We performed a Spearman's rank correlation test on the remaining descriptors to exclude one of the pairs of descriptors with correlations higher than 0.8."

Response: Thank you for your valuable comment, in fact, we used Spearman correlation. We have unified the corresponding descriptions in revised manuscript.

Comment #22: "Descripters" instead of "Descriptors" on Figures 1 and 2"descriptor-last" on Figure 2a: I guess it refers to the final descriptor with 24 parameters. Maybe write "descriptor-24"? Also, "descriptor-40" not shown.

Response: Thank you very much for your comment. Using "descriptor-last" to describe the four descriptors prediction models after feature elimination (RFE) is not easy to understand. In fact, different models have different optimal combinations of optimal descriptors according to the recursive feature elimination (RFE) based on different ML algorithms (XGBoost, n=16; LR, n =24; DT, n=30; RF, n=37). That's why we do not used descriptor-24 in this place. To make it easier for readers to understand, we changed it to Descriptor-RFE. Meanwhile, we have added "descriptor-40" in **Fig R15 (Fig. 2a** in revised manuscript) and corrected the related part of the revised manuscript and made notes to explain it under the corresponding figures and tables (**Table R4**).

Table R4. The corresponding updated part of **Supplementary Table 2**.

Models	Features	Test Accuracy		AUC#	
		Mean	Se	Mean	Se
DT	Descriptor_4175	0.65	0.01	0.65	0.02
LR	Descriptor_4175	0.65	0.02	0.67	0.02
RF	Descriptor_4175	0.63	0.01	0.72	0.02
XGBoost	Descriptor_4175	0.63	0.01	0.69	0.02
DT	Descriptor_144	0.64	0.01	0.64	0.01
LR	Descriptor_144	0.68	0.02	0.80	0.02
RF	Descriptor_144	0.67	0.01	0.75	0.02
XGBoost	Descriptor_144	0.64	0.02	0.72	0.02
DT	Descriptor_40	0.66	0.02	0.69	0.02
LR	Descriptor_40	0.70	0.01	0.81	0.02
RF	Descriptor_40	0.67	0.01	0.74	0.02
XGBoost	Descriptor_40	0.65	0.01	0.75	0.02
DT	Descriptor_REF*	0.59	0.02	0.63	0.02
LR	Descriptor_REF*	0.71	0.01	0.84	0.02
RF	Descriptor_REF*	0.67	0.01	0.75	0.02

XGBoost	Descriptor_REF*	0.70	0.02	0.79	0.02
---------	-----------------	------	------	------	------

Notes: *: Descriptors-REF: Recursive feature elimination (REF) has different optimal descriptors for different Algorithms: LR, n=24; XGBoost, n=16; DT, n=30; RF, n=37.

#: AUC: Area Under Curve

Fig R15. A scatterplot showed the distribution of AUC (area under the curve) and test accuracy for all models. The 4-point shapes represent different ML algorithm: extreme gradient boosting (XGBoost), logistic regression (LR), decision tree (DT), and random forest (RF). The colors represent eight different descriptors or fingerprints inputs. Descriptor's part: Initially obtained 4175 descriptors, 144 descriptors after rank sum test, 40 descriptors after correlation coefficient selection, and descriptors after recursive feature elimination (RFE). The optimal number of descriptors for RFE of each machine learning (ML) algorithm is different (XGBoost, n=16; LR, n=24; DT, n=30; RF, n=37). Fingerprint part: ECFP4, ECFP6, Atom Pair and Topological Torsion.

Comment #23: Figure 2b: the "Recall" and "Precision" metrics are not mentioned anywhere else in the manuscript. Maybe useful to remind what Recall/Precision/F1 score/AUC represent for non-ML expert readers?

Response: Thank you very much for your valuable suggestion, we have added the relevant description of recall, F1 score and precision in revised Supporting Information (**Supplementary Methods**).

Comment #24: Figure 4i: what are PDDF profiles?

Response: Thank you for your comments and we are very sorry for this error. "PDDF profiles" is the abbreviation of "the pair distances distribution functions profiles", which is from the variable-temperature small-angle X-ray scattering (VT-SAXS) measurements. We have changed "PDDF profiles" to "the pair distances distribution functions (PDDF) profiles" in the caption of **Fig. 4i**.

Comment #25:

Page 23: the reference to Scikit-learn is missing. Scikit-learn: Machine Learning in Python, Pedregosa et al., JMLR 12, pp. 2825-2830, 2011.

Page 24: "We first filtered descriptors with a Wilcoxon rank-sum test less than 0.05": rank-sum test p-value less than 0.05

Page 25: "Specific selection of nucleoside derivative structures" should be a title in bold. "The nucleoside derivative structures were selected from PubChem ()": missing reference.

"Data availability" section: "Source data are provided with this paper and <https://github.com/leescu/NHGPM>": remove as it is redundant with the previous sentence.

Page 31, Ref 30: Huang, J. et al. Identification of potent antimicrobial peptides via a machine-learning pipeline that mines the entire space of peptide sequences. Nature Biomedical Engineering, n/a, NA, (2023). Maybe include the doi: 10.1038/s41551-022-00991-2

Response: Thank you very much for your valuable comments. We are very sorry for these errors. According to your suggestions, we have corrected the related sections in revised manuscript.

Reviewer #3 (Remarks to the Author):

Comment #1: After the introduction, which is quite well-written, the results and discussion is littered with abbreviations and acronyms that do not make much sense to the reader... For example, the opening sentences of this section read: "To construct the prediction model, all the published nucleoside derivatives and their hydrogel-forming ability were collected by literature review, and 71 molecules were included 5,7,40-55. To unify the molecular structure, the Chemdraw software (Version 20.0) was utilized to reproduce the 71 molecular structures."

What does this mean? How many is "all published"? What does it mean to collect the hydrogel forming ability by literature review? Does this mean that every published nucleoside derivative has a documented ability to form, or not, a hydrogel?

What does it mean that 71 molecules were included?

How does one unify the molecular structure? And what does ChemDraw have to do with reproducing the molecular structures?

I could go on, but these two sentences are sufficient to illustrate the point that the results, methods and discussion of this work is not sufficiently clearly articulated to allow a realistic judgement on the quality of the work.

Response: Thanks for your valuable comments. We are sorry that our descriptions are not clear enough. We would like to reply to the comments on the following aspects.

1). What does this mean? How many is "all published"? What does it mean that 71 molecules were included?

The number of "all published" nucleoside derivatives is 71. And the purpose of including 71 molecules was to construct a dataset of nucleoside derivatives to build the prediction models. Here, the details of the screening process of "all published" nucleoside derivatives and the meaning of "71 molecules were included" are shown in the follows:

Firstly, Medical Subject Headings (MeSH) was used to collect all subjects and

free terms for nucleosides ("uridine", "thymidine", "adenosine", "guanosine", and "cytidine") and their derivatives, and a system-wide search was conducted on Medline, Web of Science, and SciFinder for all nucleoside-related studies. The specific information on the search strategy is shown in **Supplementary Data S8**.

Secondly, we read the titles, abstracts, and full texts of the literature obtained in the previous step. Only the studies providing information on whether nucleoside derivatives formed hydrogels under arbitrary conditions in pure water or aqueous solutions were selected for dataset construction. Then, the studies reported only the gelators in organic solvents were excluded. Finally, 71 nucleoside derivatives were collected as "all published" nucleoside derivatives from the literature after excluding duplicate nucleoside derivatives. Thirdly, a dataset was constructed based on the 71 nucleoside derivatives collected. This dataset including the structures of nucleoside derivatives and whether they have the hydrogel-forming ability was used to build the prediction model.

2). What does it mean to collect the hydrogel forming ability by literature review? Does this mean that every published nucleoside derivative has a documented ability to form, or not, a hydrogel?

The purpose of our study is to develop a ML model for accurate nucleoside hydrogel prediction. A dataset including the structures of nucleoside derivatives and whether they have the hydrogel-forming ability is necessary for the prediction model. Considering there is no such a dataset at present, we screened the published nucleoside derivatives which have the information of hydrogel-forming ability by literature review to build a new dataset.

Therefore, each of the nucleoside derivatives in the dataset has a documented ability to form or not form a hydrogel. The nucleoside derivatives were divided into gelator and non-gelator groups according to documented hydrogel-forming ability. Documented ability to form or not form a hydrogel is usually defined clearly in literatures. However, some substances were just used as references

or controls and were often not explicitly specified as gelators or not. They could be determined based on the results of tube-inversion tests or rheological measurements.

3) How does one unify the molecular structure? What does ChemDraw have to do with reproducing the molecular structures?

The molecular structures are generally unified by ChemDraw, where all molecules can be presented and characterized uniformly (Li, Z. J. et al., *J. Chem. Inf. Comput. Sci.*, 2004)¹. Nucleoside is a 3D chiral molecule, so only when it is unified can differences be found. Furthermore, we collected the structures of nucleoside derivatives from different journals. Since the requirements of journals are different, their molecules express in different forms. Therefore, unification of the molecular structure is a necessary step.

ChemDraw is a powerful software for designing chemical and biological compositions and has been used to convert 2D molecular diagrams into SMILES strings. It is used to draw 2D molecular diagrams of molecules and convert them into SMILES strings in many studies (Li, Z. J. et al., *J. Chem. Inf. Comput. Sci.*, 2004)¹, and has been widely applied to build ML models (Tiihonen, A. et al., *Journal of the American Chemical Society*, 2021; Lyu, R. et al., *Journal of the American Chemical Society*, 2021)²⁻³.

Therefore, in this study, we redrawn all the 2D molecular diagrams of nucleoside derivatives based on the structures provided in the published studies, using the ChemDraw function (**Fig. R16**). Thank you again for your valuable comments, we have revised the relevant content in the revised manuscript.

Fig. R16. Flowchart of unify the molecular structures.

Reference:

1. Li, Z. J., Wan, H. G., Shi, Y. H. & Ouyang, P. K. Personal experience with four kinds of chemical structure drawing software: Review on ChemDraw, ChemWindow, ISIS/Draw, and ChemSketch. *J. Chem. Inf. Comput. Sci.* **44**, 1886-1890 (2004).
2. Tiihonen, A. et al. Predicting Antimicrobial Activity of Conjugated Oligoelectrolyte Molecules via Machine Learning. *J. Am. Chem. Soc.* **143**, 18917-18931 (2021).
3. Lyu, R., Moore, C. E., Liu, T., Yu, Y. & Wu, Y. Predictive Design Model for Low-Dimensional Organic-Inorganic Halide Perovskites Assisted by Machine Learning. *J. Am. Chem. Soc.* **143**, 12766-12776 (2021).

Comment #2: There are other significant issues with the manuscript that raises some real concerns for this reviewer:

1. There are a lot of references but a number of these are not really related to the thing being referenced. For example, reference 57 is a Nature paper on the development of an ML model for planning chemical syntheses. But the authors use it to reference a molecular fingerprint method (ECFP4) which is indeed used in the article but is not defined there. Equally, the literature on the prediction and discovery of hydrogels is not representative of the major steps in this field.

Response: Many thanks for this comment and we apologize for the mistake, we have checked carefully and revised the references of the revised manuscript. The reference of ECFP4 was corrected as Reference 20, ECFP6 (Reference 20), Atom Pair (Reference 21) and Topological Torsion (Reference 22) were also changed the references; The reference of discovery of hydrogels was corrected as Reference 4; The references of prediction of hydrogels were corrected as Reference 17-19. Detailed of example correction please see below:

1). The reference of molecular fingerprint

20. Rogers D, Hahn M. Extended-Connectivity Fingerprints. J. Chem. Inf. Model. 50, 742-754 (2010).

21. Carhart RE, Smith DH, Venkataraghavan R. Atom pairs as molecular features in structure-activity studies: definition and applications. J. Chem. Inf. Comput. Sci. 25, 64-73 (1985).

22. Nilakantan R, Bauman N, Dixon JS, Venkataraghavan R. Topological torsion: A new molecular descriptor for SAR applications. Comparison with other descriptors. J. Chem. Inf. Comput. Sci. 27, 82-85 (1987).

2). The reference of discovery of hydrogels

4. Bang I. Examination of the guanyle acid. Biochem. Z. 26, 293-311 (1910).

3). The references of prediction of hydrogels

17. Gupta JK, Adams DJ, Berry NG. Will it gel? Successful computational prediction of peptide gelators using physicochemical properties and molecular

fingerprints. Chem. Sci. 7, 4713-4719 (2016).

18. Li F, et al. Design of self-assembly dipeptide hydrogels and machine learning via their chemical features. Proc. Natl. Acad. Sci. U. S. A. 116, 11259-11264 (2019).

19. Van Lommel R, Zhao J, De Borggraeve WM, De Proft F, Alonso M. Molecular dynamics based descriptors for predicting supramolecular gelation. Chem. Sci. 11, 4226-4238 (2020).

Comment #3: The authors make some claims that are not evidenced in their results - e.g., "Under the appropriate conditions, three hydrogels display a long lifetime stability of 6 months (1 and 2 in H₃BO₃ and Tris solutions, and 3 in AgNO₃ solution, Fig. 3c-e)". But Figure 3c-e only show photos of the hydrogels at a single time point? How is this 6 months timepoint to be evidenced?

Response: Thank you for your suggestive comments. We are very sorry that the photographs of hydrogels prepared for 6 months weren't shown in the manuscript before. We have rearranged the photographs of hydrogels and added the photographs of hydrogels prepared for 6 months to prove the lifetime stability of 6 months in the revised manuscript. Here, the photographs of 8AG-T, 8OHG-T, and 8aza-Ag⁺ hydrogels prepared for 6 months were shown in **Fig. R17**.

Fig. R17. Photographs of 8AG-T, 8OHG-T, and 8AzaG-Ag⁺ hydrogels prepared for 6 months.

Comment #4: 3. The results shown in Figure 4d-e, are identical to the results in Figure 5a-b. This is probably a simple mistake as the figure captions indicate that the Figure 4 panels should do something else. But this carelessness is concerning and make it very difficult to judge the veracity of the results.

Response: Thank you for pointing this out and we are very sorry for this error. We have corrected **Fig. 4d-e** (The figure is assigned to be **Fig. R18** in the response letter) in the revised manuscript.

Fig. R18. The characterizations of hydrogels.

a Photographs of 8AG-T and 8OHG-T hydrogels prepared for 6 months. **b, c** Evolution of G' and G'' as a function of frequency for of 8AG-T (**b**) and 8OHG-T(**c**) hydrogels. **d, e** The self-healing of 8AG-T (**d**) and 8OHG-T(**e**) hydrogels by rheological measurements. **f** SEM (scale bar: 50 μm) images of 8AG-T and 8OHG-T hydrogels. **g** AFM (scale bar: 200 nm) images of 8AG-T and 8OHG-T hydrogels. **h, i** The pair distances distribution functions (PDDF) profiles from VT-SAXS experiments of 8AG-T (**h**) 8OHG-T (**i**) hydrogels.

Reviewers' Comments:

Reviewer #1 (Remarks to the Author):

The previous version of this paper had a surprisingly large number of clarifications, errors, and omissions and I persevered because of the novelty of the work. The authors have provided comprehensive responses to the many points that I and my fellow reviewer have raised.

However, the clarifications provided by the authors have revealed a number of serious issues that were not apparent in the first draft. The most serious of these are that 5-fold cross-validation, not an independent test set, was used to assess the productivity of the models, and that proper feature selection was not performed (or not performed at all for the fingerprint descriptors) to ensure the model were not overfitted. Overfitting would have been identified if an independent test set had been used. Specifically...

1. the authors should not refer to the 20% of the data set aside as a test set (14 data points) because they were actually the 20% taken aside in 5-fold cross-validation. As the data set is small, the use of cross-validation is probably justified, although it would be important to see how well the final models could predict a proper independent test set.
2. Feature selection was performed on the descriptors but not on the 2048 fingerprint descriptors. With only 57 data points in the training set when cross-validation is done. the number of features should be at most 30. None of the models presented have less than 40 features and the fingerprint models use 2048 features. Clearly this is a grossly overfitted situation so none of the models really have any likelihood of being valid or useful.
3. I suspect that when the number of features is reduced to an acceptable number that the performance of all ML methods will be very similar
4. The LASSO and MLREM feature selection methods were tried but dismissed because they did not generate models quite as good as the overfitted models. These methods should be used to generate an acceptably sparse feature subset of features for both the descriptor and fingerprint features.
5. The 83% accuracy for the 12 nucleoside derivatives from the screening of a larger number of possibilities is misleading. The 'ranking' and 'probabilities' of gelling are very unconvincing as all nucleosides have very similar probability values close to the arbitrary decision boundary. Essentially this means that they only got one compound 'wrong'. However, almost all compounds were gel forming.
5. It is good that the authors used chemical similarity measure to choose the 7000 compounds for screening. However, the correct way to do this is to calculate the relevant descriptors for them all (which must have been done to make predictions) and see whether these multidimensional features fell in or near the ranges for the training set.

Reviewer #2 (Remarks to the Author):

The new version and responses from the authors answer to the remarks

Reviewer #3 (Remarks to the Author):

[Note from the Editor] This referee was unable to provide a report but made comments to the editor only.

Response:

Point-by-point comments.

Comment #1. *The authors should not refer to the 20% of the data set aside as a test set (14 data points) because they were actually the 20% taken aside in 5-fold cross-validation. As the data set is small, the use of cross-validation is probably justified, although it would be to see how well the final models could predict a proper independent test set.*

Response: Many thanks to the reviewer #1 for her/his comments, but we don't agree this comment. Although, we didn't used a proper **independent test set** to test how well the final models could predict. But we have selected externally 12 nucleoside derivatives from PubChem, and used experiment to evaluate whether the prediction performance of final model is good or not. Moreover, chose 20% (14) from 71 nucleoside derivative could not be an independent test set. All these 71 nucleoside derivatives were derived from the same dataset constructed by our literature search. We believe that continuing to split such a small data set will further degrade the performance of the model, when 20% of nucleoside derivatives are randomly selected at one time as the test set, the randomness is very large, and the results of such test set are hardly to be considered stable and reliable. Furthermore, some similar studies with small sample size also use similar method, instead of random select an independent test set in addition to the 5-fold cross-validation, they use experiments to verify as we do (Lyu, R. *et al.*, *J. Am. Chem. Soc.*, 2021; Kohoutová, L. *et al.*, *Nat. Protoc.*, 2020¹⁻²). Given that this study is a preliminary exploration of machine learning (ML) to predict the hydrogel-forming ability of nucleoside derivatives, and such a small data set, we believe that our scheme is also a viable option. The success rate of hydrogel-forming ability verification experiment of 12 nucleoside derivatives is 83.33% (10/12), which we believe the performance of final model is acceptable.

References:

- 1 Tiihonen, A. *et al.* Predicting Antimicrobial Activity of Conjugated Oligoelectrolyte Molecules via Machine Learning. *J. Am. Chem. Soc.* **143**, 18917-18931 (2021).
- 2 Kohoutová, L. *et al.* Toward a unified framework for interpreting machine-learning models in neuroimaging. *Nat. Protoc.* **15**, 1399-1435 (2020).

Comment #2. Feature selection was performed on the descriptors but not on the 2048 fingerprint descriptors. With only 57 data points in the training set when cross-validation is done. the number of features should be at most 30. None of the models presented have less than 40 features and the fingerprint models use 2048 features. Clearly this is a grossly overtained situation so none of the models really have any likelihood of being valid or useful.

Response: Thank the reviewer #1 for giving her/his comments to us, but we are curious about why the reviewer #1 said so. We have reason to think that the reviewer #1 did not look carefully at our response letter and manuscripts. Reviewer #1 mentioned that **"the number of features should be at most 30. None of the models presented have less than 40 features"**. But all the models after 3-step feature selection have less than 40 features (**logistic regression**, LR, n for features =**24**; **extreme gradient boosting**, XGBoost, n for features =**16**; **decision tree**, DT, n for features = **30**; **random forest**, RF, n for features =**37**), and the optimal model based on only **24** features, which conforms to the reviewer's comment. The information about the descriptor models have been provided in the **response letter of version 1 (Reviewer #1, Comment #1, Response part 2):** "The results indicated that the optimal number of descriptors for each ML algorithm is different (XGBoost, n=16; LR, n=24; DT, n=30; RF, n=37)". The manuscript also have provided details of the features information of all models (**Page 7 line 16-18 in Results and Discussion; Table R1**, assigned to be **Supplementary Table 1 in Supporting Information; Fig. 2a** annotation; **Supplementary Data S4**). In addition, the reason why molecular fingerprints are not screened for features is also explained in the **response letter of version 1 (Reviewer #1, Comment #1, Response part 1):** "Unlike the molecular descriptors, the individual bit string of fingerprints doesn't have specific chemical meanings, so we didn't perform the feature selection for the fingerprints". Our feature selection is indeed performed only on descriptors because we consider that only the complete molecular

fingerprint is representative of the internal structure of the compound. Further feature selection of molecular fingerprints is not only meaningless for finding the specific descriptors closely related to the hydrogel-forming ability, but also loses the original chemical or structure meaning of complete fingerprints, which is the reason why we do not perform a feature selection on fingerprints.

Table R1. Characteristics of the prediction models

Features	Algorithms	Rows	Columns
Molecular Descriptors			
Descriptors-4175	LR, DT, RF, XGBoost [#]	71 [*]	4175
Descriptors-144	LR, DT, RF, XGBoost	71	144
Descriptors-40	LR, DT, RF, XGBoost	71	40
Descriptors-REF ^{&}	LR, DT, RF, XGBoost	71	16-37 ^{&}
Molecular Fingerprints			
ECP4	LR, DT, RF, XGBoost	71	2048
ECP6	LR, DT, RF, XGBoost	71	2048
AtomPair	LR, DT, RF, XGBoost	71	2048
Topological Torsion	LR, DT, RF, XGBoost	71	2048

Notes: ^{*} Rows: Whether the nucleoside derivatives have the hydrogel-forming ability; [#]Algorithms: Logistic regression (LR), decision tree (DT), random forest (RF), and extreme gradient boosting (XGBoost); [&]Descriptors-REF: Recursive feature elimination (REF) has different optimal descriptors for different Algorithms: LR, n=24; XGBoost, n=16; DT, n= 30; RF, n=37.

Comment #3. *I suspect that when the number of features is reduced to an acceptable number that the performance of all ML methods will be very similar.*

Response: Thank the reviewer #1 for giving her/his comments to us. Indeed, as the reviewer #1 said, after feature selection, only referring to AUC and test accuracy, the performance of some ML models is similar (LR and XGBoost). However, we have also explained in our **response letter of version 1** why LR is used as the optimal model (**Reviewer #1, Comment #3, Response part 1**): "Firstly, test accuracy, AUC, precision, recall and F1 score are commonly used to determine the optimal model in related research (Theodoris, C. V. et al., Nature, 2023, AUC and F1 score; Jablonka, K. M. et al., Nat. Chem., 2021, accuracy, precision, and recall; Han, T. et al., Nat. Mach. Intell., 2022, AUC, precision and recall)¹⁻³, the five parameters were used to evaluate the performance of models comprehensively here. To select the optimal model, we mainly focused on test accuracy and AUC, and the results of precision, recall and F1 score were also used as auxiliary references. In this study, LR not only provided better results of test accuracy (0.71 ± 0.01) and AUC (0.84 ± 0.02), but also had higher recall (0.95 ± 0.01) and F1 score (0.78 ± 0.01) (Table R2; Fig. R1, the figure is assigned to be Fig. 2b in revised manuscript). So, we finally chose LR as the optimal model". Firstly, the five evaluation indexes (test accuracy, AUC, precision, recall and F1 score) were used to evaluate the performance of models comprehensively here. To select the optimal model, we mainly focused on test accuracy and AUC, and the results of precision, recall and F1 score were also used as auxiliary references. In this study, LR not only provided better results of test accuracy (0.71 ± 0.01) and AUC (0.84 ± 0.02), but also had higher recall (0.95 ± 0.01) and F1 score (0.78 ± 0.01) (Table R2; Fig. R1, the figure is assigned to be Fig. 2b in revised manuscript). So, we finally chose LR as the optimal model. Either refer to Fig. R1, Table R2, or perform any statistical difference test (DeLong test). Considering all the five

evaluation indexes' performance of the model after feature selection, these four ML methods will not be **very similar**. In fact, contrary to the suspicion of the reviewer #1, the performance of the models was similar without feature selection (Descriptor_4175, **Table R1**) in our study, but the difference between the performance of the models became more obvious after reducing the number of features by feature selection (Descriptor_REF, **Table R1**).

Fig. R1. Evaluation indexes of 4 models using descriptors after recursive feature elimination.

Table R2. The descriptor models' part of **Supplementary Table 2.**

Models	Features	Test Accuracy		AUC	
		Mean	Se	Mean	Se
DT*	Descriptor_4175	0.65	0.01	0.65	0.02
LR	Descriptor_4175	0.65	0.02	0.67	0.02
RF	Descriptor_4175	0.63	0.01	0.72	0.02
XGBoost	Descriptor_4175	0.63	0.01	0.69	0.02
DT	Descriptor_144	0.64	0.01	0.64	0.01
LR	Descriptor_144	0.68	0.02	0.80	0.02
RF	Descriptor_144	0.67	0.01	0.75	0.02
XGBoost	Descriptor_144	0.64	0.02	0.72	0.02
DT	Descriptor_40	0.66	0.02	0.69	0.02
LR	Descriptor_40	0.70	0.01	0.81	0.02
RF	Descriptor_40	0.67	0.01	0.74	0.02
XGBoost	Descriptor_40	0.65	0.01	0.75	0.02
DT	Descriptor_REF#	0.59	0.02	0.63	0.02
LR	Descriptor_REF#	0.71	0.01	0.84	0.02
RF	Descriptor_REF#	0.67	0.01	0.75	0.02
XGBoost	Descriptor_REF#	0.70	0.02	0.79	0.02

Notes: *: Logistic regression (LR), decision tree (DT), random forest (RF), and extreme gradient boosting (XGBoost);

#: Descriptors-REF: Recursive feature elimination (REF) has different optimal descriptors for different Algorithms: LR, n=24; XGBoost, n=16; DT, n= 30; RF, n=37.

References:

1. Theodoris, C. V. et al. Transfer learning enables predictions in network biology. *Nature* 618, 616–624 (2023).
2. Jablonka, K. M., Ongari, D., Moosavi, S. M. & Smit, B. Using collective knowledge to assign oxidation states of metal cations in metal–organic frameworks. *Nat. Chem.* **13**, 771-777 (2021).
3. Han, T. et al. Image prediction of disease progression for osteoarthritis by style-based manifold extrapolation. *Nat. Mach. Intell.* **4**, 1029-1039 (2022).

Comment #4. *The LASSO and MLREM feature selection methods were tried but dismissed because they did not generate models quite as good as the overfitted models. These methods should be used to generate an acceptably sparse feature subset of features for both the descriptor and fingerprint features.*

Response: Thank the reviewer #1 for giving her/his comments about LASSO and MLREM, but we don't agree her/his comments about feature selection methods. We believe that the reviewer's claim our model "**overfits**" is unfounded and emotional. LASSO and MLREM have indeed been used for feature selection of ML models for some studies, but this does not mean that it is wrong to use methods such a 3-step feature selection without LASSO or MLREM. Referring to the previous studies, the three step of feature selection method has been widely used in molecular property prediction models (Tiihonen, A. et al., *J. Am. Chem. Soc.*, 2021; Lyu, R. et al., *J. Am. Chem. Soc.*, 2021; Batra, R. et al., *Nat. Mach. Intell.*, 2020¹⁻³). Meanwhile, under the same conditions, the performance of the model with LASSO and MLREM feature selection has not reached the effect of our 3-step feature selection. After 3-step feature selection, we believe that the number of features of the models have been reduced to an acceptable range for the reviewer #1 ($n < 40$ or ≤ 30 , according from **Comment #2**).

References:

1. Tiihonen, A. et al. Predicting Antimicrobial Activity of Conjugated Oligoelectrolyte Molecules via Machine Learning. *J. Am. Chem. Soc.* **143**, 18917-18931 (2021).
2. Lyu, R., Moore, C. E., Liu, T., Yu, Y. & Wu, Y. Predictive Design Model for Low-Dimensional Organic–Inorganic Halide Perovskites Assisted by Machine Learning. *J. Am. Chem. Soc.* **143**, 12766-12776 (2021).
3. Batra, R., Chen, C., Evans, T. G., Walton, K. S. & Ramprasad, R. Prediction of water stability of metal–organic frameworks using machine learning. *Nat. Mach. Intell.* **2**, 704-710 (2020).

Comment #5. The 83% accuracy for the 12 nucleoside derivatives from the screening of a larger number of possibilities is misleading. The 'ranking' and 'probabilities' of gelling are very unconvincing as all nucleosides have very similar probability values close to the arbitrary decision boundary. Essentially this means that they only got one compound 'wrong'. However, almost all compounds were gel forming.

Response: We appreciate for the reviewer #1 to give us comments, but we still do not accept reviewer #1's this comment. As the reviewer #1 said, "**all nucleosides have very similar probability values**". But the reviewers seem not to have considered the objective circumstances of the study. Verification through experiments requires cost and time. We cannot verify all nucleoside derivatives predicted to be the hydrogel one by one. If ranking and probability are not considered, it is difficult to select part of thousands of nucleoside derivatives predicted to be gelled in a more reasonable way for verification. This study is the initial exploration to use ML to predict hydrogel-forming ability of nucleoside derivatives, and we believe that it is not unacceptable or misleading to prioritize a subset of nucleosides for verification based on probability and ranking. We used a logical scheme, chose 12 nucleoside derivatives with high probability (top 10%) of hydrogel-forming ability for verification. The vial inversion experiment also confirmed the hydrogel-forming ability, and we believed that this was an appropriate and effective way. In addition, we are curious why the reviewer #1 said "**Essentially this means that they only got one compound 'wrong'** ", as in the manuscript and response letter, we all stated that we got two 'wrong' compounds among the 12 nucleoside derivatives ($10/12 = 83.3\%$).

Although the present prediction model is our initial attempt, we still hope that the present model can be applied to the prediction of nucleoside hydrogels. This study greatly accelerates the discovery of new nucleoside hydrogels compared to previous inadvertent discoveries or modifications of existing gelators.

Comment #6. It is good that the authors used chemical similarity measure to choose the 7000 compounds for screening. However, the correct way to do this is to calculate the relevant descriptors for them all (which must have been done to make predictions) and see whether these multidimensional features fell in or near the ranges for the training set.

Response: We appreciate for the reviewer #1 to give us comments. Since the molecular descriptors are calculated based on the molecular structure, we believe that the structure selected by 3D similarity is bound to fall in or near the ranges in the feature distribution of molecular descriptors. We calculated the multidimensional features (24 molecular descriptors of the final model) of the 7257 compounds according to the reviewer #1 comment. A grouped box plot (**Fig. R2**, as **Supplementary Fig. 6** in revised manuscript) of the 24 descriptors in optimal model (LR) and a PCA plot (**Fig. R3**, as **Supplementary Fig. 7** in revised manuscript) based on these 24 descriptors, all showed that the 24 descriptors's distribution of 7257 nucleoside derivatives are near the 71 nucleoside derivatives constructed the prediction model.

Fig.R2. A grouped box plot of 24 molecular descriptors (optimal model) for nucleoside derivatives. Nucleoside derivatives are divided into two groups, including nucleoside derivatives from PubChem dataset (Pubchem group, n=7257) and all published nucleoside gelators and nongelators (Published group, n=71).

Fig.R3. 2D-PCA results of 24 features grouped by nucleoside derivatives, including PubChem dataset (Pubchem group, n=7257), published gelators (Gelator group, n=38) and published nongelators (Nongelator group, n=33).

REVIEWER COMMENTS

Reviewer #1 (Remarks to the Author):

1. The authors have significantly improved the manuscript by ensuring that the predictions of the 7000+ nucleoside analogues as geleators/nom-geletors are near or within the domain of applicability of the chosen model, and by exploring the use of sparse L1 feature selection methods LASSO and MLREM. It is interesting that these linear methods generated models with similar (albeit slightly inferior) performance to the other nonlinear ML methods. In light of this, it would be most appropriate to use the linear models to interpret the importance of the descriptors to gelation. As was pointed out earlier, feature importance for nonlinear models is a local, not global property, so depends on where on the response surface it is measured (this should be mentioned in the manuscript). I strongly recommend that the LASSO or MLREM regression coefficients be used to determine the sign and magnitude of the contribution of each sparse feature to the model. The authors could choose to compare these feature importances with those from the nonlinear models if they wish but the linear model feature importance will be the most relevant.

2. The authors have also not understood the issue with gross overfitting of models that I raised in the last review. In their rebuttal, they focused on the fact that feature selecture of descriptor-based models did indeed reduce the number of feature employed to significantly fewer than the number of training examples (good). However, they still insist on using 2048 features for the fingerprint-based models (or is it 4096 features given 4 types of 1024 fingerprints?). Clearly with between 57 and 71 training examples, such a large number of features will grossly overfit the models. This is not "unfounded and emotional" but a statistical fact. I recommend they either use sparse feature selection to reduce the number of features drastically, or they remove the models based on fingerprints entirely. These models were not used to predict the larger set of nucleoside analogues so don't add any value to the manuscript. Fingerprints can in fact have chemical meaning (at least as much as the arcane molecular descriptors used in the final model) but this point is moot given that the models are grossly overfitted and are no better than the models trained on sparse molecular descriptors.

3. Clearly, the point about the authors choosing 20% of the data set randoinly 10 times to define a 'test set' has not been made strongly enough in the last review. Normal practice, at least with larger data sets, is to partition it into a training set used to construct the model and a test set used to assess model predictive power for data it has not been trained on. The test set is chosen once and never used in training. The method employed by the authors is essentially 'leave-20%-out' cross validation which overestimates model predictivity because the training and test set are not independent. Given the relatively small data set size, the use of cross validation rather than a test set may be justified. However a test set of 20% chosen by cluster analysis (once) to ensure it covers the same domain as the training set, would be a better and more stringent measure of model predictivity and I strongly recommend the autjors include this in the revised manuscript. The authors are correct in that the final arbiter of model utilty is on predicting the properties and an external set, as they have done.

4. Continuing on with the prediction of the 7000+ set, although the screened molecules appear to be within or near the domain of applicability for the chosen model (good), it would be more persuasive if the authors had additionally chosen 12 analogues predicted to have no gel-forming ability and measured those subsequently. Can this be done for the revised manuscript?
5. The manuscript still has ample evidence of lack of care in preparation with still a substantial number of spelling and grammatical errors after multiple rounds of review e.g., 'Descriptors' in Fig. 1, 'Flow chat', 'Dabase' etc etc.
6. Supplementary Table 2 should be in the body of the paper.
7. The last paragraph of the Introduction is a summary of the results, delete this and leave it to the Results and Discussion to describe this. Similarly, pages 6-8 is simply a repeat of Methods so could be deleted here and left for the Methods section to explain.
8. Suppl Table 3 suggests only 6-7 features are important given the standard errors of these (F-test)
9. The use of DT is probably redundant as ensemble methods like RF and XGB will always perform better than DT.

Reviewer #4 (Remarks to the Author):

This is a paper by Hang Zhao and colleagues who developed a machine learning model to predict the gelation characteristics of nucleoside derivatives. The authors used data from the literature to initially train a new model then used this model to inform a few new designs for subsequent testing. This paper was very comprehensive and impressive. At first I was concerned by the large number of descriptors being used, but was later happy to see the authors reduce the number of valuable descriptors to a more reasonable number. It is interesting that logistic regression performed so well in this environment. Another area of strength was the in depth analysis of the two cation-independent hydrogels (8AG-T and 8OHG-T). Overall, the strength of the model training and subsequent analysis gives this reviewer much to praise about this paper and its methods. Also, the authors did a great job responding to the first reviewer's comments. My only suggestion would be to include a summary schematic at the beginning of the paper to help readers understand the main goals of the paper quickly. Otherwise, I am happy for this paper to be published without further revision.

Reviewer #1

Comment #1: The authors have significantly improved the manuscript by ensuring that the predictions of the 7000+ nucleoside analogues as geleators/nom-geletors are near or within the domain of applicability of the chosen model, and by exploring the use of sparse L1 feature selection methods LASSO and MLREM. It is interesting that these linear methods generated models with similar (albeit slightly inferior) performance to the other nonlinear ML methods. In light of this, it would be most appropriate to use the linear models to interpret the importance of the descriptors to gelation. As was pointed out earlier, feature importance for nonlinear models is a local, not global property, so depends on where on the response surface it is measured (this should be mentioned in the manuscript). I strongly recommend that the LASSO or MLREM regression coefficients be used to determine the sign and magnitude of the contribution of each sparse feature to the model. The authors could choose to compare these feature importance's with those from the nonlinear models if they wish but the linear model feature importance will be the most relevant.

Response:

Thank you very much for the professional comments and suggestions. As suggested by reviewer, regression coefficients of linear model methods such as LASSO or MLREM are better used as feature importance ^[1,2] and can provide valuable information. Therefore, we have revised the manuscript as follows:

Firstly, we provided the regression coefficients of the optimal model based on 24 molecular descriptors (logistic regression of L1 penalty terms) as the feature importance of the optimal model in the revised manuscript (**Fig. R1**, as **Fig. 2e** in the revised manuscript). There are only four important features (Coef > ± 0.1), which is basically consistent with the previous results. And we have briefly elaborated on these descriptors in this manuscript (**Supplementary Discussion 1.1**); Secondly, the results of permutation feature importance (PFI, as **Fig. 2e** and **Supplementary Table 2** in the previous manuscript) are reserved into **Supplementary materials (Fig. R2, Table R1, as Supplementary Fig. 8 and Supplementary Table 2** in the revised manuscript). Thirdly, as a reference, we added the regression coefficients of LASSO based on 4175 molecular descriptors as the feature importance for the descriptors (as **Supplementary Data S6** in the revised manuscript).

Perhaps for reasons we do not explain specifically, our feature importance results are actually based on logistic regression (L1 penalty), also a generalized linear model^[3]. This is also in line with the reviewer's opinion that " *the linear model feature importance will be the most relevant*". While, we used the PFI method to calculate the influence of each feature on the average accuracy of the model in the previous manuscript. We are in favour of the reviewer's

reference “*regression coefficients be used to determine the sign and magnitude of the contribution of each sparse feature to the model*”, and have made corresponding modification in the revised manuscript (**Results and Discussion**: Page 9 line 1 to line 9; **Methods**: Page 35 line 5 to line 13).

Fig. R1. The feature importance of 24 descriptors for logistic regression based on the regression coefficients.

Fig. R2. The feature importance of 24 descriptors for logistic regression based on the permutation feature importance.

Table R1. The feature importance of 24 descriptors for logistic regression.

Descriptors Information		Feature importance		
Name	Description	PFI ^{&}		Coefficient [*]
		Mean	Se	
CATS2D_06_DL	Pharmacophore descriptors	0.018	0.005	-0.090
B09[O-O]	2D Atom Pairs	0.018	0.003	0.155
P_VSA_charge_7	P_VSA-like descriptors	0.014	0.003	-0.083
H-052	Atom-centred fragments	0.014	0.003	-0.133
CATS2D_03_DL	Pharmacophore descriptors	0.011	0.004	-0.072
nN(CO)2	Functional group counts	0.009	0.004	0.131
CATS2D_04_AA	Pharmacophore descriptors	0.006	0.005	0.053
CATS2D_05_DA	Pharmacophore descriptors	0.006	0.005	0.057
C-016	Atom-centred fragments	0.004	0.005	-0.099
F07[N-O]	2D Atom Pairs	0.001	0.003	0.085
VE1sign_Dz(v)	2D matrix-based descriptors	0.000	0.006	-0.054
CATS2D_05_DL	Pharmacophore descriptors	-0.001	0.003	-0.087
F05[N-N]	2D Atom Pairs	-0.001	0.003	0.092
P_VSA_charge_4	P_VSA-like descriptors	-0.001	0.003	-0.103
F10[O-O]	2D Atom Pairs	-0.002	0.005	-0.083
MATS3p	2D autocorrelations	-0.004	0.005	-0.055
VE3sign_D/Dt	2D matrix-based descriptors	-0.005	0.004	-0.082
SM10_AEA(dm)	Edge adjacency indices	-0.005	0.005	-0.076
SpDiam_AEA(ed)	Edge adjacency indices	-0.006	0.005	-0.068
VE1sign_B(p)	2D matrix-based descriptors	-0.008	0.003	0.057
GATS6i	2D autocorrelations	-0.008	0.003	0.059
SpMAD_EA(ri)	Edge adjacency indices	-0.012	0.003	0.067
GATS7s	2D autocorrelations	-0.014	0.007	0.077
CATS2D_09_DA	Pharmacophore descriptors	-0.016	0.008	0.076

&: Permutation feature importance to calculate the mean accuracy decrease of logistic regression model

*: Regression coefficient of logistic regression model.

[1] Guidotti R., Monreale A., Ruggieri S., Turini F., Giannotti F., Pedreschi D. A survey of methods for explaining black box models. *ACM Comput Surv* **51**(5):1–42 (2018).

[2] Saarela M., Jauhiainen S. Comparison of feature importance measures as explanations for classification models. *SN. Appl. Sci.* **3**: 272 (2021).

[3] Huang J., Zhang CH. Estimation and Selection via Absolute Penalized Convex Minimization and Its Multistage Adaptive Applications. *Journal of Machine Learning Research.* **13**:1839-1864 (2012).

Comment #2: The authors have also not understood the issue with gross overfitting of models that I raised in the last review. In their rebuttal, they focused on the fact that feature selecture of descriptor-based models did indeed reduce the number of features employed to significantly fewer than the number of training examples (good). However, they still insist on using 2048 features for the fingerprint-based models (or is it 4096 features given 4 types of 1024 fingerprints?). Clearly with between 57 and 71 training examples, such a large number of features will grossly overfit the models. This is not "unfounded and emotional" but a statistical fact. I recommend they either use sparse feature selection to reduce the number of features drastically, or they remove the models based on fingerprints entirely. These models were not used to predict the larger set of nucleoside analogues so don't add any value to the manuscript. Fingerprints can in fact have chemical meaning (at least as much as the arcane molecular descriptors used in the final model) but this point is moot given that the models are grossly overfitted and are no better than the models trained on sparse molecular descriptors.

Response:

Thank you very much for the professional comments. We agree with the reviewer that using 2048 features in "between 57 and 71 training examples" may lead to overfitting of the model. So we completely removed the part of the models based on the molecular fingerprints of 2048 features.

The reason why we used four molecular fingerprints in previous manuscript were to evaluate whether the prediction models based on descriptors have a good performance in predicting the hydrogel-forming ability of nucleoside derivatives. And we considered that the dimensionality reduction of the molecular fingerprint model may lose the original chemical or structure meaning of complete fingerprints. As mentioned in rebutter letter of NCOMMS-23-16267B-Z (Comment #2, Response part 1): "Further feature selection of molecular fingerprints is not only meaningless for finding the specific descriptors closely related to the hydrogel-forming ability, but also loses the original chemical or structure meaning of complete fingerprints". While we have verified the performance of the model based on experiments, we believe that removing the models based on molecular fingerprints could not undermine the integrity of the manuscript, so we have removed the results about molecular fingerprints.

Comment #3: Clearly, the point about the authors choosing 20% of the data set randomly 10 times to define a 'test set' has not been made strongly enough in the last review. Normal practice, at least with larger data sets, is to partition it into a training set used to construct the model and a test set used to assess model predictive power for data it has not been trained on. The test set is chosen once and never used in training. The method employed by the authors is essentially 'leave-20%-out' cross validation which overestimates model predictivity because the training and test set are not independent. Given the relatively small data set size, the use of cross validation rather than a test set may be justified. However, a test set of 20% chosen by cluster analysis (once) to ensure it covers the same domain as the training set, would be a better and more stringent measure of model predictivity and I strongly recommend the authors include this in the revised manuscript. The authors are correct in that the final arbiter of model utility is on predicting the properties and an external set, as they have done.

Response:

We thank the referee for this comment. We also followed the reviewer's suggestion and add corresponding content to the revised manuscript, that is, the test set of 20% nucleoside derivative data is selected through cluster analysis (once, not for training), and the remaining 80% is used as the training set (with five-fold cross validation). The results showed that, consistent with our previous five-fold cross validation of 71 nucleoside derivatives, the three-step feature screening based logistic regression (LR-RFE) performed better in both the training set (with five-fold cross validation) and test set. In the training set, validation accuracy: 0.70 ± 0.02 , AUC: 0.84 ± 0.02 (**Table R2**, as **Supplementary Table 3** in the revised manuscript); In the test set, accuracy: 0.67, AUC: 0.81 (**Table R3** as **Table 2** in the revised manuscript). The corresponding parts have been added to the revised manuscript for reference (**Results and Discussion**: Page 8 line 13 to line 24; **Methods**: Page 33 line 22 to Page 34 line 4)

Detailed description:

We reduced dimensionality by PCA for 4175 molecular descriptors of 71 nucleoside derivatives, and 71 nucleoside derivatives were clustered by k-means (**Fig. R3**, as **Supplementary Fig. 5** in the revised manuscript). The K value was determined based on the inertia method and silhouette score (K=4, **Fig. R4**, as **Supplementary Fig. 6** in the revised manuscript). To ensure that the test set covered the same areas as the training set, we conducted stratified sampling based on clustering and gelling results, dividing nucleoside derivatives into 80% as training set (n=56) and 20% as test set (n=15) (**Fig. R5**, as **Supplementary Fig. 7** in the revised manuscript). We use 5-fold cross validation to train and hyperparameter the model on the training set, and evaluate the model's generalization ability on the additional test set.

For larger data sets, choosing 20% as a separate test set ensures that it covers the same areas as the training set. But considering that this study is based on

only 71 nucleoside derivatives, as we mentioned in the rebutter letter, we need to note that continuing to segment such a small data set will further degrade the performance of the model when 20% of nucleoside derivatives are randomly selected as the test set at once, even after passing the cluster analysis. We still need to emphasize that there is inevitably a lot of instability with this approach, and that this 20% test set is not an independent external data set. Therefore, in our revised manuscript, even though we added the method of randomly selecting 20% nucleoside derivatives as the test set as a sensitivity analysis (results were showed in the supplementary), we still used the cross-validated optimal model for the prediction of the external set, referring to the methods commonly used in similar studies with small sample size^[1,2].

We have to admit that the current data set size is relatively small. However, as a first attempt, this prediction model have also achieved meaningful results, many new nucleoside hydrogels have been discovered based on a certain accuracy (83.3%). For better accumulation in the future, we have established an online database: NHGPM (www.nhgpm.com, **Fig. R6**), which recorded a range of chemical properties of collected nucleoside derivatives, including the hydrogel-forming ability. We are also uploading the results of our prediction model into this online database. Readers could obtain the collected information of nucleoside derivatives and also could choose the nucleoside derivatives they want to further study according to the hydrogel-forming probability provided by us. All this information is freely available for scientific researchers on our website. We hope that our efforts can promote the development of this field. As the research progresses in the field of nucleoside hydrogels, more records on the hydrogel-forming ability of nucleoside derivatives will be collected. We will further optimize our model based on larger data sets in the future.

Table R2. The result of AUC (Area under Curve) and validation accuracy for all models based on training set .

Models	Features	training set performance			
		Validation accuracy		AUC	
		Mean	Se	Mean	Se
DT*	Descriptor_4175	0.68	0.02	0.68	0.02
LR	Descriptor_4175	0.58	0.02	0.58	0.03
RF	Descriptor_4175	0.64	0.02	0.72	0.02
XGBoost	Descriptor_4175	0.63	0.02	0.66	0.02
DT	Descriptor_119	0.62	0.02	0.62	0.01
LR	Descriptor_119	0.64	0.02	0.80	0.02
RF	Descriptor_119	0.67	0.02	0.74	0.02
XGBoost	Descriptor_119	0.64	0.02	0.68	0.02
DT	Descriptor_34	0.67	0.02	0.67	0.02
LR	Descriptor_34	0.70	0.01	0.84	0.02
RF	Descriptor_34	0.68	0.01	0.75	0.02
XGBoost	Descriptor_34	0.67	0.01	0.73	0.02
DT	Descriptor_REF#	0.63	0.02	0.65	0.02
LR	Descriptor_REF#	0.70	0.01	0.84	0.02
RF	Descriptor_REF#	0.67	0.02	0.75	0.02
XGBoost	Descriptor_REF#	0.67	0.02	0.74	0.02

Notes: *: Logistic regression (LR), decision tree (DT), random forest (RF), and extreme gradient boosting (XGBoost).

#: Descriptors-REF: Recursive feature elimination (REF) has different optimal descriptors for different Algorithms: LR, n=34; XGBoost, n=33; DT, n= 23; RF, n=26.

Table R3. The result of AUC (Area under Curve) and validation accuracy for all models based on test set.

Models	Features	Test set performance				
		Accuracy	F1 Score	Precision	Recall	AUC
DT	Descriptor_REF#	0.60	0.67	0.75	0.60	0.60
LR	Descriptor_REF#	0.67	0.76	1.00	0.61	0.81
RF	Descriptor_REF#	0.53	0.59	0.63	0.56	0.53
XGBoost	Descriptor_REF#	0.60	0.57	0.50	0.67	0.61

Notes: *: Logistic regression (LR), decision tree (DT), random forest (RF), and extreme gradient boosting (XGBoost).

#: Descriptors-REF: Recursive feature elimination (REF) has different optimal descriptors for different Algorithms: LR, n=34; XGBoost, n=33; DT, n= 23; RF, n=2

Fig. R3. Results of principal component analysis (PCA)

Fig. R4. Results of cluster analysis of K-means . a. result of inertia, b. result of inertia, c. result of K-means.

Fig. R5. The distribution of training set and test set

Fig. R6. The home page of www.nhgpm.com

[1]. Tiihonen, A. et al Predicting Antimicrobial Activity of Conjugated Oligoelectrolyte Molecules via Machine Learning. *J. Am. Chem. Soc.* **143**, 18917-18931 (2021).

[2]. Kohoutová, L. et al Toward a unified framework for interpreting machinelearning models in neuroimaging. *Nat. Protoc.* **15**, 1399-1435 (2020).

Comment #4. Continuing on with the prediction of the 7000+ set, although the screened molecules appear to be within or near the domain of applicability to the chosen model (good), it would be more persuasive if the authors had additionally chosen 12 analogues predicted to have no gel-forming ability and measured those subsequently. Can this be done for the revised manuscript?

Response:

Thank you for your constructive comments. According to your suggests, we have additionally chosen 12 nucleoside derivatives with low hydrogel-forming probability to validate they have no gel-forming ability. The validation methods are consistent with the methods of validating the hydrogel-forming ability in this study, and are as follows: first, the 12 nucleoside derivatives (**13**, 5,6-Dichlorobenzimidazole riboside, DRB; **14**, 9-(2-tetrahydropyranyl)adenine, 9-THPA; **15**, 9-(2-tetrahydrofuryl)adenine, 9-THFA; **16**, 2-thiocytidine, 2-TC; **17**, 2',3'-dideoxy-2'3'-didehydroadenosine, 2',3'-DA; **18**, 2',5'-dideoxyadenosine, 2',5'-DA; **19**, 2'-C-methyladenosine, 2'-MeA; **20**, gemcitabine, GCTB; **21**, 2-chloro-2',3'-O-isopropylideneadenosine-5'-N-ethylcarboxamide, 2-CIA; **22**, 2-chloro-9-(2-tetrahydropyranyl)adenine, 2-Cl-9-THPA; **23**, 7-deaza-2'-C-methyladenosine, 7-D-2'-MeA; **24**, 2'-C-methylcytidine, 2'-MeC) with low gelling prediction probability were selected in a relatively homogeneous manner based on the costs of obtaining and synthesizing. Then, the tube-inversion tests were performed to validate whether they could form hydrogels in the presence of alkali metal cations, borate anions, or metal ions. The results showed that 10 of the 12 nucleoside derivatives didn't form hydrogels, while the two others formed (**Table R4, Fig. R7-R10 as Supplementary Data S8 and Supplementary Fig. 16-19** in the revised manuscript), demonstrating the rate of not forming hydrogels is 83.33%. The validation results indicated that the nucleoside derivatives with low gelling prediction probability have low gel-forming ability. In total, the accuracy of the optimal model for predicting hydrogel-forming ability was 83,33% (20/24) for 24 nucleoside molecules which further confirmed the reliability of our ML models. Thank you again for your valuable comments, we have added the relevant content in the revised manuscript (**Abstract:** Page 2 line 8 to line 10; **Results and Discussion:** Page 16 line 3 to line 7, Page 16 line 22 to line 25, Page 17 line 1 to line 3; **Conclusion:** Page 30 line 4 to line 5; **Methods:** Page 36 line 23 to Page 37 line 11)

Fig. R7. Photographs of hydrogels or samples assembled from nucleoside derivatives in different solutions. Sol: solution. Pre: precipitate.

Fig. R8. Photographs of hydrogels or samples assembled from nucleoside derivatives in different solutions. Sol: solution. Pre: precipitate.

Fig. R9. Photographs of hydrogels or samples assembled from nucleoside derivatives in different solutions. Sol: solution. Pre: precipitate.

Fig. R10 12 nucleoside derivatives with low probability of hydrogel-forming ability were selected. The result shows 10 nucleoside derivatives (14-23) formed hydrogels, while the two others (13 and 24) did not.

Table R4. The validation for the hydrogel-forming ability of the nucleoside derivatives (**13 - 24**)

No.	Nucleoside derivatives	PMID	P for Gelability	Rank for Gelability	Test Result
13	DRB* 	5894	0.452	6400 (88.2%)	Gel (+)
14	9-THPA 	1932	0.449	6473 (89.2%)	Gel (-)
15	9-THFA 	5270	0.448	6507 (89.7%)	Gel (-)
16	2-TC 	3011746	0.447	6534 (90.0%)	Gel (-)
17	2',3'-DA 	64975	0.444	6585 (90.7%)	Gel (-)
18	2',5'-DA 	65166	0.443	6592 (90.8%)	Gel (-)
19	2'-MeA 	500900	0.430	6803 (93.7%)	Gel (-)
20	GCTB 	60750	0.424	6872 (94.7%)	Gel (-)

21	2-CIA		14775452	0.418	6949 (95.8%)	Gel (-)
22	2-Cl-9-THPA		12777819	0.415	6980 (96.2%)	Gel (-)
23	7-D-2'-MeA		3011893	0.391	7144 (98.4%)	Gel (-)
24	2'-MeC		500902	0.379	7188 (99.1%)	Gel (+)

Notes: *: **13**, 5,6-Dichlorobenzimidazole riboside, DRB; **14**, 9-(2-tetrahydropyranyl)adenine, 9-THPA; **15**, 9-(2-tetrahydrofuryl)adenine, 9-THFA; **16**, 2-thiocytidine, 2-TC; **17**, 2',3'-dideoxy-2'3'-didehydroadenosine, 2',3'-DA; **18**, 2',5'-dideoxyadenosine, 2',5'-DA; **19**, 2'-C-methyladenosine, 2'-MeA; **20**, gemcitabine, GCTB; **21**, 2-chloro-2',3'-O-isopropylideneadenosine-5'-N-ethylcarboxamide, 2-CIA; **22**, 2-chloro-9-(2-tetrahydropyranyl)adenine, 2-Cl-9-THPA; **23**, 7-deaza-2'-C-methyladenosine, 7-D-2'-MeA; **24**, 2'-C-methylcytidine, 2'-MeC.

Comment #5. The manuscript still has ample evidence of lack of care in preparation with still a substantial number of spelling and grammatical errors after multiple rounds of review e.g., 'Descriptors' in Fig. 1, 'Flow chat', 'Dabase' etc etc.

Response:

Thank you very much for your reminding. We have reviewed the manuscript and corrected the errors.

Comment #6. Supplementary Table 2 should be in the body of the paper.

Response:

Thank you very much for the professional comments, we have included previous **supplementary Table 2** in the body of the paper (As **Table 1** in the revised manuscript).

Comment #7. The last paragraph of the Introduction is a summary of the results, delete this and leave it to the Results and Discussion to describe this. Similarly, pages 6-8 is simply a repeat of Methods so could be deleted here and left for the Methods section to explain.

Response:

Thanks for your comment and we fully agree with this matter. We have modified the corresponding part. Firstly, for the last paragraph of the **Introduction**, we have deleted this and leaved it to the **Results and Discussion**. Secondly, for the content on **pages 6-8**, In order to make it easier for readers to understand, we have completely cut down the content, kept only a few result descriptions, and left for the **Methods** and **Supplementary Methods 2.1** to explain. (**Introduction**: Page 5 line 10 to line 14; **Results and Discussion**: Page 6 line 4 to line 25, Page 10 line 3 to line 7, Page 28 line 14 to line 18)

Comment #8. Suppl Table 3 suggests only 6-7 features are important given the standard errors of these (F-test)

Response:

Thanks for your comments, as you noted, given the standard error of these features, only a few descriptors are important. In addition, following your suggestion, we focus more on feature importance based on regression coefficients in the revised manuscript. According to the regression coefficients, there are only four important features ($\text{Coef} > \pm 0.1$), which is basically consistent with the previous results. And we have briefly elaborated on these descriptors in this manuscript (**Supplementary Discussion 1.1**)(**Results and Discussion**: Page 9 line 1 to line 9; **Methods**: Page 35 line 5 to line 13).

Comment #9. The use of DT is probably redundant as ensemble methods like RF and XGB will always perform better than DT.

Response:

Many thanks for this comment. As you said, RF, XGB, and LR all tend to perform better than DT. However, as a typical machine learning model [1], DT has also been widely used in previous studies to predict the properties of compounds [2], and we believe that it is also feasible to retain it as a comparison. Second, the results of DT may also provide a reference for readers, if DT is removed, only three kinds of machine learning models were left, which may seem not convincing enough. So we think that we may keep the results of DT in the revised manuscript.

[1]. Quinlan JR. Induction of decision trees. *Mach. Learn.* 1, 81-106 (1986).

[2]. Batra, R. et al. Machine learning overcomes human bias in the discovery of self-assembling peptides. *Nat. Chem.* 14, 1427-1435, (2022).

Reviewer #4

(Remarks to the Author):

This is a paper by Hang Zhao and colleagues who developed a machine learning model to predict the gelation characteristics of nucleoside derivatives. The authors used data from the literature to initially train a new model then used this model to inform a few new designs for subsequent testing. This paper was very comprehensive and impressive. At first I was concerned by the large number of descriptors being used, but was later happy to see the authors reduce the number of valuable descriptors to a more reasonable number. It is interesting that logistic regression performed so well in this environment. Another area of strength was the in depth analysis of the two cation-independent hydrogels (8AG-T and 8OHG-T). Overall, the strength of the model training and subsequent analysis gives this reviewer much to praise about this paper and its methods. Also, the authors did a great job responding to the first reviewer's comments. My only suggestion would be to include a summary schematic at the beginning of the paper to help readers understand the main goals of the paper quickly. Otherwise, I am happy for this paper to be published without further revision.

Response:

Thank you very much for your approval of our work, and taking the time to review our manuscript. Following your suggestion, we have also included a summary schematic in the introduction (Fig. R11, as Graphic abstract in the revised manuscript).

Fig. R11 An optimal ML model was constructed for nucleoside derivatives hydrogel-forming ability prediction, and potential gelators were selected based on the optimal model external application and the hydrogel-forming ability were experimentally verified. Besides, the self-assembly mechanism of the cation-independent hydrogel was explored, which could be applied in rapid visual detection of Ag⁺ and cysteine.

REVIEWERS' COMMENTS

Reviewer #1 (Remarks to the Author):

The authors have made a series attempt to address the remaining, important deficiencies in their submitted manuscript. The paper must be revised again to correct residual grammar and spelling issues as there are still a considerable number of errors. In particular there is a chunk of manuscript missing from the PDF between lines 324 and 412, and 787-920, line 615 should read 'self-assembly', incorrect spelling of descriptors in Figure 1 despite this having been flagged in prior reviews.

Reviewer #4 (Remarks to the Author):

Authors have addressed this reviewer's main concern.

Reviewer #1

Comment #1: The authors have made a series attempt to address the remaining, important deficiencies in their submitted manuscript. The paper must be revised again to correct residual grammar and spelling issues as there are still a considerable number of errors. In particular there is a chunk of manuscript missing from the PDF between lines 324 and 412, and 787-920, line 615 should read 'self-assembly', incorrect spelling of descriptors in Figure 1 despite this having been flagged in prior reviews.

Response:

Thank you very much for your reminding. We have reviewed the manuscript and corrected all the errors. We have revised the manuscript as follows:

1). The paper must be revised again to correct residual grammar and spelling issues as there are still a considerable number of errors:

We have corrected residual grammar and spelling issues in revised manuscript and Supplementary Information, such as:

- a). Used the present tense to discuss the current work in the abstract. (**Abstract**)
- b). Replaced the "**descripters**" with "**descriptors**" in Figure 1.
- c). Replaced the "**the PFI method for the accuracy of the optimal model was also provided**" with "**the PFI results (mean accuracy decrease) of the 24 molecular descriptors in the optimal model were also provided**". (Page 7 Line 7-8)
- d). Added a "**for**" before "**external application**". (Page 9 Line 1 in revised manuscript)
- e). Replaced the "**is generated**" with "**generates**".(Page 11 Line 14 in revised manuscript)
- f). Added a "**for**" before "**external application**". (Page 13 Line 9 in revised manuscript)
- g). Replaced the "**self-assemble**" with "**self-assembly**". (Page 18 Line 20 in revised manuscript)
- h). Replaced the "**self-assemble**" with "**self-assembly**". (Page 18 Line 20 in revised manuscript)

2). In particular there is a chunk of manuscript missing from the PDF between lines 324 and 412, and 787-920

We are sorry that there was an unknown error when we uploaded the manuscript, which resulted in some contents not being displayed. We have

made the corresponding changes, and now it can be displayed correctly.

3). line 615 should read 'self-assembly':

We have replaced the "**self-assemble**" with "**self-assembly**"

4). Incorrect spelling of descriptors in Figure 1 despite this having been flagged in prior reviews:

We are very sorry that we did not correct it in time and we have revised it in the latest manuscript.

We sincerely thank your effort for improving our work, and taking the time to review our manuscript.